# Achieving Dimension-Free Communication in Federated Learning via Zeroth-Order Optimization

**Zhe Li**[1*], **Bicheng Ying**[2*], **Zidong Liu**[3], **Chaosheng Dong**[4], **Haibo Yang**[1]

[1]Rochester Institute of Technology, Rochester, NY 14623, USA
[2]Google Inc., Los Angeles, CA 90034, USA
[3]ComboCurve Inc., Houston, TX 77005, USA
[4]Amazon.com Inc., Seattle, WA 98109, USA
`zl4063@rit.edu, ybc@google.com, z.liu@combocurve.com,`
`chaosd@amazon.com, hbycis@rit.edu`

## Abstract

Federated Learning (FL) offers a promising framework for collaborative and privacy-preserving machine learning across distributed data sources. However, the substantial communication costs associated with FL significantly challenge its efficiency. Specifically, in each communication round, the communication costs scale linearly with the model's dimension, which presents a formidable obstacle, especially in large model scenarios. Despite various communication-efficient strategies, the intrinsic dimension-dependent communication cost remains a major bottleneck for current FL implementations. This paper proposes a novel dimension-free communication algorithm – DeComFL, which leverages the zeroth-order optimization techniques and reduces the communication cost from $\mathcal{O}(d)$ to $\mathcal{O}(1)$ by transmitting only a constant number of scalar values between clients and the server in each round, regardless of the dimension $d$ of the model parameters. Theoretically, in non-convex functions, we prove that our algorithm achieves state-of-the-art rates, which show a linear speedup of the number of clients and local steps under standard assumptions. With additional low effective rank assumption, we can further show that the convergence rate is independent of the model dimension $d$ as well. Empirical evaluations, encompassing both classic deep learning training and large language model fine-tuning, demonstrate significant reductions in communication overhead. Notably, DeComFL achieves this by transmitting only around 1MB of data in total between the server and a client to fine-tune a model with billions of parameters. The code is available at `https://github.com/ZidongLiu/DeComFL`.

## 1 Introduction

Federated Learning (FL) is a promising distributed machine learning framework that enables a large number of clients to collaboratively train a model under the orchestration of a central server (Kairouz et al., 2021; McMahan et al., 2017). By allowing clients to train models locally without sharing their raw data, FL offers a privacy-preserving distributed learning paradigm. Thanks to these advantages, FL has become a popular learning paradigm used in many applications, such as healthcare (Xu et al., 2021) and edge devices (Nguyen et al., 2021; Wang et al., 2021), among others.

Despite its benefits, FL often encounters challenges to its efficiency due to *expensive communication costs*. Specifically, in one communication round, the server needs to broadcast the global model to all participating clients, and each of these clients is expected to transmit the newest local model to the server for global aggregation (McMahan et al., 2017). In other words, the communication costs for one participating client *scale linearly with the model dimension*, presenting a prohibitively expensive communication overhead for FL systems, especially in large model and/or low communication speed scenarios. More specifically, on the one hand, foundation models in language and vision, such as GPT-

---

*Equal Contribution; Corresponding Author: Haibo Yang

3 (Brown et al., 2020), and other models (Bommasani et al., 2021), scale with billions of parameters, leading to a tremendous total communication burden. For example, fine-tuning GPT-J-6B on 10 billion tokens with a batch size of 262K tokens across four machines would involve transferring 915.5 TB of data throughout the entire training process (Wang et al., 2023b). On the other hand, the typical communication speed for FL is several Mbps in wireless environments and up to several hundred Mbps in wired connections. Given that communication costs increase linearly with model size, this presents a significant challenge in model training and fine-tuning in the FL scenario. To achieve communication-efficient FL, several techniques have been developed, including lazy aggregation or multiple local update steps (McMahan et al., 2017), various compression techniques (Bernstein et al., 2018; Vogels et al., 2019; Yang et al., 2021; Wang et al., 2022; Hönig et al., 2022; Yi et al., 2024; Reisizadeh et al., 2020; Huang et al., 2023; Li & Li, 2023; Haddadpour et al., 2021), and client sampling strategies (Ribero & Vikalo, 2020). While these methods can reduce certain communication costs, their communication costs still scale linearly with the model dimension for each participating client in one communication round. This intrinsic *dimension-dependent communication cost* remains a major challenge for current FL systems, particularly in the era of large deep learning models.

In this paper, we propose a novel FL approach to achieve dimension-free communication per round, leveraging zeroth-order optimization techniques (Nesterov & Spokoiny, 2017; Ghadimi & Lan, 2013; Liu et al., 2020). We exploit a unique property of zeroth-order gradients: they can be decomposed into a gradient scalar (magnitude) and a perturbation vector (direction). The gradient scalar is computed by using the finite difference of function values, while the perturbation vector can be generated identically across clients from a shared random seed. Therefore, instead of transmitting entire model parameters, we can *communicate gradient scalars and random seeds to reconstruct full gradients*, resulting in constant communication costs per round. A closely related idea is proposed in (Yue et al., 2023) that projects the first-order gradient into a random direction to get a scalar, which can be viewed as a linear approximation of the ZO method. They only considered a distributed shared model case. However, in the FL setting, where clients collaborate to learn a global model, *simply transmitting seeds and gradient scalars to reconstruct gradients is insufficient to guarantee convergence to the desired global model*, as detailed in a later section. Hence, we propose a novel algorithm DeComFL, deviating from traditional FL appearance while achieving the same objective.

Although the dimension-dependent communication cost per round being addressed, the total communication cost, which is the product of the number of rounds and the communication cost per round, might still be proportional to the model size. This is because the worst-case convergence rate of ZO methods is known to depend on the model dimension (Nesterov & Spokoiny, 2017; Duchi et al., 2015). Fortunately, existing works have shown that the loss landscape of deep learning lies in a very low-dimensional subspace, where the Hessian of the loss has a remarkably low effective rank (Papyan, 2018; 2020; Malladi et al., 2023). By leveraging the low effective rank assumption, we rigorously show that DeComFL achieves a *convergence rate independent of the model dimension*. To the best of our knowledge, this is the first systematic attempt to achieve dimension-free communication per client in FL within each communication round and total communication cost.

Our main results and contributions are summarized as follows:

- We propose DeComFL, a novel dension-free communication in federated learning framework via zeroth-order optimization. In each round, both the downlink (model pulling) and uplink (model uploading) communications involve transmitting only a constant number of scalar values between the participating clients and the server. This dramatically reduces the communication cost from $\mathcal{O}(d)$ to $\mathcal{O}(1)$ in both uplink and downlink, where $d$ is the dimension of the model parameters.

- Theoretically, in non-convex functions, we prove that DeComFL achieves $\mathcal{O}(\sqrt{d/mPKR})$ under the standard conditions, where $m$ is the number of participating clients in one communication round, $P$ is the number of simultaneous perturbations, $K$ is the number of local update steps, and $R$ is the number of communication rounds. This rate highlights the linear speedup in terms of the local update step, the number of perturbations and the clients.

- Under the $\kappa$-effective rank assumption, we further prove that DeComFL achieves a dimension-free convergence rate of $\mathcal{O}(\sqrt{\kappa/mPR})$. Combined with an $\mathcal{O}(1)$ communication cost per round, the total communication cost can be established as dimension-free. To the best of our knowledge, this is the first work to achieve this rate in the distributed FL setting.

- Comprehensive experiments on both training and fine-tuning tasks demonstrate that DeComFL achieves comparable performance to existing algorithms while significantly reducing communication costs by several orders of magnitude. For instance, by traditional methods, fine-tuning an

OPT-1.3B (Zhang et al., 2022) model in an FL setting requires transmitting approximately 10 GB per round between each client and the server. In contrast, DeComFL requires only 1MB of total communication throughout the entire fine-tuning process.

## 2 RELATED WORK

**Communication-Efficient Federated Learning**: Initially, (McMahan et al., 2017) proposed the FedAvg algorithm, which uses multiple local update steps to reduce the frequency of model transfers between the server and the client, thereby lowering the total communication cost. Since then, various techniques have been developed to further optimize communication efficiency, with most approaches involving compression methods. For instance, sparsification (Han et al., 2020; Li et al., 2020; Ozfatura et al., 2021; Wang et al., 2023a; Tang et al., 2022), quantization (Hönig et al., 2022; Huang et al., 2023; Haddadpour et al., 2021; Shlezinger et al., 2020; Jhunjhunwala et al., 2021; Bouzinis et al., 2023; Liu et al., 2023; Zakerinia et al., 2024), and low-rank approximations (Vogels et al., 2019; Martin & Mahoney, 2021). However, the communication cost per round between a client and the server remains dependent on the model dimension. Taking Top-$\Bbbk$ as an example (Stich et al., 2018), only the top $\Bbbk$ largest coordinates in the gradient vector are selected for communication. Theoretically, the convergence rate of Top-$\Bbbk$ depends on both the model dimension $d$ and the hyper-parameter $\Bbbk$. In practice, the choice of $\Bbbk$ is linearly scaled with the model dimension $d$, i.e., $\Bbbk = c \times d$, where $c$ is a constant such as 0.001 (Shi et al., 2019). Despite the success of these methods, the intrinsic dimension-dependent communication cost remains a major bottleneck for current FL systems, especially in the era of large deep learning models. In this work, our DeComFL achieves a constant $\mathcal{O}(1)$ communication cost for uplink and downlink transmissions by zeroth-order optimization.

**Zeroth-Order Optimization (ZOO)**: ZOO relies solely on function evaluations, making it ideal for scenarios where explicit gradient computation is impractical, expensive, or unreliable, such as in black-box optimization (Liu et al., 2020; Cai et al., 2021; Nikolakakis et al., 2022) and reinforcement learning (Liu et al., 2020; Jing et al., 2024; Li et al., 2021). Recently, ZOO has shown significant memory advantages in deep learning due to requiring only forward propagation (Malladi et al., 2023; Zhang et al., 2024). However, existing work has not fully exploited ZOO's potential to reduce communication costs in FL, as we propose in this work. For example, FedZO (Fang et al., 2022) applies zeroth-order (ZO) stochastic gradient estimation in FedAvg, achieving a convergence rate of $\mathcal{O}(\sqrt{d/mKR})$ in non-convex cases, but its communication complexity remains $\mathcal{O}(d)$ per round, the same as FedAvg. Similarly, BAFFLE (Feng et al., 2023) employs ZOO to achieve $\mathcal{O}(P)$ communication complexity in the uplink, but the downlink communication complexity remains $\mathcal{O}(d)$.

## 3 DIMENSION-FREE COMMUNICATION IN FEDERATED LEARNING

### 3.1 PRELIMINARY OF THE ZEROTH-ORDER OPTIMIZATION AND FEDERATED LEARNING

As with most standard FL settings, we assume that there exist $M$ clients in total in our FL system. Our goal is to minimize the global loss function $f$ which can be formulated as,

$$\min_{\boldsymbol{x}\in\mathbb{R}^d} f(\boldsymbol{x}) = \min_{\boldsymbol{x}\in\mathbb{R}^d} \frac{1}{M}\sum_{i=1}^{M} f_i(\boldsymbol{x}) \quad \text{where } f_i(\boldsymbol{x}) := \mathbb{E}\left[f_i(\boldsymbol{x};\xi_i)\right], \tag{1}$$

where $\boldsymbol{x}$ is a $d$-dimensional model parameter and $f_i$ represents the loss function on client $i$. The loss function is the expectation of a stochastic loss function $f_i(\boldsymbol{x};\xi_i)$, where $\xi_i$ is sampled from different local data distributions known as data heterogeneity in FL. The typical FL algorithm comprises three steps in each round: 1) The server initially samples a set of clients and sends the current global model to them. 2) Upon receiving the global model, each client performs multiple local updates based on this model and then transmits the updated local model back to the server. 3) The server aggregates all the returned local models from the clients and updates the global model accordingly.

More specifically, when applying the (stochastic) ZO method for the local update in the classical FedAvg (McMahan et al., 2017), it becomes FedZO (Fang et al., 2022). The main recursion of server model $\boldsymbol{x}_r$ and client models $\{\boldsymbol{x}_{i,r}^k\}$ can be summarized into the following forms:

$$\boldsymbol{x}_{i,r}^1 = \boldsymbol{x}_r, \quad \forall i \in C_r \hspace{3cm} \text{(Pull Model)} \tag{2a}$$

$$\boldsymbol{x}_{i,r}^{k+1} = \boldsymbol{x}_{i,r}^k - \eta g_{i,r}^k \cdot \boldsymbol{z}_{i,r}^k, \quad k = 1, 2, \cdots, K \hspace{1cm} \text{(Local Update)} \tag{2b}$$

$$\boldsymbol{x}_{r+1} = \frac{1}{|C_r|} \sum_{i \in C_r} \boldsymbol{x}_{i,r}^{K+1}, \qquad \text{(Aggregate Model)} \qquad (2\text{c})$$

where we use the superscript $k$ for the local update step, $r$ for the communication round, $i$ for the client index, and $C_r$ for a set of sampled client indices, $\boldsymbol{z}_{i,r}^k$ typically for a random direction vector drawing either from either Gaussian or uniform ball distribution. $g_{i,r}^k$ is a gradient scalar calculated as

$$g_{i,r}^k = \frac{1}{\mu} \big( f_i(\boldsymbol{x}_{i,r}^k + \mu \boldsymbol{z}_{i,r}^k; \xi_{i,r}^k) - f_i(\boldsymbol{x}_{i,r}^k; \xi_{i,r}^k) \big), \qquad (3)$$

where $\mu > 0$ is the smooth parameter. Intuitively, when $\mu$ is sufficiently small, the scalar $g_{i,r}^k$ approximates the gradient $\nabla f_i(\boldsymbol{x}_{i,r}^k; \xi_{i,r}^k)$ inner product with the random direction $\boldsymbol{z}_{i,r}^k$. There are other types of ZO gradient estimators. However, in this paper, we only focus on this (3) form called Simultaneous Perturbation Stochastic Approximation (SPSA) (Spall, 1992) with the forward style. See (Nesterov & Spokoiny, 2017; Liu et al., 2020) for other forms and convergence properties.

When we examine the communication costs incurred between the client and the server within the framework outlined in equations (2a) to (2c), it becomes apparent that a vector of length $2d$ is transmitted during each round. Specifically, the first $d$-length vector is transmitted during the pull model step via the downlink, while the second $d$-length vector is sent after finishing all local update steps and before aggregating at the server via the uplink. In the era of LLMs, this $2d$ communication cost can be a huge burden or even prohibitively expensive for scenarios requiring low latency. This observation motivates us to design a novel FL framework wherein the lengths of the vectors communicated via both the uplink and downlink are independent of the model dimension $d$.

### 3.2 ELIMINATING DIMENSION-DEPENDENT COMMUNICATION IN THE UPLINK

The equations (2a)-(2c) are merely a straightforward substitution of the first-order method with its zeroth-order counterpart. We can further exploit the zeroth-order property to lower the uplink communication cost. It is worth noting that the random vector $\boldsymbol{z}_{i,r}^k$ is generated using a pseudo-random number generator. Consequently, given a specific seed, $\boldsymbol{z}_{i,r}^k$ can be reproduced for a vector of any length. To exploit this property, we can reformulate (2b)-(2c) as

$$\boldsymbol{x}_{i,r}^{k+1} = \boldsymbol{x}_{i,r}^k - \eta g_{i,r}^k \cdot \boldsymbol{z}_r^k, \quad k = 1, 2, \cdots, K, \qquad \text{(Local Update)} \qquad (4\text{a})$$

$$\boldsymbol{x}_{r+1} = \boldsymbol{x}_r - \eta \sum_{k=0}^{K-1} \left( \frac{1}{|C_r|} \sum_{i \in C_r} g_{i,r}^k \right) \cdot \boldsymbol{z}_r^k \qquad \text{(Aggregate Model Update)} \qquad (4\text{b})$$

**Two key modifications** are introduced here: 1) $\boldsymbol{z}_{i,r}^k$ becomes $\boldsymbol{z}_r^k$; 2) Model aggregation is now computed using the model update (i.e., the difference observed during the local update step). The first modification is feasible if all clients agree on one common seed, and this modification paves the way for grouping the $g_{i,r}^k$ in the second modification. The second modification is crucial to save communication because only $\frac{1}{|C_r|} \sum_{i \in C_r} g_{i,r}^k$ is unknown to the server, which requires the transmission, but this quantity is merely a scalar!

With this seemingly minor adjustment, we significantly reduce the second $d$-length vector within the uplink, transforming it into a small constant quantity. However, this improvement is insufficient to achieve dimension-free communication due to the inherent requirements of the pull-model step.

### 3.3 ELIMINATING DIMENSION-DEPENDENT COMMUNICATION IN THE DOWNLINK

The challenge remains to eliminate the full model transfer that occurs during the pull-model step in (2a). The solution is similar to the modification of model update in (4b), albeit with greater subtlety. **Presuming** that the client model $\boldsymbol{x}_{i,r'}^K$ is the same as the server model $\boldsymbol{x}_{r'}$, where $r'$ is the last participated round, then the process of pulling the model from the server at the r-th round can be expressed as

$$\boldsymbol{x}_{i,r}^1 = \boldsymbol{x}_{i,r'}^K - \eta \sum_{j=r'}^{r-1} \sum_{k=1}^{K} g_j^k \cdot \boldsymbol{z}_j^k, \qquad \text{(Reconstruct Model)} \qquad (5)$$

where $g_j^k = \frac{1}{|C_r|} \sum_{i \in C_r} g_{i,j}^k$ is the average gradient scalar. A crucial observation is that, at the end of the local update, the client model $\boldsymbol{x}_{i,r'}^K$ deviates from the server model in equation (4b). This discrepancy poses a problem because our approach relies on communicating the gradient via scalar values instead of directly transmitting the updated model. One straightforward solution is to **take a snapshot of the client model at the beginning of the local update and revert to it** after the local update is completed. This ensures consistency because, as implied by equation (5), the client model $\boldsymbol{x}_{i,r}^1$ at the beginning of the local update is identical to the server model $\boldsymbol{x}_r$.

The data communicated between the server and clients in (5) is reduced to just a few gradient scalars and random seeds, achieving dimension-free again! We refer to this step as "Reconstruct Model" rather than "Pull Model" because it builds the model based on the local model instead of the server model. In fact, due to this, we do not even need the server model $\boldsymbol{x}_r$ to be stored on the server.

### 3.4 DeComFL Algorithm

---

**Algorithm 1** Dimension-Free Communication in Federated Learning (DeComFL) [Server-side]

---

1: **Initialize**: $\{g_0^k\}_{k=1}^K$, learning rate $\eta$, local update steps $K$, communication rounds $R$.
2: **Allocate**: memory for recording three states: 1) state set $\{t_i\}_{i=1}^N$ storing the last round that client $i$ participated in, 2) seed set $\{\{s_r^k\}_{k=1}^K\}_{r=0}^{R-1}$, 3) gradient set$\{\{g_r^k\}_{k=1}^K\}_{r=0}^{R-1}$.
3:
4: **for** $r = 0, 1, \ldots, R-1$ **do**
5:      Uniformly sample a client set $C_r$ with cardinality $m$ and sample $K$ seeds $\{s_r^k\}_{k=1}^K$
6:      **for** each client $i \in C_r$ **in parallel do**
7:          **ClientRebuildModel**$(\{\{g_{r'}^k\}_{k=1}^K\}_{r'=t_i}^{r-1}, \{\{s_{r'}^k\}_{k=1}^K\}_{r'=t_i}^{r-1})$        $\triangleright$ Send $g$ and $s$ to client
8:          $\{g_{i,r}^k\}_{k=1}^K = $ **ClientZOLocalUpdate**$(\{s_r^k\}_{k=1}^K, r)$        $\triangleright$ Send $s$ to client and receive $g$
9:      **end for**
10:      Compute the global gradient scalars $\{g_r^k\}_{k=1}^K = \left\{\frac{1}{|C_r|} \sum_{i \in C_r} g_{i,r}^k\right\}_{k=1}^K$
11:      Store $\{g_r^k\}_{k=1}^K$ and $\{s_r^k\}_{k=1}^K$ and update the client's last update record $t_i = r$
12:      $\boldsymbol{x}_{r+1} = \boldsymbol{x}_r - \eta \sum_{k=1}^K \cdot g_r^k \cdot \boldsymbol{z}_r^k$                $\triangleright$ This step is optional.
13: **end for**

---

---

**Algorithm 2** Dimension-Free Communication in Federated Learning (DeComFL) [Client-side]

---

1: **Initialize**: maintain a local model $\boldsymbol{x}_{i,0}^1$ and standby until the following procedures triggered by server.
2: **procedure 1. ClientRebuildModel**$(\{\{g_{r'}^k\}_{k=1}^K\}_{r'=t_i}^{r-1}, \{\{s_{r'}^k\}_{k=1}^K\}_{r'=t_i}^{r-1})$
3:      **for** $r' = t_i, \ldots, r-1$ **do**                      $\triangleright$ Equivalent to Pull-model step.
4:          **for** $k = 1, \ldots, K$ **do**
5:              Generate $\boldsymbol{z}_{r'}^k \sim \mathcal{N}(\boldsymbol{0}, \mathbf{I}_d)$ by random seed $s_{r'}^k$.
6:              $\boldsymbol{x}_{i,r'}^{k+1} = \boldsymbol{x}_{i,r'}^k - \eta g_{r'}^k \cdot \boldsymbol{z}_{r'}^k$              $\triangleright$ $\boldsymbol{x}_{i,t_i}^0$ is the local model.
7:          **end for**
8:      **end for**
9: **end procedure**
10:
11: **procedure 2. ClientZOLocalUpdate**$(\{s_r^k\}_{k=1}^K, r)$        $\triangleright$ Can be replaced by other ZO methods.
12:      **for** $k = 1, \ldots, K$ **do**
13:          Generate $\boldsymbol{z}_r^k \sim \mathcal{N}(\boldsymbol{0}, \mathbf{I}_d)$ by random seed $s_r^k$
14:          $g_{i,r}^k = \frac{1}{\mu}\big(f_i(\boldsymbol{x}_{i,r}^k + \mu \boldsymbol{z}_r^k; \xi_{i,r}^k) - f_i(\boldsymbol{x}_{i,r}^k; \xi_{i,r}^k)\big)$        $\triangleright$ Forward difference style
15:          $\boldsymbol{x}_{i,r}^{k+1} = \boldsymbol{x}_{i,r}^k - \eta g_{i,r}^k \cdot \boldsymbol{z}_r^k$                 $\triangleright$ Standard ZO-SGD
16:      **end for**
17:      **revert** the local model back to $x_{i,r}^1$.                $\triangleright$ **This step is crucial.**
18:      **Return** $\{g_{i,r}^k\}_{k=1}^K$
19: **end procedure**

---

With the mathematical groundwork established, we are now prepared to present the DeComFL algorithm. A comprehensive description is provided in Algorithm 1 (from the server's perspective) and Algorithm 2 (from the client's perspective), with a high-level illustration depicted in Fig. 1.

As shown in the Algorithm tables, DeComFL deviates significantly from the traditional FL framework. We transform the standard three-step process (pulling, local update, and aggregation) as a new three-

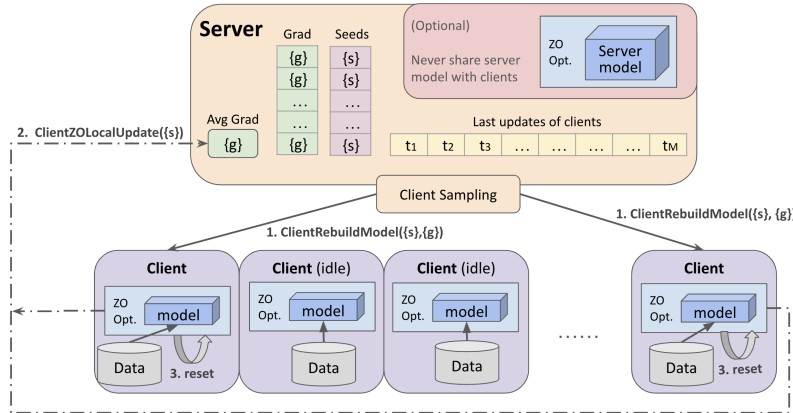

Figure 1: Illustration of DeComFL and Components Used in the Server and Clients.

step approach: reconstructing, local update with revert, and global aggregate of gradient scalars. This revised framework necessitates several additional details to ensure the implementation of the algorithm in practice. To highlight a few:

a) **Allocation** (Line 2 in Alg. 1): The server is required to maintain some states to keep track of the client's last participation round, gradient scalar history and random seeds.

b) **Seed Generation:** Server samples $K$ integers from a uniform distribution and then sends them to the clients for the base seeds to generate random vectors.

c) **ClientRebuildModel**: (Lines 2-9 in Alg. 2) Assume that the current round is $r$-th round. Before executing the local update procedure, sampled clients need to reconstruct their own model because they may not participate in training in the $(r-1)$-th round. Hence, the clients need to fill the gap in the model update between the current round and the last round where they participate in the training. It is corresponding to equation (5).

d) **ClientZOLocalUpdate**: (Lines 11-19 in Alg. 2) After each sampled client finishes rebuilding its model, local updates begin. Specifically, the client uses shared random seeds sent by the server to generate perturbations for each local update. Each perturbation is used for one corresponding local update. Then, they execute local update steps for $K$ times to train their own local models by ZOO algorithms (e.g., ZO-SGD). Finally, they revert the model as discussed in Sec. 3.3 and send scalars $\{g_{i,r}^k\}_{k=1}^K$ back to the server. It is corresponding to equation (4a).

We emphasize that all information transmitted between the client and server consists of a few scalars and random seeds, representing "Dimension-Free" for DeComFL. For clarity, we only present the simplest case of DeComFL in the main paper. For instance, the presented algorithm uses a single perturbation vector. Extending it to incorporate multiple perturbation vectors is straightforward, which is listed in Appendix B.4. *All subsequent theorems and experiments are based on this multi-perturbation version.* Moreover, as an algorithmic framework, DeComFL can be readily improved and extended into several variants. We defer this to Appendix B.

## 4 CONVERGENCE AND COMMUNICATION COST ANALYSIS

### 4.1 CONVERGENCE ANALYSIS

This section presents the convergence rate of DeComFL under standard assumptions and the additional low effective rank assumption. Due to limited space, all proofs are deferred to Appendix D. First, we list the following standard assumptions 1, 2, 3, which are commonly used in the existing literature.

**Assumption 1 (Unbiased Stochastic Gradient with Bounded Variance)** *For any $r \geq 1$, we have*

$$\mathbb{E}\left[\nabla f_i(\boldsymbol{x}_r; \xi_r)\right] = \nabla f_i(\boldsymbol{x}_r) \text{ and } \mathbb{E}\left[\|\nabla f_i(\boldsymbol{x}_r; \xi_r) - \nabla f_i(\boldsymbol{x}_r)\|^2\right] \leq \sigma^2, \ \forall i.$$

**Assumption 2 (Bounded Gradient Dissimilarity)** *For any $i \in [M]$, $\|\nabla f(\boldsymbol{x}) - \nabla f_i(\boldsymbol{x})\|^2 \leq \sigma_G^2$.*

**Assumption 3 ($L$-Lipschitz Continuous Gradient)** *$f \in \mathcal{C}_L^{1,1}(\mathbb{R}^d)$, i.e., $f$ is continuous and differentiable in first order and satisfies $L$-smooth condition:*

$$\|\nabla f(\boldsymbol{x}) - \nabla f(\boldsymbol{y})\| \leq L\|\boldsymbol{x} - \boldsymbol{y}\|, \ \forall \boldsymbol{x}, \boldsymbol{y} \in \mathbb{R}^d.$$

The convergence bound of DeComFL under the above standard assumptions is as follows:

**Theorem 1 (Standard Convergence Bound of DeComFL)** *Under Assumptions 1, 2 and 3, using Gaussian perturbations $\boldsymbol{z}_r^k \sim \mathcal{N}(\boldsymbol{0}, \boldsymbol{I}_d)$, and if $\eta \leq \min\{\frac{mP}{24L(d+4)}, \frac{2P}{mKL(d+P+4)}, \frac{1}{mK^2L}, \frac{mP(d+3)^3}{2L[3mPK(d+3)^3+(d+6)^3]}\}$, the sequence of iterates generated by DeComFL satisfies:*

$$\frac{1}{R}\sum_{r=0}^{R-1}\mathbb{E}_r\|\nabla f(\boldsymbol{x}_r)\|^2 \leq \frac{4D}{KR\eta} + \left(\frac{72KL\eta}{m} + \frac{24(d+4)L\eta}{mP}\right)\sigma_G^2 + \frac{16L(d+4)\eta}{mP}\sigma^2 + 2\mu^2L^2(d+3)^3,$$

*where $D = f(\boldsymbol{x}_0) - f(\boldsymbol{x}^\star)$, $\boldsymbol{x}_r$ is the model parameter in the $r$-th round, $P$ is the number of perturbations, $f(\boldsymbol{x}^\star)$ is the optimal loss value, $K$ is the number of local update steps, $R$ is the number of communication rounds, $d$ is the dimension of model parameters, and $m$ is the number of sampled clients in each round.* ∎

**Remark 1** The right side of the bound in Theorem 1 comprises terms with distinct interpretations. The first term represents the decrease in the loss value at the initial point, the second term quantifies the impact of data heterogeneity, the third term arises from the stochastic gradient estimate, and the finite difference approximation introduces the fourth that is often negligible since $\mu$ is typically very small. The crucial terms are the middle two, depending on the dimension of the model parameters.

**Corollary 1 (Standard Convergence Rate of DeComFL)** *Further, based on Theorem 1, supposing that $\mu \leq \frac{1}{(d+3)\sqrt{PKR}}$ and $\eta = \mathcal{O}\left(\frac{\sqrt{mP}}{\sqrt{dRK}}\right)$, the convergence rate of DeComFL is $\mathcal{O}\left(\frac{\sqrt{d}}{\sqrt{mPKR}}\right)$ when the algorithm runs with sufficient large communication round $R$.* ∎

**Remark 2** Both the number of local updates, $K$, and the number of perturbations, $P$, appear in the denominator of the final convergence rate, indicating a linear speedup with increasing client numbers and local steps. However, these parameters have opposing effects on the learning rate. A larger number of perturbations allows for a larger learning rate, while a larger number of local updates necessitates a smaller learning rate. This is intuitive, as more perturbations reduce variance between clients, while more local updates increase the dissimilarity between client models.

The above corollary shows that the convergence rate of the ZO method, unfortunately, depends on the model dimension. However, recent research has shown that many deep learning models with ZO optimizers exhibit a faster training/fine-tuning process than the pessimistic dimension-dependent rate, as we established in the previous theorem. One convincing reason is that several prior studies have demonstrated that the Hessian of the loss function for deep neural networks trained using stochastic gradient descent exhibits a remarkably low effective rank (Papyan, 2020; Yao et al., 2020; Wu et al., 2020). Inspired by this observation, a tighter convergence bound related to the low effective rank can be established rather than one based on the model dimension. To achieve this, we adapt Assumption 1 regarding low effective rank from (Malladi et al., 2023) for the FL setting.

**Assumption 4 (Low $\kappa$-Effective Rank)** *Let $G(\boldsymbol{x}_r) = \max_i \max_{\xi_{i,r} \in \mathcal{D}} \|\nabla f_i(\boldsymbol{x}_r; \xi_{i,r})\|$. There exists a Hessian matrix $\mathbf{H}(\boldsymbol{x}_r) \preceq L \cdot \mathbf{I}_d$ such that:*

- *For all $\boldsymbol{x}$ such that $\|\boldsymbol{x} - \boldsymbol{x}_r\| \leq 2\eta dG(\boldsymbol{x}_r)$, we have $\nabla^2 f(\boldsymbol{x}) \preceq \mathbf{H}(\boldsymbol{x}_r)$.*

- *The effective rank of $\mathbf{H}(\boldsymbol{x}_r)$, i.e., $\frac{tr(\mathbf{H}(\boldsymbol{x}_r))}{\|\mathbf{H}(\boldsymbol{x}_r)\|_2}$, is at most $\kappa$.*

Based on this low effective rank assumption, we obtain the convergence bound that relies on the effective rank $\kappa$ only, that is, independent of the dimension of the model parameter $d$. *We restrict the theoretical analysis regarding the low effective rank assumption to the case where $K = 1$ only.* This does not significantly limit the applicability of our findings, as the communication cost per round in our algorithm scales linearly with $K$, whereas in typical FL algorithms, this cost is independent of $K$.

Thus, the communication savings achieved by local update techniques are less pronounced in our context. Further, we assume that $z_{i,r}$ is sampled from a sphere with radius $\sqrt{d}$ and the Gaussian case is listed in the Appendix D.5.

**Theorem 2 (Convergence of DeComFL with $\kappa$-Effective Rank)** *Under assumptions 1, 2, 3 and 4, supposing $\eta \leq \frac{1}{4L}\left(1 + \frac{\kappa d + d - 2}{P(d+2)}\right)^{-1}$ and drawing $z_i^r$ from unit ball with radius $\sqrt{d}$, it holds*

$$\frac{1}{R}\sum_{r=0}^{R-1}\mathbb{E}\|\nabla f(\boldsymbol{x}_r)\|^2 \leq \frac{4D}{R\eta} + \frac{2L\eta}{m}\left(1 + \frac{\kappa d + d - 2}{P(d+2)}\right)(\sigma_G^2 + \sigma^2) + \frac{1}{2}\mu^2 L^2(d+3)^3 + 4\mu^2 L^4 d^3\eta,$$

*where $D = f(\boldsymbol{x}_0) - f(\boldsymbol{x}^\star)$. Selecting $\eta = \mathcal{O}\left(\frac{\sqrt{mP}}{\sqrt{\kappa R}}\right)$ and $\mu \leq \frac{\sqrt[4]{\kappa}}{\sqrt[4]{mRP}\sqrt{(d+3)^3}}$, we can obtain*

$$\frac{1}{R}\sum_{r=0}^{R-1}\mathbb{E}\|\nabla f(\boldsymbol{x}_r)\|^2 = \mathcal{O}\left(\frac{\sqrt{\kappa}}{\sqrt{mRP}}\right) + \mathcal{O}\left(\left(\frac{\sqrt{P}}{\sqrt{m\kappa R}} + \frac{\sqrt{\kappa}}{\sqrt{mRP}}\right)(\sigma_G^2 + \sigma^2)\right). \quad (6)$$

*Further supposing $\kappa > P$, the convergence rate is $\mathcal{O}\left(\frac{\sqrt{\kappa}}{\sqrt{mRP}}\right)$ when round $R$ is sufficient large.* ∎

**To the best of our knowledge, we are the first to quantify the impact of the smoothness parameter $\mu$ in the context of low-effective rank analysis.** Previous proofs, such as the proof in (Malladi et al., 2023), adopt the unrealistic condition that $\mu \to 0$, resulting in a convergence rate devoid of any terms related to $\mu$. With our correction, the convergence rate established above is quite similar compared with Corollary 1, with the key difference being the replacement of the dimension $d$ with the effective rank $\kappa$. This distinction is crucial because $\kappa$ is typically far smaller than the model dimension $d$, particularly in LLMs where $d$ can be several orders of magnitude larger than $\kappa$. Unfortunately, like the Lipschitz constant, determining the exact value of $\kappa$ is challenging and computationally prohibitive. However, as we will demonstrate in the following experiment section, only a few thousand rounds are sufficient to train or fine-tune models with millions or even billions of parameters.

**Remark 3** Furthermore, we note that FedMeZO (Ling et al., 2024) claims to demonstrate the applicability of the low effective rank assumption to federated learning with multiple local updates. However, their proof contains a critical error: the sign of $T_1$ in equation (16) is incorrectly handled, leading to a reversal of the inequality's direction.

## 4.2 COMMUNICATION COST ANALYSIS

We compare the communication cost of DeComFL with three representative FL algorithms: FedAvg, FedZO and FedCom. Each baseline highlights a different aspect of DeComFL. FedAvg, the most classic FL algorithm, operates in the same multi-agent setup. FedZO, employing ZO-SGD as its optimizer, matches our ZOO techniques. FedCom, focusing on compression techniques, offers a comparison point for DeComFL's seed and gradient scalar compression strategy.

Table 1: Comparison of Total Communication Complexity of Typical Algorithms

| Algorithm | Uplink Comm. Per Round | Downlink Comm. Per Round | Round Complexity | Uplink Comm. Complexity | Downlink Comm. Complexity |
|---|---|---|---|---|---|
| FedAvg | $md$ | $md$ | $\mathcal{O}(\frac{1}{mK\epsilon^2})$ | $\mathcal{O}(\frac{d}{K\epsilon^2})$ | $\mathcal{O}(\frac{d}{K\epsilon^2})$ |
| FedZO | $md$ | $md$ | $\mathcal{O}(\frac{d}{mPK\epsilon^2})$ | $\mathcal{O}(\frac{d^2}{PK\epsilon^2})$ | $\mathcal{O}(\frac{d^2}{PK\epsilon^2})$ |
| FedCom | $\beta md$ | $md$ | $\mathcal{O}(\frac{1}{mK\epsilon^2})$ | $\mathcal{O}(\frac{\beta d}{K\epsilon^2})$ | $\mathcal{O}(\frac{d}{K\epsilon^2})$ |
| Ours (Standard) | $mKP$ | $2MKP$ | $\mathcal{O}(\frac{d}{mPK\epsilon^2})$ | $\mathcal{O}(\frac{d}{\epsilon^2})$ | $\mathcal{O}(\frac{Md}{m\epsilon^2})$ |
| Ours (Low rank) | $mP$ | $2MP$ | $\mathcal{O}(\frac{\kappa}{mP\epsilon^2})$ | $\mathcal{O}(\frac{\kappa}{\epsilon^2})$ | $\mathcal{O}(\frac{M\kappa}{m\epsilon^2})$ |

Having established the round complexity in the preceding theorems, we can determine the total communication cost to reach an $\epsilon$-approximate solution by calculating the per-round communication cost. Unlike typical FL algorithms, the number of scalars communicated per round in DeComFL is

non-deterministic, depending on the lagged history of each client. Specifically, for the "reconstruct model" step (downlink communication), the transmitted vector has a length of $2mKP \times \boldsymbol{h}$, where the factor of 2 accounts for both the seeds and gradient scalars, and $\boldsymbol{h}$ denotes the (potentially random) length of lagged rounds. However, we know that by the $R-$th round, a sampled client must have communicated $2RKP$ scalars with the server to reconstruct its model since the beginning. Therefore, on average, $2MKP$ scalars per round are communicated between the server and all $M$ clients. For the "local update" step (uplink communication), the returned vector length is always fixed at $mKP$, corresponding to $K$ local updates with $P$ gradient scalars per update for each of the $m$ clients.

We summarize the communication complexity in Table 1. Besides FedAvg, we compare it against FedCom (Haddadpour et al., 2021), which can be understood as FedAvg with a generic compression operation in the communication. In the table, we use $\beta \in (0, 1]$ to represent the compression ratio, and we further assume the order of the round complexity is not impacted. As we can clearly see in the table, the communication cost of all other algorithms has linear, even quadratic, dependency on the model dimension $d$. Under the low effective-rank scenario, if $\kappa \ll \frac{md}{MK}$, the theorem indicates that DeComFL can converge much faster than the first-order FL counterparts regarding the communication cost. Estimating the effective low rank $\kappa$ in the pure theoretical domain is hard, similar to the Lipschitz constant $L$. Based on the results of the numerical experiment shown in the next section, it should be safe to say that the effective low rank should be much smaller than the total dimensions of the model.

## 5 EXPERIMENTS

Our experiment results firstly echo our theoretical conclusion that using larger perturbation amount $P$ and local update steps $K$ can make DeComFL converge faster (refer to Table 1). More importantly, our experiment results clearly highlight that DeComFL achieves enormous savings in communication overhead in training and especially fine-tuning tasks in LLMs, compared to other baselines.

We begin by training a simple Convolutional Neural Network model from scratch on the MNIST image classification task (LeCun et al., 1998). In the training tasks, our FL system comprises 100 clients, and we partition the dataset into 100 subsets by Dirichlet distribution (i.e., $\alpha = 1$). Each subset is assigned to one sampled client. In each communication round, 10% of clients are randomly selected to participate in the training process.

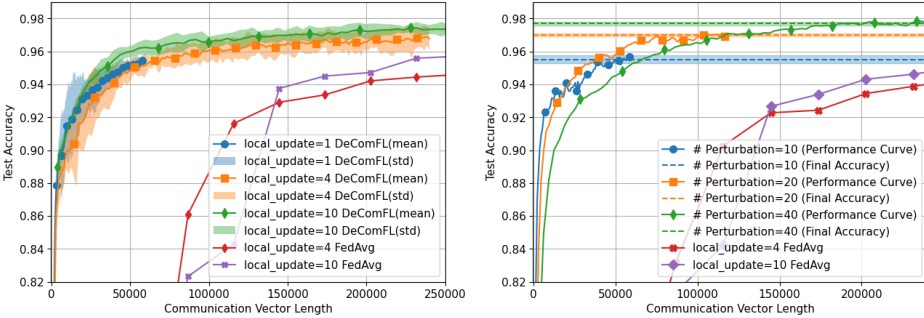

Figure 2: Ablation study of the influence of the number of local updates and perturbations on DeComFL. The communication vector length is the count of accumulated scalars transmitted between the server and a client.

**DeComFL largely reduces communication costs in training tasks.** Figure 2 is plotted based on the average communication vector length that is transferred between a client and the server. In the figure, we just compared DeComFL with the classical FedAvg as the corresponding first-order algorithm. All algorithms use SGD optimizer with momentum $0.5$. For DeComFL, we set the base learning rate as $0.001$. The figure clearly shows DeComFL's substantial communication cost savings, even for the small CNN model containing only 28,938 parameters. (Note: FedAvg has not converged yet).

**Increasing $P$ can enhance FL's performance.** Moreover, we can observe in Figure 2 that the larger number of perturbations eventually leads to higher test accuracy, which is more influential than local update, while slightly slowing down the algorithm at the beginning. Hence, we further evaluate this perturbation trick on Fashion (Xiao et al., 2017) with a larger CNN model (1,206,590 parameters). Like the learning rate scheduler, we use 25 perturbations at the beginning and double it at rounds 500, 1000, and 2000. Other settings are the same as MNIST. Besides FedAvg, we compare it against FedCom (Haddadpour et al., 2021) (using quantization) and FedAvg+Top $\mathbb{k}$ (uploading the largest $\mathbb{k}$

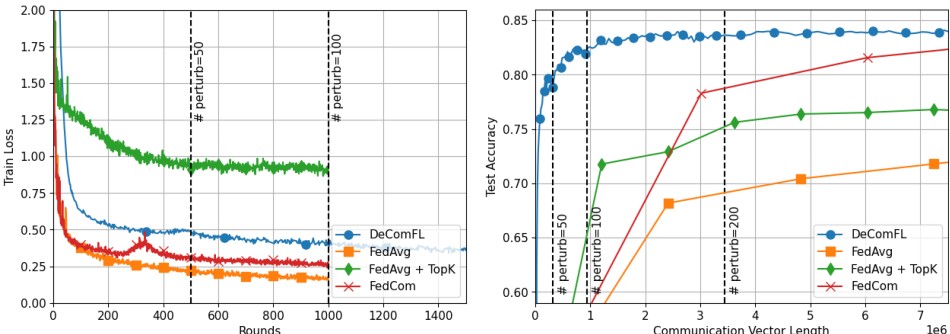

Figure 3: Comparison of Multiple FL Algorithms on the Fashion Dataset.

elements of the local update difference). We set $\Bbbk=10\%$ of parameters, and the quantization used in FedCom is compressing each element in the parameters to 8 bits. The left plot in Figure 3 shows loss against communication rounds. DeComFL converges slower than the first-order methods, except for Top $\Bbbk$, which is expected due to the inherent slower convergence of ZO methods. However, the slowness is not by the factor $d$ as the standard theorem established and supports the low effective-rank assumption. The right plot, illustrating test accuracy against effective communication vector length, reveals a different trend. DeComFL achieves higher performance with larger perturbations, while the communicated vector lengths are still significantly smaller than other algorithms.

**DeComFL can achieve tremendous communication savings in LLM fine-tuning tasks.** To further verify the DeComFL's effectiveness on LLMs, we execute fine-tuning tasks on a series of NLP datasets[1]. The models we used are OPT-125M and OPT-1.3B (Zhang et al., 2022). Due to the size of the model, we sample 2 clients from 8 clients to participate in each round to illustrate the core concept. In Table 2, we compare DeComFL with $P = 10$ against MeZO (single agent setting) and FedZO (multi-agent setting, $P = 10$) fine-tuning as baselines. All parameter settings and the definition of tasks are described in Sec. C.1 in the appendix. Although the number of rounds required for DeComFL convergence varies across different tasks (see appendix for details), the convergence consistently occurs within thousands of rounds, significantly fewer than the model's billions of dimensions and only slightly greater than that of first-order counterparts. This observation supports our low effective-rank assumption. The tables clearly demonstrate that DeComFL can match or even excel MeZO's performance. When using the same $P$, the performances of DeComFL and FedZO are almost the same, but the communication cost of DeComFL is dramatically lower than the one of FedZO. Lastly, the most important observation is that the communication costs for both model sizes are nearly identical, highlighting the dimension-free communication achieved by DeComFL.

Table 2: Test Accuracy and Communication Cost on Fine-Tuning Tasks

| Model | Dataset \ Task | FedAvg | MeZO | FedZO with $P = 10$ | DeComFL with $P = 10$ |
|---|---|---|---|---|---|
| OPT-125M | SST-2 | 87.32% (0.24 TB)[2] | 83.99% | 84.11% (0.75 TB) | 85.08% (0.36 MB) |
| | CB | 82.14% (0.12 TB) | 72.49% | 74.41% (0.38 TB) | 75.00% (0.12 MB) |
| | WSC | 63.25% (0.12 TB) | 55.18% | 59.47% (0.75 TB) | 59.59% (0.36 MB) |
| | WIC | 60.83% (0.12 TB) | 53.25% | 53.37% (0.75 TB) | 53.38% (0.36 MB) |
| | RTE | 63.96% (0.48 TB) | 52.91% | 54.16% (0.50 TB) | 57.05% (0.24 MB) |
| | BoolQ | 62.34% (0.24 TB) | 61.46% | 61.25% (0.50 TB) | 61.60% (0.24 MB) |
| OPT-1.3B | SST-2 | 90.38% (1.27 TB) | 90.23% | 90.33% (5.20 TB) | 90.78% (0.24 MB) |
| | CB | 83.93% (1.27 TB) | 74.01% | 74.49% (7.80 TB) | 75.71% (0.36 MB) |
| | WSC | 65.65% (1.27 TB) | 58.21% | 61.11% (7.80 TB) | 64.16% (0.36 MB) |
| | WIC | 65.82% (1.27 TB) | 55.95% | 56.08% (5.20 TB) | 56.14% (0.24 MB) |
| | RTE | 66.13% (2.54 TB) | 57.57% | 59.21% (3.90 TB) | 60.89% (1.80 MB) |
| | BoolQ | 63.83% (5.08 TB) | 61.98% | 62.14% (3.90 TB) | 62.50% (1.80 MB) |

---

[1]Loading and splitting datasets are based on `https://huggingface.co/datasets/super_glue`.
[2]The value enclosed in parentheses represents the total bytes of the vector transferred between the server and a single client throughout the entire fine-tuning phase. 1 TB $\approx$ 1,000,000 MB

ACKNOWLEDGMENTS

We sincerely thank Ms. Yidan(Julie) Huang from Google and Mr. Ziqi Zhou from RIT for their help and support in this work. This work has been partially supported by AI Seed Funding and the GWBC Award at RIT.

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

## A    CONCLUSION AND FUTURE WORK

We present DeComFL significantly reducing communication costs in FL, opening a new direction for combining FL with the zeroth-order optimization method. Our algorithm requires transmitting only a few scalar values, independent of model dimension, in both uplink and downlink communication. Moreover, we rigorously prove that under a low-rank assumption, DeComFL achieves convergence rates and total communication costs independent of model dimension — a first for theoretical results in federated learning. Empirical evaluations further demonstrate the communication efficiency of DeComFL in both training and fine-tuning tasks.

A limitation of DeComFL lies in its computational cost. Although zeroth-order optimization only requires forward passes, the overall computation remains high due to slower convergence. Additionally, this work does not address data heterogeneity issues, a challenge tackled by algorithms like Scaffold (Karimireddy et al., 2020), which remains an area for future exploration. Similarly, model pruning (Chen et al., 2023; Liu et al., 2018c) and integration with other optimization algorithms (Liu et al., 2020) may further reduce computational costs in training tasks.

Table 3: Notations in This Paper

| Notation | Meaning |
|---|---|
| $d$ | Total model parameter dimension |
| $m$ | Number of clients participating in each round |
| $i, M$ | Index, total number of clients |
| $r, R$ | Index, number of communication round |
| $p, P$ | Index, number of perturbations |
| $k, K$ | Index, number of local update iterations |
| $\boldsymbol{x}_r$ | Global model parameters in the $r$-th round |
| $\boldsymbol{x}_{i,r}^k$ | Local model parameters in the $k$-th iteration and $r$-th round at client $i$ |
| $\xi_{i,r}^k$ | Data sample used in the $k$-th iteration and $r$-th round at client $i$ |
| $g_{i,r}^k$ | Zeroth-order gradient estimate scalar |
| $\boldsymbol{z}_r^k$ | Perturbation in the $k$-th iteration and $r$-round |
| $f$ | Global loss function |
| $f_i$ | Local loss function at client $i$ |
| $C_r$ | Set of clients participating in $r$-th round |

## B    IMPROVEMENTS AND VARIATIONS OF ALGORITHM IMPLEMENTATION

DeComFL discussed in the main paper is a fairly general framework. In practice, several directions exist to extend and improve the Algorithm 1.

### B.1    IMPROVE THE PERFORMANCE

One well-known issue with ZO methods is that the variance of stochastic gradient introduced by the random perturbation is so significant that it takes longer to converge and requires a much smaller learning rate than the FO counterparts.

**Multiple Perturbations.** One common variation in zeroth-order optimization is to use multiple perturbations. Suppose $\boldsymbol{z}_{r,p}^k$ is a Gaussian random vector generated for $r$-th round, $k$-th local update step and $p$-th perturbation (i.i.d. for any $r, k, p$). One step of SGD local update becomes:

$$\boldsymbol{x}_{i,r}^{k+1} = \boldsymbol{x}_{i,r}^k - \eta \sum_{p=1}^{P} g_{i,r,p}^k \cdot \boldsymbol{z}_{r,p}^k, \quad \text{where } g_{i,r,p}^k = \frac{1}{\mu} \big( f_i(\boldsymbol{x}_{i,r}^k + \mu \boldsymbol{z}_{r,p}^k; \xi_{i,r}^k) - f_i(\boldsymbol{x}_{i,r}^k; \xi_{i,r}^k) \big) \quad (7)$$

That is we perturb the model $P$ times and calculate $P$ gradient scalars $\{g_{i,r,p}^k\}$. The effective update then becomes a weighted average of these perturbations. While using multiple perturbations increases the communication cost linearly with $P$ since all gradient scalars must be transmitted to the server for global aggregation, it can significantly reduce the variance of the stochastic gradient estimate

and then decrease the required rounds. Also, unlike large mini-batch size, this *does not* increase the memory consumption since we calculate the corresponding gradient scalar $\{g_{i,r,p}^k\}_{p=1}^P$ sequentially.

**Advanced Optimizers.** We can use more advanced optimizers to improve the performance or accelerate the algorithm's convergence. For clarity, Algorithm 1 is based on vanilla ZO-SGD, but it should be straightforward to extend to other zeroth-order optimization methods, such as ZO-SignSGD (Liu et al., 2018a), ZO-SVRG (Liu et al., 2018b), ZO-Adam (Chen et al., 2019), etc. A complete discussion and analysis of these optimizers are beyond the scope of this paper. However, to illustrate the necessary modifications, we present momentum SGD as an example:

$$\boldsymbol{m}_{i,r}^{k+1} = \beta \boldsymbol{m}_{i,r}^{k+1} + (1-\beta) g_{i,r}^k \cdot \boldsymbol{z}_r^k \tag{8}$$

$$\boldsymbol{x}_{i,r}^{k+1} = \boldsymbol{x}_{i,r}^k - \eta \cdot \boldsymbol{m}_{i,r}^{k+1} \tag{9}$$

where $\beta$ is a momentum factor between 0 and 1. One pitfall is that the optimizer is no longer stateless like ZO-SGD. Hence, after the local update step, we have to revert both the model parameter $\boldsymbol{x}_{i,r}^K$ and the optimizer state $\boldsymbol{m}_{i,r}^K$. In the model reconstruction step, we maintain the optimizer states and model update at the same time as well.

## B.2 Decrease the Memory Consumption

**Reduce the Memory on the Server Side.** A straightforward improvement is that the server no longer stores the model if there is no such necessity. This allows the server to function as a simple aggregator of scalar values. In a cloud service environment, this translates to utilizing less expensive CPU-only instances instead of costly GPU instances, resulting in substantial cost savings.

In our base algorithm (Algorithm 1), the server stores the entire history of selected random seeds and computed gradient scalars, potentially leading to significant memory consumption over many rounds. However, we can optimize this by recognizing that only information from the most recent round of each client is necessary. Since the server tracks each client's last updated round, a queue can efficiently manage this history. After clients complete their local updates, the server discards outdated information, minimizing memory usage. Further memory optimization can be achieved by having clients track and send their last participation round to the server, eliminating the need for the server to store this information.

**Reduce the Memory on the Client Side.** One significant memory consumption on the client side is that we have to take a snapshot before the local update step. There is an alternative, more memory-efficient solution to ensuring the prerequisite for equation (5) is satisfied - namely, that the client model $\boldsymbol{x}_{i,r'}^K$ at the end of the local update is identical to the server model $\boldsymbol{x}_{r'}$. Subtracting (4b) from (4a)[3], then we get

$$\boldsymbol{x}_{r+1} - \boldsymbol{x}_{i,r}^K = \boldsymbol{x}_r - \boldsymbol{x}_{i,r}^1 - \eta \sum_{k=1}^K \left( g_r^k - g_{i,r}^k \right) \cdot \boldsymbol{z}_r^k \tag{10}$$

Since $\boldsymbol{x}_r = \boldsymbol{x}_{i,r}^1$ after the reconstructing step, the quantity $\sum_{k=1}^K \left( g_r^k - g_{i,r}^k \right) \cdot \boldsymbol{z}_r^k$ represents the divergence between the local client model and the global server model. By compensating for this divergence, we can synchronize the two models. Crucially, this quantity can be generated using only gradient scalars and random seeds, eliminating the need to store a snapshot of the entire model. However, this technique is specific to SGD optimization.

**Comparison with FedZO.** With the above memory reduction technique, our approach offers significant memory consumption advantages over FedZO, both on the server and client sides.
Server-Side: Our algorithm can achieve near-zero server memory consumption. By discarding the server-side model and only tracking the latest random seeds and computed gradient scalars, we minimize memory usage. In contrast, FedZO requires storing at least two copies of the model (the averaged model and the aggregated update), leading to a memory peak of at least $2d$, where $d$ is the model dimensionality.
Client-Side: Our algorithm requires only $d + \phi$ memory per client, where $d$ is for the client model and $\phi$ is the largest parameter size from the parameter-wise perturbation (Malladi et al., 2023). Since

---

[3](4a) is $K$ recursions. We expand the equation from $K$ to 1 before the subtraction.

$\phi$ is typically negligible compared to $d$, the client-side memory is essentially $d$. Conversely, FedZO with multiple perturbations necessitates storing the model and intermediate gradients, resulting in a peak memory consumption of $3d$ (as illustrated in equation (7)). We summarized the comparison in the following table.

Table 4: Memory Cost Comparison between FedZO and DeComFL.

| Memory Cost | Server | Client |
|---|---|---|
| FedZO | $2d$ | $3d$ |
| DeComFL | Negligible | $\approx d$ |

## B.3 OTHERS VARIANTS

**Model Pulling for Excessive Lagging If Necessary.** If a client remains unsampled for an extended period, the model-pulling step requires retrieving all historical seeds and gradient scalars to update the model. This can be computationally demanding. In contrast, directly pulling the server's model has a fixed cost regardless of the client's lag time. This introduces a trade-off: if a client has limited computational resources and can tolerate communicating full model parameters, it might be preferable to pull the server's model simply.

**Enhanced Privacy through Private Seed Shifting.** Data privacy is paramount in FL. While our proposed DeComFL, like other FL algorithms, does not share local raw data with the server, we can further enhance privacy protection by ensuring that the server remains unaware of how the model evolves on the client side. Notice that even without direct access to local data, the server could potentially infer some information about local data distribution by comparing model updates between rounds. To address this, we introduce a simple and effective improvement: a private shift or function known only to the clients applies to the random seeds. Upon receiving a seed to generate a perturbation, the client first applies this private shift function to alter the seed. Since this shift is deterministic, it is easy to see that this modification does not affect the functionality of our algorithm while it prevents the server from reconstructing the model updates (This shift can be established via another server or a consensus protocol among clients). As a result, the random gradient scalars transmitted to the server cannot convey any information about the local data distribution, further enhancing privacy protection.

## B.4 DECOMFL WITH $P > 1$

In Algorithm 1 and 2 in the main paper, we demonstrate DeComFL using one perturbation ($P = 1$). To show that our DeComFL supports multiple perturbations ($P > 1$), we establish Algorithm 3 and 4. The primary difference between the two cases is the way in which the model is updated.

---

**Algorithm 3** DeComFL ($P > 1$) [Server-side]

---

1: **Initialize**: $\{\{g_0^k\}_{p=1}^P\}_{k=1}^K$, learning rate $\eta$, local update steps $K$, communication rounds $R$, the number of Perturbations $P$.
2: **Allocate**: memory for recording three states: 1) state set $\{t_i\}_{i=1}^N$ storing the last round that client $i$ participated in, 2) seed set $\{\{\{s_{r,p}^k\}_{p=1}^P\}_{k=1}^K\}_{r=0}^{R-1}$, 3) gradient set$\{\{\{g_r^k\}_{p=1}^P\}_{k=1}^K\}_{r=0}^{R-1}$.
3:
4: **for** $r = 0, 1, \ldots, R-1$ **do**
5:     Uniformly sample a client set $C_r$ with cardinality $m$ and sample $P * K$ seeds $\{\{s_{r,p}^k\}_{p=1}^P\}_{k=1}^K$
6:     **for** each client $i \in C_r$ **in parallel do**
7:         **ClientRebuildModel**$(\{\{\{g_{r',p}^k\}_{p=1}^P\}_{k=1}^K\}_{r'=t_i}^{r-1}, \{\{\{s_{r',p}^k\}_{p=1}^P\}_{k=1}^K\}_{r'=t_i}^{r-1})$   ▷ Send $g, s$ to client
8:         $\{\{g_{i,r,p}^k\}_{p=1}^P\}_{k=1}^K = $ **ClientZOLocalUpdate**$(\{\{s_{r,p}^k\}_{p=1}^P\}_{k=1}^K, r)$ ▷ Send $s$ to client and receive $g$
9:     **end for**
10:     Compute the global gradient scalars $\{\{g_{r,p}^k\}_{p=1}^P\}_{k=1}^K = \left\{\left\{\frac{1}{|C_r|}\sum_{i \in C_r} g_{i,r,p}^k\right\}_{p=1}^P\right\}_{k=1}^K$
11:     Store $\{\{g_{r,p}^k\}_{p=1}^P\}_{k=1}^K$ and $\{\{s_{r,p}^k\}_{p=1}^P\}_{k=1}^K$ and update the client's last update record $t_i = r$.
12:     $\boldsymbol{x}_{r+1} = \boldsymbol{x}_r - \frac{\eta}{P}\sum_{k=1}^K \sum_{p=1}^P \cdot g_{r,p}^k \cdot \boldsymbol{z}_{r,p}^k$             ▷ This step is optional.
13: **end for**

---

---

**Algorithm 4** DeComFL ($P > 1$) [Client-side]

---

1: **Initialize**: maintain a local model $\boldsymbol{x}_{i,0}^1$ and standby until the following procedures triggered by server.
2: **procedure 1. ClientRebuildModel**($\{\{\{g_{r',p}^k\}_{p=1}^P\}_{k=1}^K\}_{r'=t_i}^{r-1}, \{\{\{s_{r'}^k\}_{p=1}^P\}_{k=1}^K\}_{r'=t_i}^{r-1}$)
3:     **for** $r' = t_i, \ldots, r-1$ **do**                                    ▷ Equivalent to Pull-model step.
4:         **for** $k = 1, \ldots, K$ **do**
5:             **for** $p = 1, \ldots, P$ **do**
6:                 Generate $\boldsymbol{z}_{r',p}^k \sim \mathcal{N}(\boldsymbol{0}, \mathbf{I}_d)$ by random seed $s_{r',p}^k$.         ▷ This can be on-the-fly style
7:             **end for**
8:             $\boldsymbol{x}_{i,r'}^{k+1} = \boldsymbol{x}_{i,r'}^k - \frac{\eta}{P} \sum_{p=1}^P g_{r',p}^k \cdot \boldsymbol{z}_{r',p}^k$                ▷ $\boldsymbol{x}_{i,t_i}^0$ is the local model.
9:         **end for**
10:     **end for**
11: **end procedure**
12:
13: **procedure 2. ClientZOLocalUpdate**($\{\{s_{r,p}^k\}_{p=1}^P\}_{k=1}^K, r$)         ▷ Can be replaced by other ZO methods.
14:     **for** $k = 1, \ldots, K$ **do**
15:         $\Delta = \boldsymbol{0}$
16:         **for** $p = 1, \ldots, P$ **do**
17:             Generate $\boldsymbol{z}_{r,p}^k \sim \mathcal{N}(\boldsymbol{0}, \mathbf{I}_d)$ by random seed $s_{r,p}^k$
18:             $g_{i,r,p}^k = \frac{1}{\mu}\left(f_i(\boldsymbol{x}_{i,r}^k + \mu\boldsymbol{z}_{r,p}^k; \xi_{i,r}^k) - f_i(\boldsymbol{x}_{i,r}^k; \xi_{i,r}^k)\right)$            ▷ Forward difference style
19:             $\Delta = \Delta + g_{i,r,p}^k \cdot \boldsymbol{z}_{r,p}^k$
20:         **end for**
21:         $\boldsymbol{x}_{i,r}^{k+1} = \boldsymbol{x}_{i,r}^k - \frac{\eta}{P}\Delta$
22:     **end for**
23:     **revert** the local model back to $x_{i,r}^1$.                              ▷ This step is crucial.
24:     **Return** $\{\{g_{i,r,p}^k\}_{p=1}^P\}_{k=1}^K$
25: **end procedure**

---

### B.5 EQUIPPING DECOMFL WITH GRADIENT PROJECTION

This subsection examines a related approach from (Yue et al., 2023) that utilizes the first-order gradient information but is closely related to the zeroth-order method. **Although their framework is not applicable to federated learning, we can adapt their projection technique within the DeComFL framework**. The main idea is making an inner production of the first-order gradient with a random direction vector $\boldsymbol{z}_r$, sampled from a standard Gaussian distribution:

$$\hat{\boldsymbol{g}}_{i,r} = \langle \boldsymbol{z}_r, \nabla f_i(\boldsymbol{x}_{i,r})\rangle, \quad \boldsymbol{z}_r \sim \mathcal{N}(\boldsymbol{0}, I_d) \tag{11}$$

This formula can be viewed as projecting the first-order gradient $\nabla f_i(\boldsymbol{x}_{i,r})$ into the subspace of $\text{Span}(\boldsymbol{z}_r)$ since

$$\text{Proj}_{\boldsymbol{z}_r}(\nabla f_i(\boldsymbol{x}_{i,r})) = \frac{\langle \nabla f_i(\boldsymbol{x}_{i,r}), \boldsymbol{z}_r\rangle}{\langle \boldsymbol{z}_r, \boldsymbol{z}_r\rangle}\boldsymbol{z}_r \approx \frac{1}{d}\langle \boldsymbol{z}_r, \nabla f_i(\boldsymbol{x}_{i,r})\rangle\boldsymbol{z}_r, \tag{12}$$

where the scalar $\frac{1}{d}$ can be absorbed into the learning rate.
Note that the dimension of $\hat{\boldsymbol{g}}_{i,r}$ is just 1 (i.e., we compress a $d$-dimensional gradient information into a scalar). Hence, it is not surprising that we can replace the ZOO gradient scalar computed in (3) by this scalar obtained by random projection and produce another valid algorithm. The server-side algorithm of gradient projection is exactly the same as Algorithm 1, and the client-side algorithm is listed in Algorithm 5.

It is straightforward to see that this algorithm shares the same communication cost per round as DeComFL with the ZOO approach. However, this approach requires significantly more computation and memory sources than the ZOO approach. Specifically, for each step, the extra computation required for gradient projection is one backward propagation and one projection step. In high-dimension models with billions of parameters, this extra computation cost is substantial and cannot be ignored. Further, this approach requires the extra storage of gradient, which is in the same order as the model size.

We conducted the same experiment as shown in Figure 2. In Figure 4, two algorithms enjoy the same communication cost as mentioned before, but DeComFL with ZOO consistently

---

**Algorithm 5** DeComFL with Gradient Projection [Client-side]

---

1: **Initialize**: maintain a local model $\boldsymbol{x}_{i,0}^1$ and standby until the following procedures triggered by server.
2: **procedure 1. ClientRebuildModel**($\{\{g_{r'}^k\}_{k=1}^K\}_{r'=t_i}^{r-1}, \{\{s_{r'}^k\}_{k=1}^K\}_{r'=t_i}^{r-1}$)
3:     **for** $r' = t_i, \ldots, r-1$ **do**                                   ▷ Equivalent to Pull-model step.
4:         **for** $k = 1, \ldots, K$ **do**
5:             Generate $\boldsymbol{z}_{r'}^k \sim \mathcal{N}(\boldsymbol{0}, \mathbf{I}_d)$ by random seed $s_{r'}^k$.
6:             $\boldsymbol{x}_{i,r'}^{k+1} = \boldsymbol{x}_{i,r'}^k - \eta g_{r'}^k \cdot \boldsymbol{z}_{r'}^k$                      ▷ $\boldsymbol{x}_{i,t_i}^0$ is the local model.
7:         **end for**
8:     **end for**
9: **end procedure**
10:
11: **procedure 2. ClientGradProjLocalUpdate**($\{s_r^k\}_{k=1}^K, r$)
12:     **for** $k = 1, \ldots, K$ **do**
13:         Generate $\boldsymbol{z}_r^k \sim \mathcal{N}(\boldsymbol{0}, \mathbf{I}_d)$ by random seed $s_r^k$
14:         Taking backward gradient computation $\nabla f_i(\boldsymbol{x}_{i,r}^K)$
15:         $g_{i,r}^k = \langle \nabla f_i(\boldsymbol{x}_{i,r}^K), \boldsymbol{z}_r^k \rangle$                   ▷ Calculate the projection
16:         $\boldsymbol{x}_{i,r}^{k+1} = \boldsymbol{x}_{i,r}^k - \eta g_{i,r}^k \cdot \boldsymbol{z}_r^k$             ▷ Gradient projection approach
17:     **end for**
18:     **revert** the local model back to $x_{i,r}^1$.            ▷ **This step is crucial.**
19:     **Return** $\{g_{i,r}^k\}_{k=1}^K$
20: **end procedure**

---

performs better than DeComFL with the gradient projection. Furthermore, we observe that DeComFL with ZOO exhibits insensitivity to the hyper-parameter learning rate, while DeComFL with gradient projection is unstable for different learning rates.

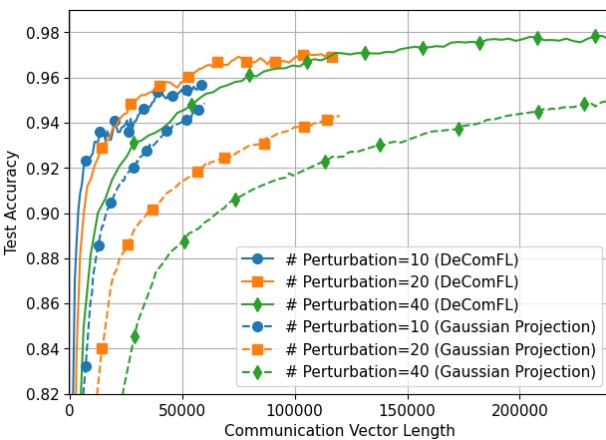

Figure 4: Comparison of DeComFL and the first-order method projected on random direction. The communication vector length is the count of accumulated scalars transmitted between the server and a client.

Here we provide two aspects to explain the preceding phenomenon. First, the zeroth-order method should not be simply reviewed as the projection of finite approximations of the true gradient. Instead, as we will state in Lemma 2 in Sec. D.2, the zeroth-order gradient is an unbiased estimate of the gradient of a smoothed function $f^\mu$:

$$f^\mu(\boldsymbol{x}) := \frac{1}{(2\pi)^{\frac{d}{2}}} \int f(\boldsymbol{x} + \mu\boldsymbol{z}) e^{-\frac{1}{2}\|\boldsymbol{z}\|^2} d\boldsymbol{z} = \mathbb{E}\left[f(\boldsymbol{x} + \mu\boldsymbol{z})\right],$$

The smoother function $f^\mu$ has a better function property and curvature to optimize, such as a better Lipschitz condition number (theorem 3.1 (a) of Ghadimi & Lan (2013)). As a result, it leads to improved convergence properties.

Second, $\langle \nabla f_i(\boldsymbol{x}_{i,r}), \boldsymbol{z}_r \rangle$ gives the directional derivative of $f(\boldsymbol{x})$ along $\boldsymbol{z}$. This is purely a first-order approximation of how $f(\boldsymbol{x})$ changes along $\boldsymbol{z}$. In contrast, $g_{ZO}$ uses function values at $x$ and $x + \mu z$ to estimate the rate of change in $f(x)$ along $z$, incorporating higher-order effects. This can be formally shown by Taylor's expansion:

$$\frac{1}{\mu}[f(x + \mu z) - f(x)] = \nabla f(x)^T z + \frac{\mu}{2} z^T \nabla^2 f(x) z + \cdots$$

Thus, the ZO gradient captures more information about the function's behavior along $\boldsymbol{z}$ than the gradient projection.

## C  ADDITIONAL EXPERIMENT DETAILS

### C.1  EXPERIMENT SETTINGS

**Datasets for LLM Fine-tuning Tasks.** We utilize a series of Natural Language Processing(NLP) datasets to execute fine-tuning tasks on LLMs (e.g., OPT-125M and OPT-1.3B), such as SST-2 (Socher et al., 2013; Wang et al., 2018) for the sentiment classification task, CB (De Marneffe et al., 2019) for hypothesis inference problem, WSC (Kocijan et al., 2020) for commonsense reasoning task, WIC (Pilehvar & Camacho-Collados, 2018) for word sense disambiguation task, RTE (Bowman et al., 2015) for natural language inference task, and BoolQ (Clark et al., 2019) for question answering.

**Hyper-parameter Settings.** In our FL system, for the experiments on LLMs, there are eight clients in total, and in each communication round, only two clients are sampled to participate in the training. In Table 5, we show the specific hyper-parameter settings about learning rate and total communication rounds. For other shared parameters, we set smooth parameter $\mu = 1e - 3$, Dirichlet concentration parameter $\alpha = 1$, and local update step $K = 1$. For DeComFL's experiments, we set $train\ batch\ size = 32$ and $test\ batch\ size = 64$.

Table 5: Experiment Settings of DeComFL

| Model+Fine Tuning | Parameter \ Dataset | SST-2 | CB | WSC | WIC | RTE | BoolQ |
|---|---|---|---|---|---|---|---|
| OPT-125M+FP[4] | Learning rate | 5e-6 | 2e-6 | 5e-6 | 2e-7 | 2e-6 | 2e-6 |
|  | Comm. rounds | 3k | 1k | 3k | 3k | 2k | 2k |
| OPT-1.3B+FP | Learning rate | 2e-6 | 5e-6 | 5e-6 | 2e-7 | 2e-6 | 2e-6 |
|  | Comm. rounds | 2k | 3k | 3k | 2k | 1.5k | 1.5k |
| OPT-125M+LoRA | Learning rate | 1e-4 | 1e-4 | 1e-4 | 1e-4 | 5e-5 | 1e-4 |
|  | Comm. rounds | 3k | 1k | 1k | 2k | 0.5k | 0.5k |

### C.2  ADDITIONAL EXPERIMENT RESULTS

**Combine DeComFL with other techniques.** Parameter-efficient fine-tuning (PEFT) techniques fine-tune just a fraction of the network parameters. As already discussed in (Malladi et al., 2023), the performance of zeroth-order optimization on LLMs does not improve substantially when tuning much fewer parameters. We confirm this surprising fact by combining LoRA (Hu et al., 2021) with DeComFL. In Table 6, the test accuracy can still be comparable to full-parameter MeZO or DeComFL. We set $alpha = 8$ and $rank = 8$ in LoRA and target all Q and V matrices in the attention layer of LLM. In addition to LoRA, DeComFL can be easily combined with other PEFT techniques, such as Prefix-tuning (Li & Liang, 2021) and LayerNorm-tuning (Zhao et al., 2023). Also, no matter what PEFT techniques we combine, the communication cost will stay the same if training rounds stay the same.

---

[4]FP means full-parameter fine-tuning in LLMs.

Table 6: Test Accuracy on Fine-Tuning Tasks (OPT-125M Model, **LoRA**)

| Dataset \ Task | MeZO | FedZO with $P = 5$ | DeComFL with $P = 5$ | DeComFL with $P = 10$ |
|---|---|---|---|---|
| SST-2 | 85.07% | 85.34% (0.66 TB) | 85.42% (0.18 MB) | 85.44% (0.36 MB) |
| CB | 69.64% | 70.55% (0.22 TB) | 71.07% (0.06 MB) | 71.43% (0.12 MB) |
| WSC | 52.66% | 54.61% (0.22 TB) | 54.53% (0.06 MB) | 57.03% (0.12 MB) |
| WIC | 53.49% | 53.12% (0.44 TB) | 53.08% (0.12 MB) | 53.71% (0.24 MB) |
| RTE | 50.15% | 50.92% (0.11 TB) | 51.40% (0.03 MB) | 51.40% (0.06 MB) |
| BoolQ | 60.68% | 60.53% (0.11 TB) | 60.12% (0.03 MB) | 60.78% (0.06 MB) |

## C.3 SUPPORTIVE EXPERIMENTS

### C.3.1 EVIDENCE FOR LOW-EFFECTIVE RANK ASSUMPTION

We start by validating the low-effective rank assumption on a small model since the memory requirement for low-effective rank validation on LLMs is quite expensive. We build a small custom ResNet (He et al., 2016) model trained with the CIFAR10 dataset, then plot the eigenvalue distribution of the Hessian after fixed training steps in the left one of Figure 5. The approach we used to estimate eigenvalue density is based on the algorithm in (Ghorbani et al., 2019), which leverages the stochastic Lanczos algorithm (Golub & Welsch, 1969). From the figure, we can clearly see that the majority of eigenvalues of the Hessian is $0$, which aligns with the low-effective rank assumption.

Although it is computationally prohibitive to validate the low effective rank assumption on LLMs since it requires the computation of a huge Hessian matrix, we can still provide indirect evidence as the right one of Figure 5 shown. We utilize the SST-2 dataset to train four LLMs with different scales, including OPT-125M, OPT-350M, OPT-1.3B, and OPT-2.7B. In Figure 5, we can observe that from the smallest OPT-125M model to the largest OPT-2.7B model, they all converge around the 1250-th round. The larger model converges slightly later, probably because it can achieve higher test accuracy. Hence, the $d$-dimensional dependent rate is a quite pessimistic estimation. This is a good implication that some low-effective rank holds in the modern large language model.

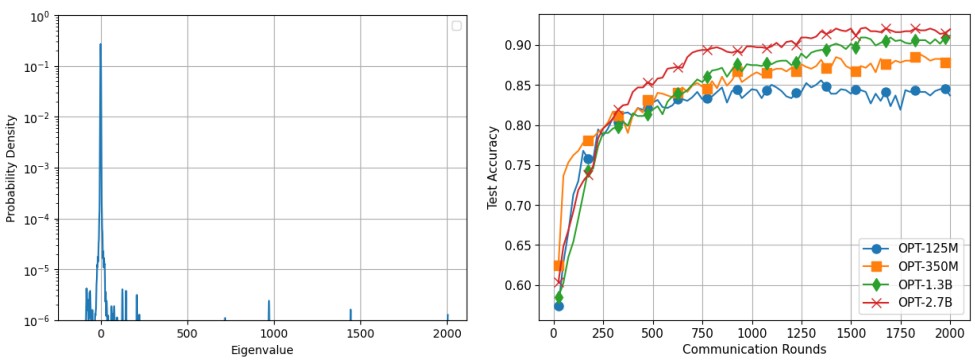

Figure 5: Hessian eigenvalue distribution with training custom ResNet on CIFAR10 dataset (Left); Comparison of convergence of DeComFL on multiple LLM scales on the SST-2 dataset (Right).

### C.3.2 COMMUNICATION COMPARISON OF DECOMFL AND FEDERATED LEARNING ALGORITHMS USING COMPRESSION

To show whether DeComFL can still provide substantial communication savings compared to FedAvg+Top$\Bbbk$ with a smaller $\Bbbk$, we use Fashion-MNIST dataset to test the FedAvg+Top$\Bbbk$'s performance with different $\Bbbk$, including $10\%$, $5\%$ and $1\%$ of the number of model parameters. Based on the experiment results in Figure 6, we can observe the following phenomenon: As a smaller $\Bbbk$, the communication cost of FedAvg+Top$\Bbbk$ can be reduced, but its communication overhead is still far higher than DeComFL. More seriously, this manner triggers severe performance degradation.

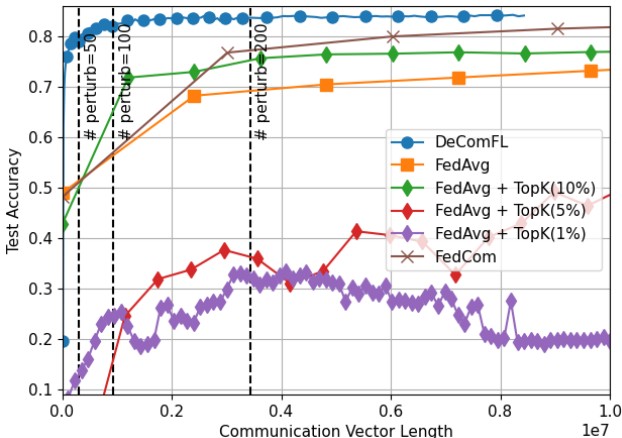

Figure 6: Comparison of DeComFL and related algorithms. The communication vector length is the count of accumulated scalars transmitted between the server and a client.

## D  THEORETICAL PROOF

### D.1  MAIN RECURSION

We can focus on the evolution of the server-side model only because after the revert of the model (or synchronize step), all sampled clients are the same as the server-side model. For other clients that are not sampled, they will sync to the server-side model after the reconstruction step, so that we can virtually assume that all clients and servers are iterated with the same server-side model.

The recursion of the server-side model can be written as

$$\boldsymbol{x}_{r+1} = \boldsymbol{x}_r - \eta \sum_{k=1}^{K} g_r^k \boldsymbol{z}_r^k = \boldsymbol{x}_r - \frac{\eta}{m} \sum_{k=1}^{K} \sum_{i \in C_r} g_{i,r}^k \boldsymbol{z}_r^k = \frac{1}{m} \sum_{i \in C_r} \Big( \underbrace{\boldsymbol{x}_{i,r} - \eta \sum_{k=1}^{K} g_{i,r}^k \boldsymbol{z}_r^k}_{:= \boldsymbol{x}_{i,r+1}} \Big).$$

where we just denote $\boldsymbol{x}_{i,r} = \boldsymbol{x}_r$ for the client's model. It is now clear that our algorithm follows the same routine as the Federated Average framework in that it combines the models after each client runs $K$ local-update steps in their local model $\boldsymbol{x}_{i,r+1}$.

### D.2  LEMMAS FOR THE ZEROTH-ORDER OPTIMIZATION

Before we present the proof of our main theorems, we first list several well-known lemmas about the zeroth-order optimization, which is the foundation for all the following proofs.

**Lemma 1** *(Nesterov, 2013) $f \in \mathcal{C}_L^{1,1}(\mathbb{R}^n)$ if it is differentiable and satisfies*

$$|f(\boldsymbol{y}) - f(\boldsymbol{x}) - \langle \nabla f(\boldsymbol{x}), \boldsymbol{y} - \boldsymbol{x} \rangle| \le \frac{L}{2} \|\boldsymbol{y} - \boldsymbol{x}\|^2. \tag{13}$$

**Lemma 2** *(Nesterov & Spokoiny, 2017; Ghadimi & Lan, 2013) We define a smooth approximation of objective function $f_i$ as $f_i^\mu(\cdot)$ that can be formulated as*

$$f_i^\mu(\boldsymbol{x}) := \frac{1}{(2\pi)^{\frac{d}{2}}} \int f_i(\boldsymbol{x} + \mu \boldsymbol{z}) e^{-\frac{1}{2}\|\boldsymbol{z}\|^2} d\boldsymbol{z} = \mathbb{E}\left[f_i(\boldsymbol{x} + \mu \boldsymbol{z})\right], \tag{14}$$

*where $\mu > 0$ is the smoothing parameter, and $\boldsymbol{z}$ is one n-dimensional standard Gaussian random vector. Then, for any $f_i \in \mathcal{C}_L^{1,1}$, the following statements hold.*

*(a) The gradient of $f_i^\mu(\cdot)$ is $L_\mu$-Lipschitz continuous where $L_\mu \le L$. $\nabla f_i^\mu(\boldsymbol{x})$ can be shown as*

$$\nabla f_i^\mu(\boldsymbol{x}) = \frac{1}{(2\pi)^{\frac{d}{2}}} \int \frac{f_i(\boldsymbol{x} + \mu \boldsymbol{z}) - f_i(\boldsymbol{x})}{\mu} \boldsymbol{z} e^{-\frac{1}{2}\|\boldsymbol{z}\|^2} d\boldsymbol{z}. \tag{15}$$

*(b) For any $\boldsymbol{x} \in \mathbb{R}^d$,*

$$|f_i^\mu(\boldsymbol{x}) - f_i(\boldsymbol{x})| \leq \frac{1}{2}\mu^2 L d \tag{16}$$

$$\|\nabla f_i^\mu(\boldsymbol{x}) - \nabla f_i(\boldsymbol{x})\| \leq \frac{1}{2}\mu L (d+3)^{\frac{3}{2}} \tag{17}$$

*(c) For any $\boldsymbol{x} \in \mathbb{R}^d$,*

$$\frac{1}{\mu^2}\mathbb{E}_{\boldsymbol{z}}\left[\left(f_i(\boldsymbol{x}+\mu\boldsymbol{z}) - f_i(\boldsymbol{x})\right)^2\|\boldsymbol{z}\|^2\right] \leq \frac{\mu^2}{2}L^2(d+6)^3 + 2(d+4)\|\nabla f_i(\boldsymbol{x})\|^2 \tag{18}$$

Following form (17) and utilizing Jensen's inequality $\|a\|^2 \leq 2\|a-b\|^2 + 2\|b\|^2$, we have

$$\|\nabla f_i^\mu(\boldsymbol{x})\|^2 \leq 2\|\nabla f_i(\boldsymbol{x})\|^2 + \frac{1}{2}\mu^2 L^2(d+3)^3, \tag{19}$$

$$\|\nabla f_i(\boldsymbol{x})\|^2 \leq 2\|\nabla f_i^\mu(\boldsymbol{x})\|^2 + \frac{1}{2}\mu^2 L^2(d+3)^3. \tag{20}$$

Moreover, we denote $f_i^\mu(\boldsymbol{x}^*) := \min_{\boldsymbol{x}\in\mathbb{R}^d} f_i^\mu(\boldsymbol{x})$ and conclude $|f_i^\mu(\boldsymbol{x}^*) - f_i(\boldsymbol{x}^*)| \leq \frac{\mu^2 L d}{2}$ from (16). Then, we further conclude that

$$-\mu^2 L d \leq [f_i^\mu(\boldsymbol{x}) - f_i^\mu(\boldsymbol{x}^*)] - [f_i(\boldsymbol{x}) - f_i(\boldsymbol{x}^*)] \leq \mu^2 L d. \tag{21}$$

### D.3 PROOF OF THEOREM 1

Our main theorem is based on multiple perturbations. To light the notation, we first introduce $G_{i,r}^k$ that stands for the stochastic zeroth-order gradient estimate on $\boldsymbol{x}_{i,r}^k$ averaging over $P$-perturbation directions:

$$G_{i,r}^k := \frac{1}{P}\sum_{p=1}^P G_{i,r,p}^k = \frac{1}{P}\sum_{p=1}^P \frac{f_i(\boldsymbol{x}_{i,r}^k + \mu\boldsymbol{z}_{r,p}^k; \xi_{i,r}^k) - f_i(\boldsymbol{x}_{i,r}^k; \xi_{i,r}^k)}{\mu}\boldsymbol{z}_{r,p}^k = \frac{1}{P}\sum_{p=1}^P g_{i,r,p}^k \cdot \boldsymbol{z}_{r,p}^k \tag{22}$$

To begin with, we start with a few lemmas about the property of $G_{i,r}^k$.

**Lemma 3 (Bounds on the Stochastic Zeroth-Order Gradient Variance)** *The variance of the stochastic zeroth-order gradient $\mathbb{E}\|G_{i,r}^k - \nabla f_i^\mu(\boldsymbol{x}_{i,r}^k)\|^2$ can be bounded by the true gradient $\|\nabla f(\boldsymbol{x}_r)\|^2$ on the starting point of round $r$, the local update distance $\|\boldsymbol{x}_{i,r}^k - \boldsymbol{x}_r\|^2$ and several constants:*

$$\mathbb{E}_r\left\|G_{i,r}^k - \nabla f_i^\mu(\boldsymbol{x}_{i,r}^k)\right\|^2 \leq \frac{6(d+4)}{P}\|\nabla f(\boldsymbol{x}_r)\|^2 + \frac{6L^2(d+4)}{P}\mathbb{E}_r\left\|\boldsymbol{x}_{i,r}^k - \boldsymbol{x}_r\right\|^2 + \frac{6(d+4)}{P}\sigma_G^2$$
$$+ \frac{2(d+4)}{P}\sigma^2 + \frac{\mu^2 L^2(d+6)^3}{2P}. \tag{23}$$

*Proof.* For any independent and identically distributed random variables $\{\boldsymbol{y}_p\}_{p=1}^P$ with the mean $\bar{y}$, we know

$$\mathbb{E}\left\|\frac{1}{P}\sum_{p=1}^P \boldsymbol{y}_p - \bar{y}\right\|^2 = \frac{1}{P^2}\sum_{p=1}^P \mathbb{E}\left\|\boldsymbol{y}_p - \bar{y}\right\|^2 \tag{24}$$

Recall that $G_{i,r}^k = \frac{1}{P}\sum_{p=1}^P G_{i,r,p}^k$, $\mathbb{E}\left[G_{i,r,p}^k|\boldsymbol{x}_{i,r}^k\right] = \nabla f_i^\mu(\boldsymbol{x}_{i,r}^k)$, and lemma 2 shows that

$$\mathbb{E}_r\left\|G_{i,r,p}^k - \nabla f_i^\mu(\boldsymbol{x}_{i,r}^k)\right\|^2 \leq 2(d+4)\left\|\nabla f_i(\boldsymbol{x}_{i,r}^k; \xi_{i,r}^k)\right\|^2 + \frac{\mu^2}{2}L^2(d+6)^3.$$

Substituting $G_{i,r}^k$ and above properties into (24), we establish

$$\mathbb{E}_r\left\|G_{i,r}^k - \nabla f_i^\mu(\boldsymbol{x}_{i,r}^k)\right\|^2 \leq \frac{2(d+4)}{P}\mathbb{E}_r\left\|\nabla f_i(\boldsymbol{x}_{i,r}^k; \xi_{i,r}^k)\right\|^2 + \frac{\mu^2 L^2(d+6)^3}{2P}$$

$$\leq \frac{2(d+4)}{P} \mathbb{E}_r \left\| \nabla f_i(\boldsymbol{x}_{i,r}^k) \right\|^2 + \frac{2(d+4)}{P} \sigma^2 + \frac{\mu^2 L^2 (d+6)^3}{2P}$$

Next, we bound the $\mathbb{E}_r \left\| \nabla f_i(\boldsymbol{x}_{i,r}^k) \right\|^2$ via the Jensen's inequality:

$$
\begin{aligned}
\mathbb{E}_r \left\| \nabla f_i(\boldsymbol{x}_{i,r}^k) \right\|^2 =& \mathbb{E}_r \left\| \nabla f_i(\boldsymbol{x}_{i,r}^k) - \nabla f_i(\boldsymbol{x}_r) + \nabla f_i(\boldsymbol{x}_r) - \nabla f(\boldsymbol{x}_r) + \nabla f(\boldsymbol{x}_r) \right\|^2 \\
\leq& 3\mathbb{E}_r \left\| \nabla f_i(\boldsymbol{x}_{i,r}^k) - \nabla f_i(\boldsymbol{x}_r) \right\|^2 + 3\mathbb{E}_r \| \nabla f_i(\boldsymbol{x}_r) - \nabla f(\boldsymbol{x}_r) \|^2 + 3\| \nabla f(\boldsymbol{x}_r) \|^2 \\
\leq& 3L^2 \mathbb{E}_r \left\| \boldsymbol{x}_{i,r}^k - \boldsymbol{x}_r \right\|^2 + 3\sigma_G^2 + 3\| \nabla f(\boldsymbol{x}_r) \|^2
\end{aligned}
$$

Lastly, plugging back, we finish the proof of lemma

$$
\begin{aligned}
\mathbb{E}_r \left\| G_{i,r}^k - \nabla f_i^\mu(\boldsymbol{x}_{i,r}^k) \right\|^2 \leq& \frac{2(d+4)}{P} \left( 3L^2 \mathbb{E}_r \left\| \boldsymbol{x}_{i,r}^k - \boldsymbol{x}_r \right\|^2 + 3\sigma_G^2 + 3\| \nabla f(\boldsymbol{x}_r) \|^2 \right) \\
& + \frac{2(d+4)}{P} \sigma^2 + \frac{\mu^2 L^2 (d+6)^3}{2P} \\
=& \frac{6(d+4)}{P} \| \nabla f(\boldsymbol{x}_r) \|^2 + \frac{6L^2(d+4)}{P} \mathbb{E}_r \left\| \boldsymbol{x}_{i,r}^k - \boldsymbol{x}_r \right\|^2 \\
& + \frac{6(d+4)}{P} \sigma_G^2 + \frac{2(d+4)}{P} \sigma^2 + \frac{\mu^2 L^2 (d+6)^3}{2P}
\end{aligned}
$$

∎

Similarly, we can also bound the second-order moments of $\mathbb{E}_r \| G_{i,r}^k \|^2$ as follows.

**Lemma 4 (Bounds on the Stochastic Zeroth-Order Gradient Second-Order Moments)**
$\mathbb{E} \| G_{i,r}^k \|^2$ can be bounded by the true gradient $\| \nabla f(\boldsymbol{x}_r) \|^2$ on the starting point of round $r$, the local update distance $\| \boldsymbol{x}_{i,r}^k - \boldsymbol{x}_r \|^2$ and several constants:

$$
\begin{aligned}
\mathbb{E}_r \left\| G_{i,r}^k \right\|^2 \leq& \frac{6(d+P+4)}{P} \| \nabla f(\boldsymbol{x}_r) \|^2 + \frac{6L^2(d+P+4)}{P} \mathbb{E}_r \left\| \boldsymbol{x}_{i,r}^k - \boldsymbol{x}_r \right\|^2 \\
& + \frac{6(d+P+4)}{P} \sigma_G^2 + \frac{2(d+4)}{P} \sigma^2 + \frac{\mu^2 L^2 (d+6)^3}{2P} + \frac{1}{2} \mu^2 L^2 (d+3)^3
\end{aligned}
\tag{25}
$$

*Proof.* Using Jensen's inequality, we know

$$
\mathbb{E}_r \left\| G_{i,r}^k \right\|^2 = \mathbb{E}_r \left\| G_{i,r}^k - \nabla f_i^\mu(\boldsymbol{x}_{i,r}^k) \right\|^2 + \mathbb{E}_r \left\| \nabla f_i^\mu(\boldsymbol{x}_{i,r}^k) \right\|^2
\tag{26}
$$

From Lemma 2, we have

$$
\begin{aligned}
& \mathbb{E}_r \left\| \nabla f_i^\mu(\boldsymbol{x}_{i,r}^k) \right\|^2 \\
\leq& 2\mathbb{E}_r \left\| \nabla f_i(\boldsymbol{x}_{i,r}^k) \right\|^2 + \frac{1}{2} \mu^2 L^2 (d+3)^3 \\
=& 2\mathbb{E}_r \left\| \nabla f_i(\boldsymbol{x}_{i,r}^k) - \nabla f_i(\boldsymbol{x}_r) + \nabla f_i(\boldsymbol{x}_r) - \nabla f(\boldsymbol{x}_r) + \nabla f(\boldsymbol{x}_r) \right\|^2 + \frac{1}{2} \mu^2 L^2 (d+3)^3 \\
\leq& 6L^2 \mathbb{E}_r \left\| \boldsymbol{x}_{i,r}^k - \boldsymbol{x}_r \right\|^2 + 6\sigma_G^2 + 6\| \nabla f(\boldsymbol{x}_r) \|^2 + \frac{1}{2} \mu^2 L^2 (d+3)^3
\end{aligned}
$$

Combining with the result (23), we conclude the proof of lemma. ∎

Furthermore, we denote $\chi_r = \mathbb{E}_r \left[ \frac{1}{M} \sum_{i=1}^M \sum_{k=1}^K \left\| \boldsymbol{x}_{i,r}^k - \boldsymbol{x}_r \right\|^2 \right]$ for the local update steps, which is closely related to $\mathbb{E}_r \left\| G_{i,r}^k \right\|^2$. Using the previous lemma, we can easily establish the upper bound on $\chi_r$.

**Lemma 5 (Bounds on Local Update Steps)** *With Assumptions 1-3 and the learning rate satisfying $\eta \leq \frac{2P}{\sqrt{6}LK\sqrt{d+P+4}}$, the local update distance $\chi_r$ satisfies*

$$
\begin{aligned}
\chi_r \leq& \frac{6K^3(d+P+4)\eta^2}{P} \| \nabla f(\boldsymbol{x}_r) \|^2 + \frac{6K^3(d+P+4)\eta^2}{2P} \sigma_G^2 + \frac{2K^3(d+4)\eta^2}{P} \sigma^2 \\
& + \frac{\mu^2 L^2 K^3 (d+6)^3 \eta^2}{P} + \frac{1}{2} \mu^2 L^2 K^3 (d+3)^3 \eta^2
\end{aligned}
$$

*Proof.* Utilizing the relationship $\boldsymbol{x}_{i,r}^k - \boldsymbol{x}_r = \eta \sum_{\tau=1}^k G_{i,r}^\tau$, we have

$$
\begin{aligned}
\chi_r =& \mathbb{E}_r \left[ \frac{\eta^2}{M} \sum_{i=1}^M \sum_{k=1}^K \left\| \sum_{\tau=1}^k G_{i,r}^\tau \right\|^2 \right] \\
\leq& \frac{\eta^2}{M} \sum_{i=1}^M \sum_{k=1}^K \sum_{\tau=1}^k k \mathbb{E}_r \left\| G_{i,r}^\tau \right\|^2 \\
\leq& \frac{K^2 \eta^2}{2M} \sum_{i=1}^M \sum_{k=1}^K \left\| G_{i,r}^k \right\|^2,
\end{aligned}
$$

where the last inequality holds since $\sum_{k=1}^K \sum_{\tau=1}^k k X_\tau = \sum_{\tau=1}^K (\sum_{k=\tau}^K k) X_\tau$. Substituting (25), we get

$$
\begin{aligned}
\chi_r \leq & \frac{K^2 \eta^2}{2M} \sum_{i=1}^M \sum_{k=1}^K \left( \frac{6(d+P+4)}{P} \|\nabla f(\boldsymbol{x}_r)\|^2 + \frac{6L^2(d+P+4)}{P} \mathbb{E} \left\| \boldsymbol{x}_{i,r}^k - \boldsymbol{x}_r \right\|^2 \right. \\
& \left. + \frac{6(d+P+4)}{P} \sigma_G^2 + \frac{2(d+4)}{P} \sigma^2 + \frac{\mu^2 L^2 (d+6)^3}{2P} + \frac{1}{2} \mu^2 L^2 (d+3)^3 \right)
\end{aligned} \tag{27}
$$

Moving the term $\mathbb{E} \left\| \boldsymbol{x}_{i,r}^k - \boldsymbol{x}_r \right\|^2$, which is $\chi_r$ again after the double summations, to the left-hand side, we have

$$
\begin{aligned}
\left( 1 - \frac{6L^2 K^2 (d+P+4)\eta^2}{2P} \right) \chi_r \leq & \frac{3K^3(d+P+4)\eta^2}{P} \|\nabla f(\boldsymbol{x}_r)\|^2 + \frac{3K^3(d+P+4)\eta^2}{2P} \sigma_G^2 \\
& + \frac{K^3(d+4)\eta^2}{P} \sigma^2 + \frac{\mu^2 L^2 K^3 (d+6)^3 \eta^2}{2P} \\
& + \frac{1}{4} \mu^2 L^2 K^3 (d+3)^3 \eta^2
\end{aligned} \tag{28}
$$

When $\eta \leq \frac{2P}{\sqrt{6}LK\sqrt{d+P+4}}$, the coefficient on the l.h.s. is larger than $\frac{1}{2}$. Plugging back, we complete the proof of this lemma. ∎

Now, we are ready to present the proof of the main theorem with the above lemmas. To ease the reference, we restate the theorem here again:

**Theorem 3 (Restated; Standard Convergence Bound of DeComFL)** *Under the assumptions 1, 2 and 3, supposing that the perturbation $\boldsymbol{z}_r^k \sim \mathcal{N}(\boldsymbol{0}, \boldsymbol{I}_d)$, i.e., follows the Gaussian distribution, and the learning rate satisfies $\eta \leq \min\{ \frac{mP}{24L(d+4)}, \frac{2P}{mKL(d+P+4)}, \frac{1}{mK^2L}, \frac{mP(d+3)^3}{2L[3mPK(d+3)^3 + (d+6)^3]} \}$, then it holds*

$$
\begin{aligned}
\frac{1}{R} \sum_{r=0}^{R-1} \mathbb{E}_r \|\nabla f(\boldsymbol{x}_r)\|^2 \leq & \frac{4\big(f(\boldsymbol{x}_0) - f(\boldsymbol{x}^\star)\big)}{KR\eta} + \left( \frac{72KL\eta}{m} + \frac{24(d+4)L\eta}{mP} \right) \sigma_G^2 \\
& + \frac{16L(d+4)\eta}{mP} \sigma^2 + 2\mu^2 L^2 (d+3)^3,
\end{aligned} \tag{29}
$$

*where $\boldsymbol{x}_r$ is the model parameter in the $r$-th round, $P$ is the number of perturbations, $\boldsymbol{x}^*$ is the optimal point, $K$ is the number of local update steps, $R$ is the number of communication rounds, $d$ is the dimension of model parameters, and $m$ is the number of sampled clients in each round.*

*Proof.* First, applying the $L-$Lipschitz smooth property on the global loss function $f$, we have

$$
f(\boldsymbol{x}_{r+1}) \leq f(\boldsymbol{x}_r) + \langle \nabla f(\boldsymbol{x}_r), \boldsymbol{x}_{r+1} - \boldsymbol{x}_r \rangle + \frac{L}{2} \|\boldsymbol{x}_{r+1} - \boldsymbol{x}_r\|^2 \tag{30}
$$

$$
= f(\boldsymbol{x}_r) - \eta \left\langle \nabla f(\boldsymbol{x}_r), \frac{1}{m} \sum_{i \in C_r} \sum_{k=1}^K G_{i,r}^k \right\rangle + \eta^2 \frac{L}{2} \left\| \frac{1}{m} \sum_{i \in C_r} \sum_{k=1}^K G_{i,r}^k \right\|^2, \tag{31}
$$

Taking conditional expectation $\mathbb{E}_r$ given the filtration $\boldsymbol{x}_r$ and information before round $r$, we obtain

$$\mathbb{E}_r[f(\boldsymbol{x}_{r+1})] \leq f(\boldsymbol{x}_r) \underbrace{- \eta \mathbb{E}_r \left\langle \nabla f(\boldsymbol{x}_r), \frac{1}{m} \sum_{i \in C_r} \sum_{k=1}^{K} G_{i,r}^k \right\rangle}_{A_1} + \underbrace{\frac{L}{2} \eta^2 \mathbb{E}_r \left\| \frac{1}{m} \sum_{i \in C_r} \sum_{k=1}^{K} G_{i,r}^k \right\|^2}_{A_2} \quad (32)$$

Observing $\mathbb{E}_{r,\xi} \left[ \frac{1}{M} \sum_{i=1}^{M} \sum_{k=1}^{K} \left( G_{i,r}^k - \nabla f_i^\mu(\boldsymbol{x}_{i,r}^k) \right) \right] = 0$, the cross product term $A_1$ satisfies

$$A_1 = - K\eta \mathbb{E}_r \left[ \left\langle \nabla f(\boldsymbol{x}_r), \frac{1}{MK} \sum_{i=1}^{M} \sum_{k=1}^{K} \nabla f_i^\mu(\boldsymbol{x}_{i,r}^k) \right\rangle \right] \quad (33)$$

Utilizing the Parallelogram Identity, we know

$$\begin{aligned}
A_1 = &- \frac{1}{2} K\eta \|\nabla f(\boldsymbol{x}_r)\|^2 - \frac{1}{2} K\eta \mathbb{E}_r \left\| \frac{1}{MK} \sum_{i=1}^{M} \sum_{k=1}^{K} \nabla f_i^\mu(\boldsymbol{x}_{i,r}^k) \right\|^2 \\
&+ \frac{1}{2} K\eta \mathbb{E}_r \left\| \frac{1}{MK} \sum_{i=1}^{M} \sum_{k=1}^{K} \left[ \nabla f_i^\mu(\boldsymbol{x}_{i,r}^k) - \nabla f_i(\boldsymbol{x}_r) \right] \right\|^2 \\
\leq &- \frac{1}{2} K\eta \|\nabla f(\boldsymbol{x}_r)\|^2 - \frac{1}{2} K\eta \mathbb{E}_r \left\| \frac{1}{MK} \sum_{i=1}^{M} \sum_{k=1}^{K} \nabla f_i^\mu(\boldsymbol{x}_{i,r}^k) \right\|^2 \\
&+ \frac{1}{2} K\eta \frac{1}{MK} \mathbb{E}_r \sum_{i=1}^{M} \sum_{k=1}^{K} \left\| \nabla f_i^\mu(\boldsymbol{x}_{i,r}^k) - \nabla f_i(\boldsymbol{x}_r) \right\|^2 \\
\leq &- \frac{1}{2} K\eta \|\nabla f(\boldsymbol{x}_r)\|^2 - \frac{1}{2} K\eta \mathbb{E}_r \left\| \frac{1}{MK} \sum_{i=1}^{M} \sum_{k=1}^{K} \nabla f_i^\mu(\boldsymbol{x}_{i,r}^k) \right\|^2 \\
&+ \frac{\eta}{M} \mathbb{E}_r \sum_{i=1}^{M} \sum_{k=1}^{K} \left\| \nabla f_i^\mu(\boldsymbol{x}_{i,r}^k) - \nabla f_i(\boldsymbol{x}_{i,r}^k) \right\|^2 + \frac{\eta}{M} \mathbb{E}_r \sum_{i=1}^{M} \sum_{k=1}^{K} \left\| \nabla f_i(\boldsymbol{x}_{i,r}^k) - \nabla f_i(\boldsymbol{x}_r) \right\|^2 \\
\leq &- \frac{1}{2} K\eta \|\nabla f(\boldsymbol{x}_r)\|^2 - \frac{1}{2} K\eta \mathbb{E}_r \left\| \frac{1}{MK} \sum_{i=1}^{M} \sum_{k=1}^{K} \nabla f_i^\mu(\boldsymbol{x}_{i,r}^k) \right\|^2 + \frac{1}{4} \mu^2 K L^2 (d+3)^3 \eta \\
&+ \frac{L^2 \eta}{M} \sum_{k=1}^{K} \sum_{i=1}^{M} \mathbb{E}_r \left\| \boldsymbol{x}_{i,r}^k - \boldsymbol{x}_r \right\|^2,
\end{aligned}$$

where we utilize Jensen's Inequality in the first two inequalities and apply $L$-smoothness and 17 to get the last inequality. Next, we focus on the quadratic term $A_2$.

$$\begin{aligned}
A_2 = &\frac{L}{2} \eta^2 \mathbb{E}_r \left\| \frac{1}{m} \sum_{i \in C_r} \sum_{k=1}^{K} G_{i,r}^k \right\|^2 \\
\leq &L\eta^2 \mathbb{E}_r \left\| \frac{1}{m} \sum_{i \in C_r} \sum_{k=1}^{K} [G_{i,r}^k - \nabla f_i^\mu(\boldsymbol{x}_{i,r}^k)] \right\|^2 + L\eta^2 \mathbb{E}_r \left\| \frac{1}{m} \sum_{i \in C_r} \sum_{k=1}^{K} \nabla f_i^\mu(\boldsymbol{x}_{i,r}^k) \right\|^2 \\
= &\frac{L\eta^2}{mM} \sum_{i=1}^{M} \sum_{k=1}^{K} \mathbb{E}_r \left\| G_{i,r}^k - \nabla f_i^\mu(\boldsymbol{x}_{i,r}^k) \right\|^2 + L\eta^2 \mathbb{E}_r \left\| \frac{1}{m} \sum_{i \in C_r} \sum_{k=1}^{K} \nabla f_i^\mu(\boldsymbol{x}_{i,r}^k) \right\|^2 \\
\leq &\frac{6L^3(d+4)\eta^2}{mMP} \sum_{i=1}^{M} \sum_{k=1}^{K} \mathbb{E}_r \left\| \boldsymbol{x}_{i,r}^k - \boldsymbol{x}_r \right\|^2 + \frac{KL\eta^2}{m} \left[ \frac{6(d+4)}{P} \|\nabla f(\boldsymbol{x}_r)\|^2 + \frac{6(d+4)}{P} \sigma_G^2 \right]
\end{aligned}$$

$$+ \frac{KL\eta^2}{m} \left[ \frac{2(d+4)}{P}\sigma^2 + \frac{\mu^2 L^2 (d+6)^3}{2P} \right] + \underbrace{L\eta^2 \mathbb{E}_r \left\| \frac{1}{m} \sum_{i \in C_r} \sum_{k=1}^{K} \nabla f_i^{\mu}(\boldsymbol{x}_{i,r}^k) \right\|^2}_{A_3}, \tag{34}$$

where we applied Jensen's inequality in the first inequality; the second equality holds since each term $\left[ G_{i,r}^k - \nabla f_i^{\mu}(\boldsymbol{x}_{i,r}^k) \right]$ is zero mean and independent to each other; the last inequality utilized the Lemma 3. For $A_3$, it can be bounded as follows

$$A_3 = L\eta^2 \mathbb{E}_r \left\| \frac{1}{m} \sum_{i \in C_r} \sum_{k=1}^{K} \nabla f_i^{\mu}(\boldsymbol{x}_{i,r}^k) \right\|^2$$

$$= L\eta^2 \mathbb{E}_r \left\| \frac{1}{M} \sum_{i=1}^{M} \sum_{k=1}^{K} \nabla f_i^{\mu}(\boldsymbol{x}_{i,r}^k) \right\|^2 + \underbrace{L\eta^2 \mathbb{E}_r \left\| \frac{1}{m} \sum_{i \in C_r} \sum_{k=1}^{K} \nabla f_i^{\mu}(\boldsymbol{x}_{i,r}^k) - \frac{1}{M} \sum_{i=1}^{M} \sum_{k=1}^{K} \nabla f_i^{\mu}(\boldsymbol{x}_{i,r}^k) \right\|^2}_{A_4}$$

$$\tag{35}$$

Continuing bounding the $A_4$ term, we have

$$A_4 = \mathbb{E}_r \left\| \frac{1}{m} \sum_{i \in C_r} \sum_{k=1}^{K} \nabla f_i^{\mu}(\boldsymbol{x}_{i,r}^k) - \frac{1}{M} \sum_{i=1}^{M} \sum_{k=1}^{K} \nabla f_i^{\mu}(\boldsymbol{x}_{i,r}^k) \right\|^2 \tag{36}$$

$$\overset{(a)}{\leq} 3\mathbb{E}_r \left\| \frac{1}{m} \sum_{i \in C_r} \sum_{k=1}^{K} \left[ \nabla f_i^{\mu}(\boldsymbol{x}_{i,r}^k) - \nabla f_i(\boldsymbol{x}_{i,r}^k) \right] \right\|^2$$

$$+ 3\mathbb{E}_r \left\| \frac{1}{m} \sum_{i \in C_r} \sum_{k=1}^{K} \nabla f_i(\boldsymbol{x}_{i,r}^k) - \frac{1}{M} \sum_{i=1}^{M} \sum_{k=1}^{K} \nabla f_i(\boldsymbol{x}_{i,r}^k) \right\|^2$$

$$+ 3\mathbb{E}_r \left\| \frac{1}{M} \sum_{i=1}^{M} \sum_{k=1}^{K} \left[ \nabla f_i(\boldsymbol{x}_{i,r}^k) - \nabla f_i^{\mu}(\boldsymbol{x}_{i,r}^k) \right] \right\|^2$$

$$\overset{(b)}{\leq} \frac{3}{2} \mu^2 K^2 L^2 (d+3)^3 + \underbrace{3\mathbb{E}_r \left\| \frac{1}{m} \sum_{i \in C_r} \sum_{k=1}^{K} \nabla f_i(\boldsymbol{x}_{i,r}^k) - \frac{1}{M} \sum_{i=1}^{M} \sum_{k=1}^{K} \nabla f_i(\boldsymbol{x}_{i,r}^k) \right\|^2}_{A_5}, \tag{37}$$

where in step (a), we plus and minus the $\frac{1}{m} \sum_{i \in C_r} \sum_{k=1}^{K} \nabla f_i(\boldsymbol{x}_{i,r}^k)$ and $\frac{1}{M} \sum_{i=1}^{M} \sum_{k=1}^{K} \nabla f_i(\boldsymbol{x}_{i,r}^k)$ then applies Jensen's inequality; in step (b), we restore to the lemma 2 on the first and last terms. Next, we use a similar trick to bound $A_5$:

$$A_5 = 3\mathbb{E}_r \left\| \frac{1}{m} \sum_{i \in C_r} \sum_{k=1}^{K} \nabla f_i(\boldsymbol{x}_{i,r}^k) - \frac{1}{m} \sum_{i \in C_r} \sum_{k=1}^{K} \nabla f_i(\boldsymbol{x}_r) + \frac{1}{m} \sum_{i \in C_r} \sum_{k=1}^{K} \nabla f_i(\boldsymbol{x}_r) \right.$$

$$\left. - \frac{1}{M} \sum_{i=1}^{M} \sum_{k=1}^{K} \nabla f_i(\boldsymbol{x}_r) + \frac{1}{M} \sum_{i=1}^{M} \sum_{k=1}^{K} \nabla f_i(\boldsymbol{x}_r) - \frac{1}{M} \sum_{i=1}^{M} \sum_{k=1}^{K} \nabla f_i(\boldsymbol{x}_{i,r}^k) \right\|^2$$

$$\overset{(a)}{\leq} 9\mathbb{E}_r \left\| \frac{1}{m} \sum_{i \in C_r} \sum_{k=1}^{K} \left[ \nabla f_i(\boldsymbol{x}_{i,r}^k) - \nabla f_i(\boldsymbol{x}_r) \right] \right\|^2$$

$$+ 9\mathbb{E}_r \left\| \frac{1}{m} \sum_{i \in C_r} \sum_{k=1}^{K} \nabla f_i(\boldsymbol{x}_r) - \frac{1}{M} \sum_{i=1}^{M} \sum_{k=1}^{K} \nabla f_i(\boldsymbol{x}_r) \right\|^2$$

$$+ 9\mathbb{E}_r \left\| \frac{1}{M} \sum_{i=1}^{M} \sum_{k=1}^{K} \left[ \nabla f_i(\boldsymbol{x}_r) - \nabla f_i(\boldsymbol{x}_{i,r}^k) \right] \right\|^2$$

$$\overset{(b)}{\leq} 18KL^2\mathbb{E}_r\left[\frac{1}{M}\sum_{i=1}^{M}\sum_{k=1}^{K}\left\|\boldsymbol{x}_{i,r}^k - \boldsymbol{x}_r\right\|^2\right] + 9K^2\mathbb{E}_r\left\|\frac{1}{m}\sum_{i\in C_r}\left[\nabla f_i(\boldsymbol{x}_r) - \nabla f(\boldsymbol{x}_r)\right]\right\|^2$$

$$\overset{(c)}{=} 18KL^2\mathbb{E}_r\left[\frac{1}{M}\sum_{i=1}^{M}\sum_{k=1}^{K}\left\|\boldsymbol{x}_{i,r}^k - \boldsymbol{x}_r\right\|^2\right] + \frac{9K^2}{m^2}\mathbb{E}_r\sum_{i\in C_r}\left\|\nabla f_i(\boldsymbol{x}_r) - \nabla f(\boldsymbol{x}_r)\right\|^2$$

$$\overset{(d)}{\leq} 18KL^2\mathbb{E}_r\left[\frac{1}{M}\sum_{i=1}^{M}\sum_{k=1}^{K}\left\|\boldsymbol{x}_{i,r}^k - \boldsymbol{x}_r\right\|^2\right] + \frac{9K^2}{m}\sigma_G^2,$$

where step (a) applies Jensen's inequality; step (b) utilizes the $L-$Lipschitze condition; the equality in step (c) holds because each term $\left[\nabla f_i(\boldsymbol{x}_r) - \nabla f_i(\boldsymbol{x}_{i,r}^k)\right]$ is independent and zero-mean; step (d) results from the data heterogeneous assumption.

Plugging $A_4$ and $A_5$ into $A_3$, we establish

$$A_3 \leq L\eta^2\mathbb{E}_r\left\|\frac{1}{M}\sum_{i=1}^{M}\sum_{k=1}^{K}\nabla f_i^\mu(\boldsymbol{x}_{i,r}^k)\right\|^2 + \frac{3}{2}\mu^2K^2L^3(d+3)^3\eta^2$$

$$+ 18KL^3\eta^2\mathbb{E}_r\left[\frac{1}{M}\sum_{i=1}^{M}\sum_{k=1}^{K}\left\|\boldsymbol{x}_{i,r}^k - \boldsymbol{x}_r\right\|^2\right] + \frac{9K^2L\eta^2}{m}\sigma_G^2 \tag{38}$$

Now, we are ready to put $A_3$ back to $A_2$ and group the terms

$$A_2 \leq \frac{6KL(d+4)\eta^2}{mP}\|\nabla f(\boldsymbol{x}_r)\|^2 + \frac{6L^3(d+4)\eta^2}{mMP}\sum_{i=1}^{M}\sum_{k=1}^{K}\mathbb{E}_r\left\|\boldsymbol{x}_{i,r}^k - \boldsymbol{x}_r\right\|^2$$

$$+ \frac{6KL(d+4)\eta^2}{mP}\sigma_G^2 + \frac{2KL(d+4)\eta^2}{mP}\sigma^2 + \frac{\mu^2KL^3(d+6)^3\eta^2}{2mP}$$

$$+ 18KL^3\eta^2\mathbb{E}_r\left[\frac{1}{M}\sum_{i=1}^{M}\sum_{k=1}^{K}\left\|\boldsymbol{x}_{i,r}^k - \boldsymbol{x}_r\right\|^2\right] + \frac{3}{2}\mu^2K^2L^3(d+3)^3\eta^2 + \frac{9K^2L\eta^2}{m}\sigma_G^2$$

$$+ L\eta^2\mathbb{E}_r\left\|\frac{1}{M}\sum_{i=1}^{M}\sum_{k=1}^{K}\nabla f_i^\mu(\boldsymbol{x}_{i,r}^k)\right\|^2 \tag{39}$$

Combining all pieces and denoting $\chi_r = \mathbb{E}_r\left[\frac{1}{M}\sum_{i=1}^{M}\sum_{k=1}^{K}\left\|\boldsymbol{x}_{i,r}^k - \boldsymbol{x}_r\right\|^2\right]$, we have

$$\mathbb{E}_r[f(\boldsymbol{x}_{r+1})] \leq f(\boldsymbol{x}_r) - \frac{1}{2}K\eta\|\nabla f(\boldsymbol{x}_r)\|^2 - \frac{1}{2}K\eta\mathbb{E}_r\left\|\frac{1}{MK}\sum_{i=1}^{M}\sum_{k=1}^{K}\nabla f_i^\mu(\boldsymbol{x}_{i,r}^k)\right\|^2 + \frac{1}{4}\mu^2KL^2(d+3)^3\eta$$

$$+ L^2\eta\chi_r + \frac{6(d+4)LK\eta^2}{mP}\|\nabla f(\boldsymbol{x}_r)\|^2 + \frac{6L^3(d+4)\eta^2}{mP}\chi_r$$

$$+ \frac{6(d+4)LK\eta^2}{mP}\sigma_G^2 + \frac{2(d+4)LK\eta^2}{mP}\sigma^2 + \frac{\mu^2(d+6)^3L^3K\eta^2}{2mP}$$

$$+ 18KL^3\eta^2\chi_r + \frac{3}{2}\mu^2K^2L^3\eta^2(d+3)^3$$

$$+ \frac{9K^2L\eta^2}{m}\sigma_G^2 + L\eta^2\mathbb{E}_r\left\|\frac{1}{M}\sum_{i=1}^{M}\sum_{k=1}^{K}\nabla f_i^\mu(\boldsymbol{x}_{i,r}^k)\right\|^2 \tag{40}$$

$$\leq f(\boldsymbol{x}_r) - \left(\frac{1}{2}K\eta - \frac{6(d+4)LK\eta^2}{mP}\right)\|\nabla f(\boldsymbol{x}_r)\|^2 + \frac{1}{4}\mu^2KL^2(d+3)^3\eta$$

$$+ \left(L^2\eta + \frac{6L^3(d+4)\eta^2}{mP} + 18KL^3\eta^2\right)\chi_r + \frac{2KL(d+4)\eta^2}{mP}\sigma^2$$

$$+ \left(\frac{9K^2L\eta^2}{m} + \frac{6(d+4)LK\eta^2}{mP}\right)\sigma_G^2 + \frac{\mu^2KL^3(d+6)^3\eta^2}{2mP} + \frac{3}{2}\mu^2K^2L^3(d+3)^3\eta^2 \tag{41}$$

Plugging lemma 5 into 41 and following $\eta \leq \min \left\{ \frac{mP}{24L(d+4)}, \frac{2P}{mKL(d+P+4)}, \frac{1}{mK^2L}, \frac{mP(d+3)^3}{2L[3mPK(d+3)^3+(d+6)^3]} \right\}$, we can further simplified it into

$$\mathbb{E}_r[f(\boldsymbol{x}_{r+1})] \leq f(\boldsymbol{x}_r) - \frac{1}{4}K\eta\|\nabla f(\boldsymbol{x}_r)\|^2 + \left( \frac{18K^2L\eta^2}{m} + \frac{6(d+4)LK\eta^2}{mP} \right) \sigma_G^2$$

$$+ \frac{4KL(d+4)\eta^2}{mP}\sigma^2 + \frac{1}{2}\mu^2KL^2(d+3)^3\eta \tag{42}$$

Rearranging the terms, we have

$$\frac{1}{4}K\eta\|\nabla f(\boldsymbol{x}_r)\|^2 \leq f(\boldsymbol{x}_r) - \mathbb{E}_r[f(\boldsymbol{x}_{r+1})] + \left( \frac{18K^2L\eta^2}{m} + \frac{6(d+4)LK\eta^2}{mP} \right) \sigma_G^2$$

$$+ \frac{4KL(d+4)\eta^2}{mP}\sigma^2 + \frac{1}{2}\mu^2KL^2(d+3)^3\eta \tag{43}$$

Dividing $\frac{1}{4}K\eta$ on both sides, then we get

$$\|\nabla f(\boldsymbol{x}_r)\|^2 \leq \frac{4}{K\eta}\Big(f(\boldsymbol{x}_r) - \mathbb{E}_r[f(\boldsymbol{x}_{r+1})]\Big) + \left( \frac{72KL\eta}{m} + \frac{24(d+4)L\eta}{mP} \right) \sigma_G^2$$

$$+ \frac{16L(d+4)\eta}{mP}\sigma^2 + 2\mu^2L^2(d+3)^3 \tag{44}$$

Recursively executing (44) $R$ rounds, we can obtain

$$\frac{1}{R}\sum_{r=0}^{R-1}\mathbb{E}_r\|\nabla f(\boldsymbol{x}_r)\|^2 \leq \frac{4\big(f(\boldsymbol{x}_0) - f(\boldsymbol{x}^*)\big)}{KR\eta} + \left( \frac{72KL\eta}{m} + \frac{24(d+4)L\eta}{mP} \right) \sigma_G^2$$

$$+ \frac{16L(d+4)\eta}{mP}\sigma^2 + 2\mu^2L^2(d+3)^3 \tag{45}$$

∎

### D.4 PROOF OF THEOREM 2

In this section, we only consider the case that local update step $K = 1$, so we ignore superscript $k$ in this proof. The following proof is inspired by the MeZO work (Malladi et al., 2023) and extends the proof for the multiple-client case. In (Malladi et al., 2023), it directly utilized the following form of the effective zeroth-order gradient:

$$\hat{\nabla}f(\boldsymbol{x}_r;\xi_r) = \frac{1}{mP}\sum_{i\in C_r}\sum_{p=1}^{P}g_{r,p}\cdot\boldsymbol{z}_{r,p}$$

$$= \frac{1}{mP}\sum_{i\in C_r}\sum_{p=1}^{P}\frac{1}{\mu}\big(f_i(\boldsymbol{x}_r + \mu\boldsymbol{z}_{r,p};\xi_{i,r}) - f_i(\boldsymbol{x}_r;\xi_{i,r})\big)\boldsymbol{z}_{r,p}$$

$$\neq \frac{1}{mP}\sum_{i\in C_r}\sum_{p=1}^{P}\boldsymbol{z}_{r,p}\boldsymbol{z}_{r,p}^{\top}\nabla f_i(\boldsymbol{x}_r;\xi_{i,r}) \tag{46}$$

**However, the last equality does not hold in general unless** $\mu \to 0$. Fortunately, leveraging the basic lemma stated in the previous section and using the mean value theorem, we can fix the proof. Using the mean-value theorem, we know

$$\frac{1}{\mu}\big(f_i(\boldsymbol{x}_r + \mu\boldsymbol{z}_{r,p};\xi_{i,r}) - f_i(\boldsymbol{x}_r;\xi_{i,r})\big) = \boldsymbol{z}_{r,p}^{\top}\nabla f_i(\boldsymbol{x}'_{r,p};\xi_{i,r}), \tag{47}$$

where $\boldsymbol{x}'_{r,p} = \boldsymbol{x}_r + \mu\theta\boldsymbol{z}_{r,p}$ for some $\theta \in [0,1]$ (we do not need to know what value the $\theta$ is), so that the effective zeroth-order gradient we utilized is

$$\hat{\nabla}f(\boldsymbol{x}_r;\xi_r) = \frac{1}{mP}\sum_{i\in C_r}\sum_{p=1}^{P}\boldsymbol{z}_{r,p}\boldsymbol{z}_{r,p}^{\top}\nabla f_i(\boldsymbol{x}'_{r,p};\xi_{i,r}) \tag{48}$$

To light the notation, we introduce

$$\bar{\nabla} f(\boldsymbol{x}_r; \xi_r) := \frac{1}{mP} \sum_{i \in C_r} \sum_{p=1}^{P} \boldsymbol{z}_{r,p} \boldsymbol{z}_{r,p}^{\top} \nabla f_i(\boldsymbol{x}_r; \xi_{i,r}) \tag{49}$$

$$\epsilon_r := \frac{1}{mP} \sum_{i \in C_r} \sum_{p=1}^{P} \boldsymbol{z}_{r,p} \boldsymbol{z}_{r,p}^{\top} \Big( \nabla f_i(\boldsymbol{x}'_{r,p}; \xi_{i,r}) - \nabla f_i(\boldsymbol{x}_r; \xi_{i,r}) \Big) \tag{50}$$

so that we can compactly write $\hat{\nabla} f(\boldsymbol{x}_r; \xi_r) = \bar{\nabla} f(\boldsymbol{x}_r; \xi_r) + \epsilon_r$. The motivation of introducing $\bar{\nabla} f(\boldsymbol{x}_r; \xi_r)$ is because it is much easier to handle compared with $\hat{\nabla} f(\boldsymbol{x}_r; \xi_r)$ and it is straightforward to bound the difference:

$$\begin{aligned}
\|\epsilon_r\|^2 \leq & \frac{1}{mP} \sum_{i \in C_r} \sum_{p=1}^{P} \|\boldsymbol{z}_{r,p} \boldsymbol{z}_{r,p}^{\top} \Big( \nabla f_i(\boldsymbol{x}'_{r,p}; \xi_{i,r}) - \nabla f_i(\boldsymbol{x}_r; \xi_{i,r}) \Big) \|^2 \\
\leq & \frac{L^2}{mP} \sum_{i \in C_r} \sum_{p=1}^{P} \|\boldsymbol{z}_{r,p}^{\top} \boldsymbol{z}_{r,p}\|^2 \|\boldsymbol{x}'_{r,p} - \boldsymbol{x}_r\|^2 \\
\leq & \frac{L^2 \mu^2}{mP} \sum_{i \in C_r} \sum_{p=1}^{P} \|\boldsymbol{z}_{r,p}\|^6 \\
= & L^2 \mu^2 d^3. \tag{51}
\end{aligned}$$

Another difference from the proof in (Malladi et al., 2023) is noticing the conditional expectation of the stochastic zeroth-order gradient is unbiased to the Gaussian smooth function $f^{\mu}$

$$\mathbb{E}_r \left[ \hat{\nabla} f(\boldsymbol{x}_r; \xi_r) \right] = \mathbb{E}_r \left[ \frac{1}{m} \sum_{i \in C_r} \nabla f_i^{\mu}(\boldsymbol{x}_r) \right] = \nabla f^{\mu}(\boldsymbol{x}_r) \tag{52}$$

Notice it is not unbiased to the gradient of the original function, i.e., $\mathbb{E}_r[\hat{\nabla} f(\boldsymbol{x}_r; \xi_r)] \neq \nabla f(\boldsymbol{x}_r)$, in general. The covariance matrix of $\hat{\nabla} f(\boldsymbol{x}_r; \xi_r)$ has the following relationship.

**Lemma 6** *The covariance matrix of the stochastic zeroth-order gradient $\bar{\nabla} f(\boldsymbol{x}_r; \xi_r)$ is equivalent to*

$$\begin{aligned}
\mathbb{E} \left[ \bar{\nabla} f(\boldsymbol{x}_r; \xi_r) \bar{\nabla} f(\boldsymbol{x}_r; \xi_r)^{\top} \right] = & \frac{d}{P(d+2)} \left( \|\nabla f(\boldsymbol{x}_r)\|^2 + \frac{1}{m} \mathrm{Tr}(\Sigma_r) + \frac{1}{m} \mathrm{Tr}(\Delta_{f,r}) \right) \cdot \boldsymbol{I} \\
& + \left( 1 + \frac{d-2}{P(d+2)} \right) \left( \nabla f(\boldsymbol{x}_r) \nabla f(\boldsymbol{x}_r)^{\top} + \frac{1}{m} \Sigma_r + \frac{1}{m} \Delta_{f,r} \right),
\end{aligned} \tag{53}$$

*where the stochastic gradient noise is denoted as $\boldsymbol{s}_{i,r} = \nabla f_i(\boldsymbol{x}_r; \xi_r) - \nabla f_i(\boldsymbol{x}_r)$, $d$ is the dimension of model parameters, $P$ is the number of perturbations, and the definitions of $\Sigma_r$ and $\Delta_{f,r}$ are*

$$\Sigma_r = \frac{1}{M} \sum_{i=1}^{M} \boldsymbol{s}_{i,r} \boldsymbol{s}_{i,r}^{\top} \tag{54}$$

$$\Delta_{f,r} = \frac{1}{M} \sum_{i=1}^{M} \nabla f_i(\boldsymbol{x}_r) \nabla f_i(\boldsymbol{x}_r)^{\top} - \nabla f(\boldsymbol{x}_r) \nabla f(\boldsymbol{x}_r)^{\top} \tag{55}$$

*$\Sigma_r$ can be understood as the covariance matrix introduced by the data randomness and $\Delta_{f,r}$ as the covariance matrix introduced by sampling the client's different local loss functions.*

*Proof.* Substituting the equation (49) of $\bar{\nabla} f(\boldsymbol{x}_r; \xi_r)$ into the covariance matrix, we obtain

$$\mathbb{E} \left[ \bar{\nabla} f(\boldsymbol{x}_r; \xi_r) \bar{\nabla} f(\boldsymbol{x}_r; \xi_r)^{\top} \right] = \underbrace{\mathbb{E} \left[ \frac{1}{m^2 P^2} \sum_{i,i' \in C_r} \sum_{p,p'=1}^{P} \boldsymbol{z}_p \boldsymbol{z}_p^{\top} \nabla f_i(\boldsymbol{x}_r) \nabla f_{i'}(\boldsymbol{x}_r)^{\top} \boldsymbol{z}_{p'} \boldsymbol{z}_{p'}^{\top} \right]}_{:= T_1}$$

$$+ \mathbb{E}\left[\frac{1}{m^2 P^2} \sum_{i,i'\in C_r} \sum_{p,p'=1}^{P} \boldsymbol{z}_p \boldsymbol{z}_p^\top \boldsymbol{s}_{i,r} \boldsymbol{s}_{i,r}^\top \boldsymbol{z}_{p'} \boldsymbol{z}_{p'}^\top\right]}_{:=T_2}, \qquad (56)$$

where we dropped the cross term due to the zero mean and independent properties of the stochastic gradient noise $\boldsymbol{s}_{i,r}$. To find the value of $T_1$, we note that[5]

$$\mathbb{E}_{i,i'}\left[\sum_{i,i'\in C_r} \nabla f_i(\boldsymbol{x}_r)\nabla f_{i'}(\boldsymbol{x}_r)^\top\right]$$

$$=(m^2-m)\nabla f(\boldsymbol{x}_r)\nabla f(\boldsymbol{x}_r)^\top + \frac{m}{M}\sum_{i=1}^{M}\nabla f_i(\boldsymbol{x}_r)\nabla f_i(\boldsymbol{x}_r)^\top$$

$$=m^2\nabla f(\boldsymbol{x}_r)\nabla f(\boldsymbol{x}_r)^\top + m\underbrace{\left(\frac{1}{M}\sum_{i=1}^{M}\nabla f_i(\boldsymbol{x}_r)\nabla f_i(\boldsymbol{x}_r)^\top - \nabla f(\boldsymbol{x}_r)\nabla f(\boldsymbol{x}_r)^\top\right)}_{:=\Delta_{f,r}} \qquad (57)$$

(The real randomness comes from $C_r$ instead of $i,i'$. Here $\mathbb{E}_{i,i'}$ is just for simplicity.) We do not know what $\Delta_{f,r}$ is, but it can be bounded that

$$\mathrm{Tr}(\Delta_{f,r}) = \frac{1}{M}\sum_{i=1}^{M}\|\nabla f_i(\boldsymbol{x}_r)\|^2 - \|\nabla f(\boldsymbol{x}_r)\|^2 = \frac{1}{M}\sum_{i=1}^{M}\|\nabla f(\boldsymbol{x}_r)-\nabla f_i(\boldsymbol{x}_r)\|^2 \le \sigma_G^2 \qquad (58)$$

Focusing on the first term $T_1$, we have

$$T_1 = \frac{P-1}{m^2 P}\mathbb{E}_{i,i'}\left[\sum_{i,i'\in C_r}\nabla f_i(\boldsymbol{x}_r)\nabla f_{i'}(\boldsymbol{x}_r)^\top\right]$$

$$+\frac{1}{m^2 P^2}\mathbb{E}_{i,i'}\left[\sum_{i,i'\in C_r}\mathbb{E}_{\boldsymbol{z}_p}\left[\sum_{p=1}^{P}\boldsymbol{z}_p\boldsymbol{z}_p^\top\nabla f_i(\boldsymbol{x}_r)\nabla f_{i'}(\boldsymbol{x}_r)^\top\boldsymbol{z}_p\boldsymbol{z}_p^\top\right]\right]$$

$$=\frac{P-1}{P}\nabla f(\boldsymbol{x}_r)\nabla f(\boldsymbol{x}_r)^\top + \frac{P-1}{mP}\Delta_{f,r} + \frac{1}{P^2}\mathbb{E}_{\boldsymbol{z}_p}\left[\sum_{p=1}^{P}\boldsymbol{z}_p\boldsymbol{z}_p^\top\nabla f(\boldsymbol{x}_r)\nabla f(\boldsymbol{x}_r)^\top\boldsymbol{z}_p\boldsymbol{z}_p^\top\right]$$

$$+\frac{1}{mP^2}\mathbb{E}_{\boldsymbol{z}_p}\left[\sum_{p=1}^{P}\boldsymbol{z}_p\boldsymbol{z}_p^\top\Delta_{f,r}\boldsymbol{z}_p\boldsymbol{z}_p^\top\right]$$

$$=\frac{P-1}{P}\nabla f(\boldsymbol{x}_r)\nabla f(\boldsymbol{x}_r)^\top + \frac{P-1}{mP}\Delta_{f,r} + \frac{d}{P(d+2)}\|\nabla f(\boldsymbol{x}_r)\|^2\boldsymbol{I} + \frac{2d}{P(d+2)}\nabla f(\boldsymbol{x}_r)\nabla f(\boldsymbol{x}_r)^\top$$

$$+\frac{d}{mP(d+2)}\mathrm{Tr}(\Delta_{f,r})\boldsymbol{I} + \frac{2d}{mP(d+2)}\Delta_{f,r},$$

where the first equality is split according to $p=p'$ and $p\neq p'$, and we plug the established result (57) into the second equality, the third one is because of the properties of uniform distribution

$$\mathbb{E}\left[\boldsymbol{z}_p\boldsymbol{z}_p^\top uv^\top \boldsymbol{z}_p\boldsymbol{z}_p^\top\right] = \frac{d}{d+2}\mathrm{Tr}(uv^\top)I + \frac{2d}{d+2}uv^\top. \qquad (59)$$

See the proof of lemma 2 in (Malladi et al., 2023).

Next, the stochastic gradient noise satisfies

$$\mathbb{E}_{i,i'}\left[\sum_{i,i'\in C_r}\boldsymbol{s}_{i,r}\boldsymbol{s}_{i,r}^\top\right] = \frac{m}{M}\sum_{i=1}^{M}\boldsymbol{s}_{i,r}\boldsymbol{s}_{i,r}^\top := m\Sigma_r, \qquad \text{where } \mathrm{Tr}(\Sigma_r) \le \sigma^2. \qquad (60)$$

---

[5] For the proof here, the notation $\mathbb{E}_x$ means the expectation with respect to $x$ instead of the given condition on $x$ or filtration before $x$.

The second term $T_2$ is similar but easier:

$$
\begin{aligned}
T_2 &= \frac{1}{mP^2} \mathbb{E}_{\boldsymbol{z}_p, \boldsymbol{z}_p'} \left[ \sum_{p,p'=1}^{P} \boldsymbol{z}_p \boldsymbol{z}_p^\top \Sigma_r \boldsymbol{z}_{p'} \boldsymbol{z}_{p'}^\top \right] \\
&= \frac{P-1}{mP} \Sigma_r + \frac{d}{mP(d+2)} \left( \text{Tr}(\Sigma_r) \boldsymbol{I} + 2\Sigma_r \right)
\end{aligned}
\tag{61}
$$

Combining the result $T_1$ and $T_2$, we establish

$$
\begin{aligned}
\mathbb{E}\left[ \bar{\nabla} f(\boldsymbol{x}_r; \xi_r) \bar{\nabla} f(\boldsymbol{x}_r'; \xi_r)^\top \right] =& \frac{d}{P(d+2)} \left( \|\nabla f(\boldsymbol{x}_r)\|^2 + \frac{1}{m} \text{Tr}(\Sigma_r) + \frac{1}{m} \text{Tr}(\Delta_{f,r}) \right) \cdot \boldsymbol{I} \\
&+ \frac{P-1}{P} \left( \nabla f(\boldsymbol{x}_r) \nabla f(\boldsymbol{x}_r)^\top + \frac{1}{m} \Delta_{f,r} + \frac{1}{m} \Sigma_r \right) \\
&+ \frac{2d}{P(d+2)} \left( \nabla f(\boldsymbol{x}_r) \nabla f(\boldsymbol{x}_r)^\top + \frac{1}{m} \Delta_{f,r} + \frac{1}{m} \Sigma_r \right)
\end{aligned}
$$

Regrouping the terms and simplifying the coefficients concludes the proof of this lemma. ∎

To ease the reference, we restate the Theorem here again.

**Theorem 4 (Restated; Convergence of DeComFL with $\kappa$-Effective Rank)** *Under the assumptions 1, 2, 3 and 4, supposing $\eta \leq \frac{1}{4L} \left( 1 + \frac{d\kappa + d - 2}{P(d+2)} \right)^{-1}$ and drawing $\boldsymbol{z}_i^r$ from the unit ball with radius $\sqrt{d}$, it holds*

$$
\frac{1}{R} \sum_{r=0}^{R-1} \mathbb{E} \|\nabla f(\boldsymbol{x}_r)\|^2 \leq \frac{4(f(\boldsymbol{x}_0) - f(\boldsymbol{x}^\star))}{R\eta} + \frac{2L\eta}{m} \left( 1 + \frac{d\kappa + d - 2}{P(d+2)} \right) (\sigma_G^2 + \sigma^2)
$$

$$
+ \frac{1}{2} \mu^2 L^2 (d+3)^3 + 4\mu^2 L^4 d^3 \eta
$$

*Selecting $\eta = \mathcal{O}\left( \frac{\sqrt{mP}}{\sqrt{R\kappa}} \right)$ and $\mu \leq \frac{\sqrt[4]{\kappa}}{\sqrt[4]{mRP}\sqrt{(d+3)^3}}$, we can get*

$$
\frac{1}{R} \sum_{r=0}^{R-1} \mathbb{E} \|\nabla f(\boldsymbol{x}_r)\|^2 = \mathcal{O}\left( \frac{\sqrt{\kappa}}{\sqrt{mRP}} \right) + \mathcal{O}\left( \left( \frac{\sqrt{P}}{\sqrt{mR\kappa}} + \frac{\sqrt{\kappa}}{\sqrt{mRP}} \right) (\sigma_G^2 + \sigma^2) \right).
\tag{62}
$$

*Further suppose $\kappa \gg P$, the convergence rate is $\mathcal{O}\left( \frac{\sqrt{\kappa}}{\sqrt{mRP}} \right)$ when the algorithm runs with sufficient large round $R$.* ∎

*Proof.* Taking the conditional expectation over the recursion of the loss function $f$ and expanding the function value via Taylor's theorem:

$$
\mathbb{E}_r[f(\boldsymbol{x}_{r+1})]
$$

$$
= f(\boldsymbol{x}_r) - \eta \left\langle \nabla f(\boldsymbol{x}_r), \mathbb{E}\left[ \hat{\nabla} f(\boldsymbol{x}_r; \xi_r) \right] \right\rangle + \frac{\eta^2}{2} \mathbb{E}\left[ \hat{\nabla} f(\boldsymbol{x}_r; \xi_r)^\top \nabla^2 f(\boldsymbol{x}_r') \hat{\nabla} f(\boldsymbol{x}_r; \xi_r) \right]
$$

$$
\leq f(\boldsymbol{x}_r) - \eta \langle \nabla f(\boldsymbol{x}_r), \nabla f^\mu(\boldsymbol{x}_r) \rangle + \eta^2 \mathbb{E}\left[ \bar{\nabla} f(\boldsymbol{x}_r; \xi_r)^\top \nabla^2 f(\boldsymbol{x}_r') \bar{\nabla} f(\boldsymbol{x}_r; \xi_r) \right] + \eta^2 L^2 \|\epsilon_r\|^2, \tag{63}
$$

where the $\boldsymbol{x}_r'$ in the first equality is some value lying between $\boldsymbol{x}_r$ and $\boldsymbol{x}_{r+1}$ and the inequality applied the Jensen's inequality on the quadratic term. Using the identity $\langle a, b \rangle = \frac{1}{2}(\|a\|^2 + \|b\|^2 - \|a-b\|^2)$, we have

$$
\begin{aligned}
\mathbb{E}_r[f(\boldsymbol{x}_{r+1})] \leq & f(\boldsymbol{x}_r) - \frac{\eta}{2} \|\nabla f(\boldsymbol{x}_r)\|^2 - \frac{\eta}{2} \|\nabla f^\mu(\boldsymbol{x}_r)\|^2 + \frac{\eta}{2} \|\nabla f(\boldsymbol{x}_r) - \nabla f^\mu(\boldsymbol{x}_r)\|^2 \\
&+ \eta^2 \mathbb{E}\left[ \bar{\nabla} f(\boldsymbol{x}_r; \xi_r)^\top \nabla^2 f(\boldsymbol{x}_r') \bar{\nabla} f(\boldsymbol{x}_r; \xi_r) \right] + \eta^2 L^2 \|\epsilon_r\|^2
\end{aligned}
\tag{64}
$$

Discarding $\|\nabla f^\mu(\boldsymbol{x}_r)\|^2$ term and applying (17) and (51), we have

$$
\mathbb{E}_r[f(\boldsymbol{x}_{r+1})] \leq f(\boldsymbol{x}_r) - \frac{\eta}{2} \|\nabla f(\boldsymbol{x}_r)\|^2 + \frac{\eta \mu^2 L^2}{8} (d+3)^3
$$

$$+ \eta^2 \mathbb{E}\left[ \bar{\nabla} f(\boldsymbol{x}_r; \xi_r)^\top \nabla^2 f(\boldsymbol{x}'_r) \bar{\nabla} f(\boldsymbol{x}_r; \xi_r) \right] + \eta^2 L^4 \mu^2 d^3 \tag{65}$$

Based on the assumption 4, we bound the Hessian $\nabla^2 f^\mu(\boldsymbol{x}'_r)$ by $\boldsymbol{H}(\boldsymbol{x}_r)$ and arrive

$$\begin{aligned}
\mathbb{E}_r[f(\boldsymbol{x}_{r+1})] \leq & f(\boldsymbol{x}_r) - \frac{\eta}{2}\|\nabla f^\mu(\boldsymbol{x}_r)\|^2 + \frac{\eta\mu^2 L^2}{8}(d+3)^3 + \eta^2 L^4 \mu^2 d^3 \\
& + \eta^2 \underbrace{\left\langle \boldsymbol{H}(\boldsymbol{x}_r), \mathbb{E}\left[ \bar{\nabla} f(\boldsymbol{x}_r; \xi_r) \bar{\nabla} f(\boldsymbol{x}_r; \xi_r)^\top \right] \right\rangle}_{:=T_3}
\end{aligned} \tag{66}$$

Next we focus on bounding the term $T_3$. Plugging the conclusion of Lemma 6, we get

$$\begin{aligned}
T_3 = & \frac{d}{P(d+2)} \left( \|\nabla f(\boldsymbol{x}_r)\|^2 + \frac{1}{m}\mathrm{Tr}(\Sigma_r) + \frac{1}{m}\mathrm{Tr}(\Delta_{f,r}) \right) \mathrm{Tr}(\boldsymbol{H}(\boldsymbol{x}_r)) \\
& \left( 1 + \frac{d-2}{P(d+2)} \right) \left( \nabla f(\boldsymbol{x}_r)^\top \boldsymbol{H}(\boldsymbol{x}_r)\nabla f(\boldsymbol{x}_r) + \frac{1}{m}\langle \Sigma_r, \boldsymbol{H}(\boldsymbol{x}_r) \rangle + \frac{1}{m}\langle \Delta_{f,r}, \boldsymbol{H}(\boldsymbol{x}_r) \rangle \right)
\end{aligned}$$

By the assumption, the Hessian upper bound $\boldsymbol{H}(x_r)$ satisfies $\|\boldsymbol{H}(x_r)\|_2 \leq L$ and $\mathrm{Tr}(\boldsymbol{H}(x_r)) \leq L\kappa$. Thus, we obtain

$$\begin{aligned}
T_3 \leq & \frac{Ld\kappa}{P(d+2)} \left( \|\nabla f(\boldsymbol{x}_r)\|^2 + \frac{1}{m}\mathrm{Tr}(\Sigma_r) + \frac{1}{m}\mathrm{Tr}(\Delta_{f,r}) \right) \\
& + L \cdot \left( 1 + \frac{d-2}{P(d+2)} \right) \left( \|\nabla f(\boldsymbol{x}_r)\|^2 + \frac{1}{m}\mathrm{Tr}(\Sigma_r) + \frac{1}{m}\mathrm{Tr}(\Delta_{f,r}) \right) \tag{67} \\
= & L \left( 1 + \frac{d\kappa + d - 2}{P(d+2)} \right) \left( \|\nabla f(\boldsymbol{x}_r)\|^2 + \frac{1}{m}\mathrm{Tr}(\Sigma_r) + \frac{1}{m}\mathrm{Tr}(\Delta_{f,r}) \right) \\
\leq & L \left( 1 + \frac{d\kappa + d - 2}{P(d+2)} \right) \left( \|\nabla f(\boldsymbol{x}_r)\|^2 + \frac{1}{m}(\sigma_G^2 + \sigma^2) \right) \tag{68}
\end{aligned}$$

Substituting back, we obtain

$$\begin{aligned}
\mathbb{E}_r[f(\boldsymbol{x}_{r+1})] \leq & f(\boldsymbol{x}_r) - \frac{\eta}{2}\|\nabla f(\boldsymbol{x}_r)\|^2 + \frac{\eta\mu^2 L^2}{8}(d+3)^3 + \eta^2 L^4 \mu^2 d^3 \\
& + \eta^2 L \cdot \left( 1 + \frac{d\kappa + d - 2}{P(d+2)} \right) \left( \|\nabla f(\boldsymbol{x}_r)\|^2 + \frac{1}{m}(\sigma_G^2 + \sigma^2) \right) \tag{69}
\end{aligned}$$

To establish the convergence rate, we move the $\|\nabla f(\boldsymbol{x}_r)\|^2$ to the left-hand side and take the expectation over both sides

$$\begin{aligned}
& \eta \left( \frac{1}{2} - \eta L \cdot \left( 1 + \frac{d\kappa + d - 2}{P(d+2)} \right) \right) \mathbb{E}\|\nabla f(\boldsymbol{x}_r)\|^2 \\
& \leq \mathbb{E} f(\boldsymbol{x}_r) - \mathbb{E} f(\boldsymbol{x}_{r+1}) + \frac{\eta^2 L}{m} \left( 1 + \frac{d\kappa + d - 2}{P(d+2)} \right) (\sigma_G^2 + \sigma^2) + \frac{1}{8}\eta\mu^2 L^2 (d+3)^3 + \eta^2 L^4 \mu^2 d^3
\end{aligned}$$

Take telescoping sum from $r = 1$ to $R$ and require $\eta \leq \frac{1}{4L} \left( 1 + \frac{d\kappa+d-2}{P(d+2)} \right)^{-1}$, we obtain

$$\begin{aligned}
\frac{1}{R}\sum_{r=1}^R \mathbb{E}\|\nabla f(\boldsymbol{x}_r)\|^2 \leq & \frac{4(f(\boldsymbol{x}_0) - f(\boldsymbol{x}_R))}{R\eta} + \frac{4\eta L}{m} \left( 1 + \frac{d\kappa + d - 2}{P(d+2)} \right) (\sigma_G^2 + \sigma^2) \\
& + \frac{1}{2}\mu^2 L^2 (d+3)^3 + 4\eta\mu^2 L^4 d^3 \tag{70}
\end{aligned}$$

This completes the first part of the proof.

Selecting $\eta = \mathcal{O}\left( \frac{\sqrt{mP}}{\sqrt{R\kappa}} \right)$ and $\mu \leq \frac{\sqrt[4]{\kappa}}{\sqrt[4]{mRP}\sqrt{(d+3)^3}}$, we can get

$$\frac{1}{R}\sum_{r=1}^R \mathbb{E}\|\nabla f(\boldsymbol{x}_r)\|^2 = \mathcal{O}\left( \frac{\sqrt{\kappa}}{\sqrt{mRP}} \right) + \mathcal{O}\left( \left( \frac{\sqrt{P}}{\sqrt{mR\kappa}} + \frac{\sqrt{\kappa}}{\sqrt{mRP}} \right)(\sigma_G^2 + \sigma^2) \right) \tag{71}$$

Typically $\kappa > P$ so the convergence rate is $\mathcal{O}\left( \frac{\sqrt{\kappa}}{\sqrt{mRP}} \right)$. $\blacksquare$

**Remark 4** The result (70) is intuitive. Besides the terms related to $\mu$, which come from that we use the exact form instead of approximation, it is similar to MeZO's result except for one more term $\frac{1}{m}\text{Tr}(\Delta_{f,r})$, which is corresponding to the data heterogeneity between the clients. The intuition can be gained from the rule of total variance:

$$\text{Var}(\nabla f_i(x, \xi_i)) = \text{Var}(\nabla f_i(x)) + \mathbb{E}_i[\text{Var}(\nabla f_i(x, \xi_i)|i))] \leq \sigma_G^2 + \sigma^2. \tag{72}$$

This implies that our algorithm in $K = 1$ case is equivalent to the MeZO algorithm with larger stochastic gradient noise. In the FL scenario, the effective gradient noise is equivalent to local mini-batch randomness (in-group variance) plus the sampling randomness (between-group variance).

### D.5 THE PROOF OF CONVERGENCE OF DECOMFL WITH $\kappa$-EFFECTIVE RANK; GAUSSIAN CASE

Lastly, we present the case that the $\boldsymbol{z}_{i,p}$ is Gaussian. The main proof idea is that $\|\boldsymbol{z}_{i,p}\|$ can be unbounded so that Assumption 4 cannot be applied directly. Nevertheless, the probability of large $\|\boldsymbol{z}_{i,p}\|$ value decreases exponentially fast. Thus, we can establish the following bound based on two probability events.

**Theorem 5 (Convergence of DeComFL with $\kappa$-Effective Rank; Gaussian)** *Under the assumptions 1, 2, 3 and 4, supposing $\eta \leq \frac{1}{4L}\left(1 + \frac{d\kappa+d-2}{P(d+2)}\right)^{-1}$ and $\boldsymbol{z}_{i,r}$ generated from the standard Gaussian distribution, then it holds*

$$\frac{1}{R}\sum_{r=1}^{R}\mathbb{E}\|\nabla f(\boldsymbol{x}_r)\|^2 \leq \frac{4(f(\boldsymbol{x}_0) - f(\boldsymbol{x}^\star))}{R\eta} + \frac{2\eta L}{m}\left(1 + \frac{d\kappa + d - 2}{P(d+2)}\right)(\sigma_G^2 + \sigma^2)$$
$$+ \eta^2 LG^2 \exp(-\Omega(mdP)) + \mathcal{O}(\mu^2),$$

*where $G$ is defined as the largest value among $\{G(\boldsymbol{x}_r)\}_{r=1}^{R}$ and $\Omega(mdP)$ means some function values that can be lower bounded by $mdP$.*

*Proof.* Let $\mathcal{A}$ be the event that $\|\boldsymbol{x}_{r+1} - \boldsymbol{x}_r\| \leq 2\eta dG(\boldsymbol{x}_r)$. Similarly, we can compute the bound based on the event $\mathcal{A}$ happens and the event $\mathcal{A}$ does not happen:

$$\mathbb{E}_r[f(\boldsymbol{x}_{r+1})] \leq f(\boldsymbol{x}_r) - \frac{\eta}{2}\|\nabla f(\boldsymbol{x}_r)\|^2 + \frac{\eta^2}{2}\left\langle \boldsymbol{H}(\boldsymbol{x}_r), \mathbb{E}\left[\hat{\nabla}f(\boldsymbol{x}_r;\xi_r)\hat{\nabla}f(\boldsymbol{x}_r;\xi_r)^\top\right]\cdot\mathbb{1}(\mathcal{A})+\right\rangle$$
$$+ \frac{\eta^2 L}{2}\|\hat{\nabla}f(\boldsymbol{x}_r;\xi_r)\cdot\mathbb{1}(\mathcal{A}^c)\|^2 + \frac{\eta\mu^2 L^2}{8}(d+3)^3 + \eta^2 L^4\mu^2 d^3$$
$$= f(\boldsymbol{x}_r) - \frac{\eta}{2}\|\nabla f(\boldsymbol{x}_r)\|^2 + \frac{\eta^2}{2}\left\langle \boldsymbol{H}(\boldsymbol{x}_r), \mathbb{E}\left[\hat{\nabla}f(\boldsymbol{x}_r;\xi_r)\hat{\nabla}f(\boldsymbol{x}_r;\xi_r)^\top\right]\right\rangle$$
$$+ \frac{\eta^2}{2}\left\langle L\boldsymbol{I} - \boldsymbol{H}(\boldsymbol{x}_r), \mathbb{E}\left[\hat{\nabla}f(\boldsymbol{x}_r;\xi_r)\hat{\nabla}f(\boldsymbol{x}_r;\xi_r)^\top\cdot\mathbb{1}(\mathcal{A}^c)\right]\right\rangle + \mathcal{O}(\mu^2), \tag{73}$$

where the symbol $\mathbb{1}(\mathcal{A})$ is the indicating functions that is 0 when event $\mathcal{A}$ does not happen and 1 when event $\mathcal{A}$ happens, $\mathcal{A}^c$ stands for the complementary of event $\mathcal{A}$.

Note on the event $\mathcal{A}^c$, we have

$$2\eta dG(\boldsymbol{x}_r) \leq \|\boldsymbol{x}_{r+1} - \boldsymbol{x}_r\| = \eta\left\|\frac{1}{mP}\sum_{i\in C_r}\sum_{p=1}^{P}\boldsymbol{z}_p\boldsymbol{z}_p^\top\nabla f_i(\boldsymbol{x}_r';\xi_{i,r})\right\| \leq \frac{\eta}{mP}\sum_{i\in C_r}\sum_{p=1}^{P}\|\boldsymbol{z}_p\|^2 G(\boldsymbol{x}_r) \tag{74}$$

We conclude that

$$\Pr[\mathcal{A}^c] \leq \Pr\left[\frac{1}{mP}\sum_{i\in C_r}\sum_{p=1}^{P}\|\boldsymbol{z}_p\|^2 \geq 2d\right] \tag{75}$$

Utilizing the i.i.d. property, the right-hand side can be calculated via the Chi-square distribution

$$\Pr\left[\frac{1}{mP}\sum_{i\in C_r}\sum_{p=1}^{P}\|\boldsymbol{z}_p\|^2 \leq 2d\right] = \Pr[\chi_{mdP} > 2mdP] \leq \exp\left(-\frac{mdP}{16}\right), \tag{76}$$

where $\chi_{mdP}$ is the Chi-square distribution with the degrees of freedom $mdP$. Using the same technique used in (Malladi et al., 2023, Lemma 6), we can conclude

$$\frac{\eta^2}{2} \left\langle L\boldsymbol{I} - \boldsymbol{H}(\boldsymbol{x}_r), \mathbb{E}\left[\hat{\nabla}f(\boldsymbol{x}_r;\xi_r)\hat{\nabla}f(\boldsymbol{x}_r;\xi_r)^\top \cdot \mathbb{1}(\mathcal{A}^c)\right]\right\rangle \leq \eta^2 LG(\boldsymbol{x}_r)^2 \exp(-\Omega(mdP)), \quad (77)$$

where $\Omega(mdP)$ means some function value that can be lower bounded by $mdP$. Typically, $mdP$ is a very large value, so this term is vanishing very quickly. Lastly, combining with the proof of Theorem 2, we arrive at the claim of this theorem. ∎

