# OpenReview forum: "Achieving Dimension-Free Communication in Federated Learning via Zeroth-Order Optimization"
_ICLR.cc/2025/Conference — ICLR 2025 Poster_

### Official Review · Reviewer_HVSK · 2024-10-22

**Soundness:** 3
**Presentation:** 4
**Contribution:** 2
**Rating:** 8
**Confidence:** 3

**Summary:**

The authors propose a new approach to reducing communication overhead between a server and clients. They have developed a new algorithm that reduces the overhead using zero-order optimization techniques. Noting that in zero-order optimization, it is sufficient to send scalars between a server and clients (unlike first-order optimization, which requires sending vectors), they introduce a new method, called DeComFL, that works with function values. This method is validated by theory and experiments.

**Strengths:**

The paper addresses the significant problem of communication overhead in the field of Federated Learning (FL). Unlike many previous approaches, which typically focus on reducing the communication overhead of gradients, this paper proposes leveraging techniques from zero-order (ZO) optimization, where only function values, which are scalars, need to be transmitted. The idea is promising (though not new; see Weaknesses). I haven't checked the proofs in detail, but they, along with the final results, appear reasonable and clean.

**Weaknesses:**

Let me point to the weaknesses of the paper:

1. Unfortunately, I'm not sure if the authors are aware, but exactly the same idea to utilize Assumption 4 in the FL setting was in [1]. Albeit, [1] consider the gradient estimator
$$\langle \nabla f_i(\cdot), z\rangle z$$
instead of (3) from this paper (if $\mu \to 0,$ they are equivalent); after that, the idea and the proof techniques are almost the same between this paper and [1].
2. Additionally, the theory from this paper almost replicates the theory [2] adapted to the multi-client setting.
3. As a base method, the authors take the FedAvg method. While it is the most famous FL method in the literature, there are numerous more modern methods that should be discussed and compared to this approach (e.g. [3]).

[1]: Yue, Pengyun, et al. "Core: Common random reconstruction for distributed optimization with provable low communication complexity." arXiv preprint arXiv:2309.13307 (2023).

[2]: Malladi, Sadhika, et al. "Fine-tuning language models with just forward passes." Advances in Neural Information Processing Systems 36 (2023): 53038-53075.

[3] Mishchenko, Konstantin, et al. "Proxskip: Yes! local gradient steps provably lead to communication acceleration! finally!." International Conference on Machine Learning. PMLR, 2022.

**Questions:**

-

---

> ### Author Response · Authors · 2024-11-19
> **Response to Weakness 1**
>
> **Our response to W1 [Difference between DeComFL and Core]:**
>
> Thanks for letting us know about this literature. We should list this reference in the related work. While we acknowledge that the concept that the Core algorithm from [1] utilized the Gaussian projection to save the communication is very similar as we use the ZO method, we would like to emphasize the fundamental differences between our work and [1] in terms of algorithm design, zeroth-order vs first-order, gradient estimation, and theoretical contributions:
>
> 1) **Algorithm design.** The Core algorithm considers a much simpler case that all machines always maintain the same state/parameters. In contrast, DeComFL addresses the more realistic federated learning scenario, that at the same time, different clients may have different model parameters. Hence, we need to address the model pulling problem. Our proposed method eliminates the model-pulling via the local reset, which enables local updates across multiple clients, a feature absent in the Core algorithm. Furthermore, our algorithm explicitly considered client sampling, a defining characteristic of federated learning, whereas the Core algorithm primarily focuses on distributed optimization across a complete set of computing devices. Thus, our approach integrates two key elements of federated learning —local updates and client sampling—neither of which are considered in the Core algorithm. Adapting the Core algorithm to a federated learning setting would require a substantial redesign. Roughly, they need to discard all of their algorithms but keep the random projection step to replace our ZO method approach.
>
> 2) **Zeroth-order vs First-order methods.** Our approach employs zeroth-order optimization (ZOO) by directly estimating the gradient via perturbed function values, effectively leveraging the inherent decomposition of a scalar function value and a perturbation vector. In contrast, [1] employs a first-order method that first computes the gradient and then projects it onto a specific direction for approximation. This distinction highlights the fundamental difference between both the idea and implementation of the two approaches. Furthermore, examining the Taylor expansion $\frac{1}{\mu} [f(x+\mu z) - f(x)] = \nabla f(x)^T z + \frac{\mu}{2} z^T \nabla^2 f(x) z + \cdots$, it becomes evident that [1] is only a first-order approximation of our ZOO approach $\frac{1}{\mu} [f(x+\mu z) - f(x)]$. Hence, [1] is a special case of our algorithm as $\mu \rightarrow 0$.
>
> 3) **Gradient Estimation: Biased vs. Unbiased.** Our method involves a biased gradient estimation due to the nature of zeroth-order methods, with the bias dependent on $\mu$, which is typically a constant in practice. In contrast, [1] provides an unbiased gradient estimation through its projection-based approach (see Lemma 3.1 of [1]). This distinction significantly impacts the theoretical analysis and the practical implications of the two methods.
>
> 4) **Theoretical techniques and conclusions.** 1) Weaker Assumptions: Our analysis does not require the strong assumption of an H-Lipschitz continuous Hessian, which is a key requirement in [1]. 2) Stronger Results: One of our primary contributions is demonstrating a dimension-free rate for the general case $\mu \neq 0$ in non-convex functions, while [1] shows a dimension-dependent rate $\mathcal{O}(d^{3/4})$ (refer to their Theorem 1 and Remark 5.3) Clearly, our theoretical techniques and conclusions differ due to the inherent differences in gradient estimation methods and assumptions.
>
> We appreciate the reviewer’s observation and incorporated a detailed discussion of these distinctions in the revision to clarify the connections and emphasize the broader contributions of our work.

---

> ### Author Response · Authors · 2024-11-19
> **Response to Weakness 2**
>
> **Our response to W2 [The contribution of Theorem 2]:**
>
> We acknowledge that both [2] and our work are in the zeroth-order optimization methods, and both study low-effective rank regimes, and we appreciate the reviewer acknowledging the different settings of our multi-agent setting. We would like to highlight the differences and our key contributions from the perspective of theory.
>
> 1) **General $\mu$ vs $\mu \rightarrow 0$.** One of our main theoretical contributions is providing a dimension-free convergence rate for non-convex functions under general $\mu$. In contrast, the theoretical analysis in [2] is built specifically on the condition $\mu$ and, therefore, cannot capture the impact of the smoothness parameter $\mu$ on the convergence bound.
> Technically, [2] directly assumes the approximation $\frac{1}{\mu} [f(x+\mu z) - f(x)] z = z^T z \nabla f(x)^T$ holds as $\mu \rightarrow 0$. However, this equivalence does not hold for$\mu \neq 0$, where $\frac{1}{\mu} [f(x+\mu z) - f(x)] z \neq z^T z \nabla f(x)^T$. Our analysis addresses this gap by employing the mean value theorem to provide a new proof technique for general $\mu$.
> Even in the centralized learning (single-agent) setting, where $m=1$, our proof differs from [2] due to its applicability to general $\mu$, rather than being restricted to the asymptotic case of $\mu \rightarrow 0$. This contribution was highlighted after Theorem 2 in the main paper, with detailed technical differences and novelty outlined in Appendix C.4.
>
> 2) **Forward-Difference vs. Central-Difference Estimator.** While [2] employs a central-difference ZO gradient estimator $g_{\rm central} = \frac{1}{2\mu}(f(x+\mu z) - f(x - \mu z))$, we utilize a forward-difference estimator $g_{\rm forward} = \frac{1}{\mu}(f(x+\mu z) - f(x)).$ Our choice of the forward-difference estimator is motivated by its computational efficiency in federated learning settings with multiple perturbations. However, this comes at the cost of additional theoretical complexity, as the forward-difference estimator is only asymptotically unbiased ($\mathcal{O}(\mu)$ bias), whereas the central-difference estimator achieves a higher-order bias of
> $\mathcal{O}(\mu^2)$. This necessitated more careful theoretical treatment in our work.
>
> 3) **Federated learning setting.** The theoretical analysis of our DecomFL algorithm in the federated learning (multi-agent) setting introduces more complexities compared to centralized learning. These challenges arise from client sampling noise, multiple perturbations, and the interactions among clients. For example, we developed a new covariance matrix characterization, as shown in Lemma 6, for the stochastic zeroth-order gradient in the federated setting, which required entirely new proof techniques beyond what is used in centralized settings.
>
> In conclusion, while we recognize [2] also explores zeroth-order optimization, our work introduces a new algorithm specifically designed for federated learning with dimension-free communication efficiency. Our contributions include a novel proof framework, theoretical results applicable to general $\mu$, and advances tailored to the federated learning setting. These elements reflect the key differences and the novelty of our approach.

---

> ### Author Response · Authors · 2024-11-19
> **Response to Weakness 3**
>
> **Our response to W3 [More experiments with other FL algorithms]:**
>
> We appreciate the reviewer’s suggestion to expand our comparisons. In response, we have incorporated three more baseline algorithms into our study, including FedProx [4], SCAFFOLD [5] and CyCP [6]. These additions help provide a broader perspective on our results. We did not add the Proxskip because it does not incorporate client sampling, which makes a direct comparison of its communication efficiency with other algorithms (that involve partial client participation) potentially inequitable.
>
> To better clarify the rationale behind our baseline selection, we have categorized the algorithms as follows:
> 1) FedAvg: Classic first-order FL algorithm.
> 2) FedCom, FedAvg+TopK: Gradient compression methods in FL.
> 3) FedProx, SCAFFOLD, CyCP: Modern first-order FL algorithms with acceleration design, as also suggested by the reviewer.
> 4) FedZO: Zeroth-order FL methods.
>
> **All new experimental results have been added to Figure 7 in Appendix B.3.4 of the revision.** The experimental results demonstrate a significant advantage for DeComFL in terms of communication efficiency, aligning with our theoretical analysis of the dimension-free communication rate.
>
> [4] Tian Li, Anit Kumar Sahu, Manzil Zaheer, Maziar Sanjabi, Ameet Talwalkar, and Virginia Smith. "Federated optimization in heterogeneous networks". Proceedings of Machine Learning and Systems, 2:429–450, 2020b.
>
> [5] Yae Jee Cho, Pranay Sharma, Gauri Joshi, Zheng Xu, Satyen Kale, and Tong Zhang. "On the convergence of federated averaging with cyclic client participation". In International Conference on Machine Learning, pp.5677–5721. PMLR, 2023
>
> [6] Sai Praneeth Karimireddy, Satyen Kale, Mehryar Mohri, Sashank Reddi, Sebastian Stich, and Ananda Theertha Suresh. "Scaffold: Stochastic controlled averaging for federated learning". In International conference on machine learning, pp. 5132–5143. PMLR, 2020.

---

> > ### Comment · Reviewer_HVSK · 2024-11-19
> > **Official Comment by Reviewer**
> >
> > Thank you for the long and detailed response. I've read the rebuttal.
> >
> > Minor comment:
> > > Our analysis does not require the strong assumption of an H-Lipschitz continuous Hessian
> >
> > You have Assumption 4, which is almost the same. I guess there are some little differences, but they have the same nature.
> >
> > ---
> >
> > Anyway, this is good work, and I recommend the acceptance of this paper.

---

> > > ### Author Response · Authors · 2024-11-20
> > >
> > > We agree with the reviewer that these assumptions share a similar nature concerning the Hessian. Thank you for your feedback and for your appreciation of our work.

---

### Official Review · Reviewer_Ftuh · 2024-11-03

**Soundness:** 4
**Presentation:** 3
**Contribution:** 3
**Rating:** 6
**Confidence:** 4

**Summary:**

This paper discusses a federated framework using zeroth order (ZO) optimization called DeComFL. It improves the previous ZO method by Fang et al. (2022) because, in each iteration, DeComFL only uses a constant number of bits of communication for each agent.

The authors prove the standard convergence theorem and provide a corollary under the $\kappa$-effective rank assumption. Additionally, they present experiments on training DeComFL, comparing it to the traditional first order method and other zeroth order methods.

**Strengths:**

It is quite novel to see the use of a zeroth order method for federated learning, and this paper makes a valuable contribution to this area.

With small and clever modifications to the previous algorithm by Fang et al. (2022), this research effectively reduces the per-iteration communication costs to a constant for each agent. Supported by both theoretical and experimental evidence, this new method significantly outperforms FedAvg in terms of communications costs.

**Weaknesses:**

The assumption made in Theorem 2 is not very standard. I am not sure if $\kappa$ can be truly seen as $O(1)$ constant and independent from $d$. What will be the consequence if $\kappa$ will scale up with $d$, even if it is not $\Theta(d)$?

Minor:
1. I think the algorithm was stated for $P=1$. When reading pages 4 and 5, $P$ does not appear to be any part of the algorithm. It was confusing what role the constant $P$ plays in the algorithm.
2. In assumption 4, the second maximum should be over $\xi_{i,r}$? Could it be a typo?

**Questions:**

Can you provide any evidence for the low-rank assumption you made?

---

> ### Author Response · Authors · 2024-11-19
>
> > *W1: The assumption made in Theorem 2 is not very standard. I am not sure if $\kappa$ can be truly seen as $\mathcal{O}(1)$ constant and independent from $d$. What will be the consequence if $\kappa$ will scale up with $d$, even if it is not $\Theta(d)$?*
>
> > *Q1: Can you provide any evidence for the low-rank assumption you made?*
>
> **Our response to W1 & Q1 [Support of low-rank assumption]:**
>
> Thanks for the comment. We would like to address the concerns regarding the low-rank assumption and its implications from three perspectives.
> 1) **The low-rank property in deep learning:** The low-rank assumption is both well-recognized and widely adopted in deep learning, particularly in the context of large language model (LLM) fine-tuning. Prior works have consistently observed that the Hessian of the loss function for deep neural networks exhibits a remarkably low effective rank [1–5]. Specifically for LLM fine-tuning, it has been demonstrated that the optimization process often operates within a very low-dimensional subspace [6, 7].
> 2) **Broad adoption in optimization:** This assumption is also broadly utilized in optimization, including both first-order [8] and zeroth-order methods [9]. Its widespread application underscores its practical relevance and theoretical grounding.
> 3) **Empirical evidence from our added experiments:** To support the low-rank assumption in our study, we conducted experiments with four OPT models (OPT-125M, OPT-350M, OPT-1.3B, and OPT-2.7B), whose model dimensions $d$ range from 125 million to 2.7 billion. **As shown in Figure 4 (Appendix B.3.1), all models demonstrate similar convergence rates regardless of their differing model dimensions.** This observation aligns with our theoretical dimension-free rate and provides empirical evidence for the validity of the low-rank assumption. We have included the results and a detailed discussion about the evidence for low-effective rank in the revised **Figure 4 and Appendix B.3.1**.
>
>
> [1] Papyan, Vardan. "The full spectrum of deepnet hessians at scale: Dynamics with sgd training and sample size." arXiv preprint arXiv:1811.07062 (2018).
>
> [2] Ghorbani, Behrooz, Shankar Krishnan, and Ying Xiao. "An investigation into neural net optimization via hessian eigenvalue density." International Conference on Machine Learning. PMLR, 2019.
>
> [3] Papyan, Vardan. "Traces of class/cross-class structure pervade deep learning spectra." Journal of Machine Learning Research 21.252 (2020): 1-64.
>
> [4] Wu, Yikai, et al. "Dissecting hessian: Understanding common structure of hessian in neural networks." arXiv preprint arXiv:2010.04261 (2020).
>
> [5] Yao, Zhewei, et al. "Pyhessian: Neural networks through the lens of the hessian." 2020 IEEE international conference on big data (Big data). IEEE, 2020.
>
> [6] Aghajanyan, Armen, Luke Zettlemoyer, and Sonal Gupta. "Intrinsic dimensionality explains the effectiveness of language model fine-tuning." arXiv preprint arXiv:2012.13255 (2020).
>
> [7] Li, Chunyuan, et al. "Measuring the intrinsic dimension of objective landscapes." arXiv preprint arXiv:1804.08838 (2018).
>
> [8] Yue, Pengyun, et al. "Core: Common random reconstruction for distributed optimization with provable low communication complexity." arXiv preprint arXiv:2309.13307 (2023).
>
> [9] Malladi, Sadhika, et al. "Fine-tuning language models with just forward passes." Advances in Neural Information Processing Systems 36 (2023): 53038-53075.
>
> > *W2: I think the algorithm was stated for $P=1$. When reading pages 4 and 5, $P$ does not appear to be any part of the algorithm. It was confusing what role the constant $P$ plays in the algorithm.*
>
> **Our response to W2 [Multi-perturbations]:**
>
> Thank you for the comment. In Algorithms 1 and 2, we present the DeComFL algorithm with $P=1$ perturbation for simplicity and clarity. Extending the algorithm to handle multiple perturbations is straightforward; it involves incorporating multiple perturbations at each step. The parameter $P$ serves as a control variable for managing zeroth-order gradient estimation variance. As reflected in our convergence rate, $\mathcal{O}\left( \frac{\sqrt{d}}{\sqrt{mPKR}} \right)$, a larger $P$ reduces the estimation variance and accelerates convergence. Experimentally, this is validated in Figure 2, where increasing $P$ demonstrably improves our DeComFL’s performance.
>
> > *W3: In assumption 4, the second maximum should be over $\xi_{i,r}$? Could it be a typo?*
>
> **Our response to W3:**
>
> Yes, it should be $\xi_{i,r}$, and we have modified it in our revised paper. Thank you for pointing out this!

---

> > ### Comment · Reviewer_Ftuh · 2024-11-22
> >
> > Thank you for the discussions and additional experiments.
> >
> > I understand that it could be computationally hard to validate the low rank assumptions, but a direct evidence would be more convincing.

---

> > > ### Author Response · Authors · 2024-11-24
> > >
> > > Thanks for your further comment. We can provide direct evidence of low-effective rank assumption on a small model (i.e., ResNet). Please see Figure 4 (left one) in Appendix B.3.1 of our revision.
> > >
> > > We used the CIFAR10 dataset to train a custom ResNet and observed the **Hessian Eigenvalue Density** by utilizing the approach proposed in [1], which leverages the stochastic Lanczos algorithm [2]. From Figure 4, we can find that the majority of eigenvalues of the Hessian of loss function is 0, which aligns with our low-effective rank assumption.
> > >
> > > [1] Ghorbani, Behrooz, Shankar Krishnan, and Ying Xiao. "An investigation into neural net optimization via hessian eigenvalue density." International Conference on Machine Learning. PMLR, 2019.
> > >
> > > [2] Golub, Gene H., and John H. Welsch. "Calculation of Gauss quadrature rules." Mathematics of computation 23.106 (1969): 221-230.

---

### Official Review · Reviewer_tpD4 · 2024-11-06

**Soundness:** 3
**Presentation:** 2
**Contribution:** 2
**Rating:** 6
**Confidence:** 4

**Summary:**

The paper uses zero-order optimization in federated learning for achieving communication-efficiency. The main idea is that since the update for zero-order optimization consists of (1) random direction sampled from a gaussian distribution and (2) magnitude 1-d value computed on the data, only (2) needs to be transmitted and (1) could be recovered by the server/other clients if they know the random seed, thus significantly reducing communication cost per each round of training.

The paper provides convergence guarantees of their proposed method, particularly showing that if the loss function has small effective rank, then convergence does not depend on the dimension of the problem d, but rather depends on the rank of the loss function.

The paper additionally provides experimental verification of the proposed algorithm.

**Strengths:**

The problem tackled is interesting and important and the proposed method saves a lot of communication (order of 1000s in experiments). Theoretical analysis allows to reason about potential communication savings during the overall course of training. Experiments are done on large models (up to OPT-1.3 B).

**Weaknesses:**

1. The paper does not state how exactly the random seeds are chosen, which might affect the distribution of the generated sequence.
As far as I know, random generators guarantee the distribution of sampling a sequence of numbers from the same generator initialized once at some random seed, however with each number having its own random generator with its own random seed, I am not sure what guarantees exist and I imagine it depends on the distributions of the random seeds and particular implementation of random number generator, i.e. if random seed are deterministically chosen in the increasing order (i.e. the next random seed is equal to s + 1, where s is the previous random seed), then the generated numbers probably won’t follow the gaussian distribution. Also, if the random seeds are sampled from the uniform distribution, it is unclear to me, which distribution will follow the generated vectors.
Therefore, the authors should add a formal statement about the generated sequence of vectors and specify how to generate a sequence of random seeds.

2. In experimental comparison on MNIST, the learning rate is set as the same constant across all the algorithms and settings, which might favor some of the algorithms/settings. For fair comparison it would be better to tune the learning rate separately for each experiment.

3. Experiments on OPT do not compare to the fine tuning with federated averaging. I am wondering, how close to the finetuning with fed avg can zeroth order optimization get.

4. On Fig 3. FedAvg + Topk converges much faster than DeComFL in terms of the number of rounds, and it is only slow on the right plot because k is quite large. I am wondering, if you reduce k, so that FedAvg + Topk converge with similar speed as DeComFL, would ZO optimization still provide substantial communication savings compared to FedAvg + Topk for that smaller k?

**Questions:**

1. What is the difference between the result in Theorem 1 and the prior work that analyzed federated learning with zero-order optimization, e.g. (Fang et al., 2022)? As I understood, algorithmically your method is exactly very similar with only a difference of how direction gradients are samplied & how the communication is performed. Does it pose some extra challenges for the analysis?

2. Why do all the $z_r^k$ are equal on different nodes? I think algorithmically nothing prevents $z_r^k$ to be different on different nodes? The server would just need m times more memory to save all of $z_r^k$. Would such a modification provide a faster convergence?

3. On lines 052-053 paper comments that “because the models become large, communication becomes a bottleneck” howether, for modern models the computation cost can scale quadratically with model dimension, while communication cost scales only linearly. See e.g. [1]. While I do believe communication cost is an important issue in federated learning, I would recommend rephrasing this sentence.

    [1] SWARM Parallelism: Training Large Models Can Be Surprisingly Communication-Efficient, Ryabinin et al.

4. communication cost analysis - might not be accurate?

5. I think $m$ was never introduced in the paper. I understood that $m = | C_r |$, but I didn’t find where it was defined.

6. Could you give an intuition how large the server memory is for training some standard benchmarks? Is it bigger/smaller than saving the full model?

7. I think setting $\mu$ as in Corollary 1 does not give the desired result, as the term $2 \mu^2 L^2 (d + 3)^3$ would still have d at the nominator instead of $\sqrt{d}$.
8. In Assumption 4, which matrix norm do you use?

9. In theorem 2 $d_{\kappa}$ wasn’t defined before.

10. Could you also analyze local steps in Theorem 2? What is the difficulty?

11. I think that condition $\kappa >> P$ could be replaced with just $\kappa > P$.

12. In Theorem 2, why is setting round R sufficiently large allows to remove the $\sigma_G^2 + \sigma^2$ term?

---

> ### Author Response · Authors · 2024-11-19
> **Response to Weaknesses 1-4**
>
> **Our response to W1 [Random seed generation]**
>
> Thanks for your comments. To clarify, our implementation follows a standard **two-step** procedure for the perturbation vector generation: 1) Random Seed Generation: We sample random seeds from a uniform distribution. 2) Random Vector Generation: These seeds are then used to initialize random number generators, which subsequently sample random vectors from a Gaussian distribution. This process ensures that the generated vectors adhere to the Gaussian distribution. The implementation details are explicitly provided in our code and included in the supplementary material. Specifically, line 126 in `cezo_fl/server.py` handles the random seed generation on the server side, while line 74 in `cezo_fl/random_gradient_estimator.py` manages Gaussian random vector generation on the client side. This approach aligns with standard practices for generating random Gaussian vectors, as demonstrated in prior zeroth-order methods. For example, [1] employs a uniform distribution for random seed generation (the source code is `numpy.random.randint(1000000000)` in Line 770 of MeZO/large_models/trainer.py). We thank the reviewer for highlighting this point and have added a detailed explanation of this procedure in Section 3.4 of the revised manuscript.
>
> [1]: Malladi, Sadhika, et al. "Fine-tuning language models with just forward passes." Advances in Neural Information Processing Systems 36 (2023): 53038-53075.
>
> **Our response to W2 [Learning rate selection]:**
>
> Thank you for your valuable suggestion. In the revised manuscript, we conducted a thorough tuning of the learning rates for each algorithm using a grid search and provided an ablation study on learning rates. Specifically, we evaluated model performance using various learning rates $\eta$, including 0.1, 0.05, 0.01, 0.005, and 0.001, to identify the optimal value for each experiment.
>
> The results of this analysis are presented in **Figure 5 in Appendix B.3.2** of the revision. From Figure 5, we observe that a learning rate of 0.1 consistently delivers the best performance among all tested values. Consequently, we updated the results in Figure 2 to reflect the performance achieved with the best learning rate. Despite this adjustment, the conclusions drawn in Figure 2 remain unchanged: **DeComFL maintains a significant advantage in communication savings**. Please refer to the revised manuscript for full details.
>
> **Our response to W3 [FedAvg on OPT models]:**
>
> Thank you for your suggestion. In response, we have included a comparison between FedAvg and our DeComFL across six datasets using two OPT models. The results are presented in the following two tables. For some datasets, such as BoolQ and SST2, the performance gap is negligible. However, for other datasets, there are reasonable performance gaps, which are expected given the significantly reduced communication costs achieved by DeComFL. These results have been incorporated into the revised manuscript for clarity and completeness.
>
> Table 1: Comparison of FedAvg and DeComFL on LLM Fine-tuning Tasks (OPT-1.3B)
> |Dataset|FedAvg|DeComFL($P=10$)|
> |-------|------|-------|
> |SST2|90.38% (1.27 TB)|90.78% (0.24 MB)|
> |CB|83.93% (1.27 TB)|75.71% (0.36 MB)|
> |WSC|65.65% (1.27 TB) |64.16% (0.36 MB)|
> |WIC|65.82% (1.27 TB)|56.14% (0.24 MB)|
> |RTE|66.13% (2.54 TB)|60.89% (1.80 MB)|
> |BoolQ|63.83% (5.08 TB)|62.50% (1.80 MB)|
>
> Table 2: Comparison of FedAvg and DeComFL on LLM Fine-tuning Tasks (OPT-125M)
> |Dataset|FedAvg|DeComFL($P=10$)|
> |-------|------|-------|
> |SST2|87.32% (0.24 TB)|85.08% (0.36 MB)|
> |CB|82.14% (0.12 TB)|75.00% (0.12 MB)|
> |WSC|63.25% (0.12 TB)|59.59% (0.36 MB)|
> |WIC|60.83% (0.12 TB)|53.38% (0.36 MB)|
> |RTE|63.96% (0.48 TB)|57.05% (0.24 MB)|
> |BoolQ|62.34% (0.24 TB)|61.60% (0.24 MB)|
>
> **Our response to W4 [Different K in TopK]:**
>
> Thank you for your thoughtful question. In our paper, we used $k = 0.1d$ (note that $k$ in this response means the parameter $k$ of Top-$k$), identified as the optimal $k$ from our ablation study. If $k$ is further reduced to values such as $k = 0.05d$ or $k = 0.01d$, we observe severe performance degradation, while the communication costs still remain significantly higher than those of our DeComFL. These results have been added to **Figure 6 in Appendix B.3.3** of our revised manuscript for a comprehensive comparison.

---

> ### Author Response · Authors · 2024-11-19
> **Response to Questions 1-5**
>
> **Our response to Q1:**
>
> Thank you for the question. We appreciate the reviewer recognizing the novelty of our algorithm design, which incorporates a carefully designed communication protocol between the client and server, a new synchronization mechanism among clients, and a tailored local update process that balances optimization efficiency, memory usage, and communication overhead. From a theoretical perspective, our analysis in Theorem 1 demonstrates significant differences from the prior work by Fang et al. (2022). A key distinction lies in the assumptions used: Fang et al. rely on a stronger *bounded stochastic gradient assumption*, which simplifies their analysis by enabling straightforward bounding of local updates via the triangle inequality. In contrast, our work adopts standard assumptions, avoiding this unrealistic constraint. This makes it significantly more challenging to bound the local updates, especially given the inherently biased estimation for zeroth-order methods. We address this challenge through Lemmas 3, 4, and 5, which allow us to establish tighter bounds. As a result, our analysis in Theorem 1 achieves an important advantage: it matches the convergence rate of centralized learning in the special case of $m = 1$ (one client)[1], whereas Fang et al.'s bounds do not degenerate to the centralized learning setting, even with their stronger assumptions (refer to their Theorems 1 and 2). This highlights the robustness and generality of our theoretical contributions.
>
> [1] Ghadimi, Saeed, and Guanghui Lan. "Stochastic first-and zeroth-order methods for nonconvex stochastic programming." SIAM journal on optimization 23.4 (2013): 2341-2368.
>
> **Our response to Q2:**
>
> Thanks for the question. We agree with the reviewer that algorithmically, clients could use different random seeds $z_r^k$. However, our design intentionally employs the same random seeds $z_r^k$ across clients to achieve equivalent performance while offering three key advantages over using different random seeds:
> - **Reduced communication between client and server**: By initializing the pseudorandom number generator with a fixed seed and leveraging consistent random sampling methods (e.g., numpy.random.default_rng), identical Gaussian vectors can be reliably generated in sequence on different machines. This means our approach reduces communication to a single fixed seed for the pseudorandom number generator. In contrast, using different random seeds would require transmitting $m\times K \times P$ seeds for a federated learning system with $m$ clients, $K$ local updates, and $P$ perturbations, significantly increasing communication costs.
> - **Reduced memory for server**: Storing the same random seeds requires
> $m$-times less memory on the server compared to managing different random seeds for each client, as the reviewer also noted.
> - **Reduced computation for client model synchronization**: Using the same random seeds simplifies the synchronization of local models. This approach reduces the computational load for model construction by a factor of $m$, as clients do not need to manage and process unique random seed-based perturbations individually.
>
> **Our response to Q3:**
>
> Thank you for pointing this out. We understand the reviewer's concern that computation can also act as a bottleneck. Following the reviewer's suggestion, we have rephrased the sentence for clarity. It now reads: "Given that communication costs scale linearly with model size, they remain a significant challenge in model training and fine-tuning, particularly in FL environments." This revision has been incorporated into the paper.
>
> **Our response to Q4:**
>
> In theoretical computer science, communication complexity(or communication cost) studies the amount of communication required to solve a problem when learning is distributed across multiple parties, which was first proposed in [1]. This metric is also widely adopted in federated learning, where it supports formal communication cost analysis in a theoretical framework (e.g., [2]). We would greatly appreciate further clarification if we misunderstood the reviewer's question.
>
> [1] AC-C, Y. A. O. "Some complexity questions related to distributed computing." Proc. 11th Annual ACM Symposium on Theory of Computing, 1979. 1979.
>
> [2] Khanduri, Prashant, et al. "Stem: A stochastic two-sided momentum algorithm achieving near-optimal sample and communication complexities for federated learning." Advances in Neural Information Processing Systems 34 (2021): 6050-6061.
>
> **Our response to Q5:**
>
> Thank you for your comment. In the paper, $m$ refers to the number of participating clients in each round. This is defined in multiple places, including line 090 of the Introduction, line 5 of Algorithm 1, the last sentence of Theorem 1, and Table 3 in the Appendix. To ensure clarity, we have reiterated this definition explicitly in the Algorithm section in the revised manuscript.

---

> ### Author Response · Authors · 2024-11-19
> **Response to Questions 6-12**
>
> **Our response to Q6:**
>
> Thank you for your insightful question. The memory required on the server is very small. For all six benchmark datasets and two OPT models, the memory usage for our DeComFL approach is at around **1MB**, which is significantly smaller than the memory required to store the full models. For reference, the OPT-125M and OPT-1.3B models require approximately 250MB and 2.6GB, respectively, when stored in 16-bit precision.
>
> We can also provide an analysis of the server's memory requirements. The server in our algorithm needs to store all historical random seeds and gradient scalars. Therefore, the required memory is roughly $4 \times (2 \times R \times P \times K)$ bytes, where $R$ is the number of total rounds, $P$ is the number of perturbations, and $K$ is the number of local update steps. The factor of 2 accounts for the random seeds and gradient scalars, and the factor of 4 assumes 4-byte (float32) storage for each value. In our experiments, the total iterations ($R \times K$) are typically around $10^4 - 10^5$, and we use $P \sim 10^1$. As a result, the total memory usage is approximately $10^6$ bytes or 1MB.
>
> **Our response to Q7:**
>
> Thank you for the comment. We believe there may be a misunderstanding regarding the convergence rate. With the specific setting of $\mu \leq \frac{1}{(d+3)\sqrt{PKR}}$, the convergence rate is $\mathcal{O} \left(\frac{\sqrt{d}}{\sqrt{mPKR}} + \frac{d}{PKR}\right)$, as noted by the reviewer with the $d$ in the numerator of the last term. In Corollary 1, we focus on the asymptotic regime where $R$ is sufficiently large, making the term $\mathcal{O} \left(\frac{\sqrt{d}}{\sqrt{mPKR}}\right)$ dominant over $\mathcal{O} \left(\frac{d}{PKR}\right)$. This is explicitly stated in line 342 of the corollary: "when the algorithm runs for a sufficiently large number of communication rounds $R$." We note that the chosen hyperparameter $\mu$ and the resulting convergence rate are consistent with those in centralized zeroth-order methods.
>
> **Our response to Q8:**
>
> In the paper, we use the $\ell_2$-induced matrix norm as the default, including in Assumption 4. For clarity, we add them explicitly.
>
> **Our response to Q9:**
>
> Thank you for pointing this out. We believe there may be a misunderstanding regarding the notation. The term in Theorem 2 is $d\kappa$, meaning $d \times \kappa$, rather than $d_k$. Both $d$ and $\kappa$ are clearly defined in the paper. To avoid any confusion, we have revised the notation from $d\kappa$ to $\kappa d$ in the updated version.
>
> **Our response to Q10:**
>
> Technically, extending the analysis to include local steps is non-trivial. Lemma 6 relies on a closed-form solution for the covariance matrix of the zeroth-order (ZO) gradient, enabling the exploitation of the low-rank assumption. Introducing local updates disrupts this analysis. The non-linear relationship between parameter updates and gradient expressions necessitates the use of upper bounds, potentially leading to looser bounds and hindering the effective utilization of the low-rank assumption.
>
> Increasing local steps in traditional federated learning reduces communication costs, as the communication per round is independent of local updates. However, in DeComFl, each local step necessitates transmitting gradient scalars, leading to a linear increase in communication overhead. This diminishes the potential benefits of multiple local steps.
> Therefore, while acknowledging the potential relevance of analyzing local steps, we focused our analysis on the single local step scenario due to these inherent complexities within the DeComFL framework.
>
> **Our response to Q11:**
>
> We agree with the reviewer and have fixed it to $\kappa > P$ in our revision.
>
> **Our response to Q12:**
>
> Thank you for the question. We believe that there may be a misunderstanding regarding the use of the $\mathcal{O}$ notation in the convergence rate. In Eq. (6) of Theorem 2, we explicitly include the $\sigma_G^2 + \sigma^2$ terms (constants) to highlight the sources of error in the rate. For the specific regime $\kappa > P$, the convergence rate is expressed in $\mathcal{O}$ notation, where all constants, including $\sigma_G^2 + \sigma^2$, are naturally absorbed.

---

> > ### Comment · Reviewer_tpD4 · 2024-11-21
> >
> > I have read the author’s reply and other reviewer’s comments. Replies clarified most of my concerns. However, some remaining concerns.
> >
> > I agree with the other reviewers that projecting the gradient to the random direction might be a better method with exactly the same computation and communication cost, but better estimate of the gradient. I did not understand the author’s replies that such a method is an approximation of the zero-order method. In my understanding zero-order gradients are finite approximations of the true gradient, introducing a bias with respect to $\mu$, which should go to zero when $\mu = 1$ representing the true gradient. So the convergence rate of projecting a true gradient on the random subspaces should be better than computing a zero-order gradient over the same subspace. Could you explain why that is not the case in your experiments?
> >
> > Response to Q7: why does it matter then that $\mu \leq \frac{1}{(d+3)\sqrt{PKR}}$, why would you not simplify it to $\mu < 1 / \sqrt{R}$?
> >
> > Response to Q10: the communication cost consists of bandwidth and latency. If latency is high, then sending infrequent but larger sized packages should be still better than sending information after each gradient update. Moreover, several numbers could be packed into the same network packet, so that there is no actual difference between the communication time of sending just one integer number and several of them.
> >
> > Response to Q12: In (6) $\sigma_G^2 + \sigma^2$ are not treated as constants in $O$ notation while below it is treated as constants. Could you please clarify which variables are treated as constants in $O$ and which are not.

---

> ### Author Response · Authors · 2024-11-23
> **Our response to DeComFL with gradient projection approach**
>
> **Our response to DeComFL with gradient projection approach**
>
> Thanks for the question. To avoid confusion, we first write the expression of gradient estimation for the zeroth-order method and gradient projection suggested by the reviewer: $g_{ZO} := \frac{1}{\mu} (f(x+\mu z) - f(x)) z$ and $g_{PJ} = \left<\nabla f(x), z \right> z$, where $z$ is the perturbation/projection vector. Also, **we want to emphasize that we compare the ZO gradient with the FO gradient projection instead of the FO gradient itself. Otherwise, it is an unfair comparison since FO gradient cannot achieve the dimension-free communication.**
>
> First, we would like to address two key misunderstandings in the review:
> 1. **Computation Cost: Gradient projection incurs significantly higher computational costs compared to the zeroth-order method.** Calculating gradients requires backpropagation, which is considerably more computationally expensive than the forward pass needed for function evaluations in the zeroth-order method.
> Besides, each step in gradient projection requires an additional projection step $\langle \nabla f(x), z \rangle$, which involves the multiplication of two $d$-dimensional vectors. For high-dimensional models (e.g., $d \sim 10^9$ in the case of LLMs used in our experiments), this extra computation is substantial and cannot be overlooked. Moreover, the gradient projection method also necessitates significantly higher memory requirements.
>
> 2. We believe that zeroth-order gradients cannot represent true gradients when $\mu = 1$.
>
> There seems to be a potential misunderstanding regarding zeroth-order (ZO) methods. Our experiment has shown that the DeComFL framework equipped with the ZO method exhibits a faster convergence than the DeComFL framework equipped with the gradient projection approach. We believe that it is because **the zeroth-order method should not be simply reviewed as the projection of finite approximations of the true gradient**, as suggested by the reviewer. We would like to interpret it in the following two aspects:
> 1. **$g_{ZO}$ is an unbiased estimator of the gradient of a smoothed function** $f^{\mu}$, i.e., $E_z [g_{ZO}] = \nabla f^\mu(x)$. It is a known fact and also shown in our Lemma 2 in Appendix C.1 in our paper that:
> $$\nabla f^\mu(x) = \frac{1}{(2\pi)^\frac{d}{2}} \int \frac{f(x +\mu z) - f (x)}{\mu} z e^{- \frac{1}{2} \|z\|^2} dz = E_z [\frac{1}{\mu} (f(x +\mu z) - f (x))z],$$
> where the smoothed function $f^\mu(x) := \frac{1}{(2\pi)^\frac{d}{2}} \int f (x + \mu z) e^{-\frac{1}{2} \| z \|^2} dz = E [f (x + \mu z)].$ Note that the expectation of $g_{ZO}$ is the exact gradient of this smooth function instead of the finite approximation. The smoother function $f^{\mu}$ has a better function property and curvature to optimize, such as a better Lipschitz condition number (theorem 3.1 (a) of [1]). As a result, it leads to improved convergence properties for $g_{ZO}$. This is verified in our experiments that 1) DeComFL with the ZOO exhibits insensitivity to hyper-parameter learning rate while DeComFL with the gradient projection is unstable for different learning rates, and 2) DeComFL with the gradient projection has a faster convergence.
>
> 2. $g_{ZO}$ is better than the random gradient projection $g_{PJ}$ as $g_{ZO}$ contains more information. $\left<\nabla f(x), z \right>$ gives the directional derivative of $f(x)$ along $z$. This is purely a first-order approximation of how $f(x)$ changes along $z$. In contrast, $g_{ZO}$ uses function values at $x$ and $x+\mu z$ to estimate the rate of change in $f(x)$ along $z$, incorporating higher-order effects. This can be formally shown by Taylor expansion: $\frac{1}{\mu} [f(x+\mu z) - f(x)] = \nabla f(x)^T z + \frac{\mu}{2} z^T \nabla^2 f(x) z + \cdots$. That is, $g_{ZO} = g_{PJ} + (\frac{\mu}{2} z^T \nabla^2 f(x) z + \cdots)z$. Thus, $g_{ZO}$ captures more information about the function's behavior along z than $g_{PJ}$. Hence, we can expect that the ZO method performs better than the gradient projection, which has indeed been validated by our experiments.
>
> In summary, **simply viewing ZO gradients as the projection of finite approximations of the true gradient is an incomplete understanding as it overlooks their inherent smoothing property and the potential for enhanced optimization**. Therefore, viewing ZO gradients as the projection of finite approximations of the true gradient cannot explain the superiority of the zeroth-order method observed in the experiments.
>
>
> [1] Ghadimi, Saeed, and Guanghui Lan. "Stochastic first-and zeroth-order methods for nonconvex stochastic programming." SIAM journal on optimization 23.4 (2013): 2341-2368.

---

> ### Author Response · Authors · 2024-11-23
> **Our response to the follow-up Q7, Q10, and Q12**
>
> **Our response to Q7**: Thank you for the question. Technically, $\mu$ can take any valid value as long as $2\mu^2 L^2 (d+3)^3$ does not dominate the convergence bound. We choose $\mu \leq \frac{1}{(d+3)\sqrt{PKR}}$ because it corresponds well to the $\mu$ commonly used in zeroth-order stochastic gradient descent (SGD). For instance, $\mu \leq \frac{1}{(d+4)\sqrt{T}}$ is a standard choice in centralized learning (see Eq. 3.25 in [1], where $T$ is the number of iterations, which equals $KR$ in our case). This choice ensures consistency with centralized settings, making our results more interpretable and generalizable.
>
> **Our response to Q10**:
> Thank you for the insightful comments. We would like to clarify the definitions of communication costs. In the context of optimization algorithm design for distributed/federated learning, **communication cost is typically defined by the total volume of data exchanged** (e.g., the data size communicated between the server and clients in federated learning), without explicitly considering networking or system-level factors. This definition has been widely adopted in numerous works on distributed and federated learning [1, 2, 3, 4, 5, 6, 7]. Under this definition, our algorithm is designed to minimize the total volume of data exchanged, and our work is the first to provably achieve the dimension-free communication cost for federated learning.
>
> We acknowledge the reviewer’s point that **communication in practice also depends on bandwidth and latency and that multiple numbers could be packed into a single network packet, potentially reducing the practical difference in communication time.** However, this assumption does not always hold, particularly when the amount of data exceeds the maximum transmission unit (MTU), making it impossible to pack all numbers into a single packet. In our work, **DeComFL indeed transmits multiple floating-point numbers** from a client to the server in each round, represented by the perturbation number $P$ in our paper. In our experiments, $P$ ranges from dozens to hundreds and could scale to thousands or more for extremely LLMs. In such cases, the data may exceed the MTU, necessitating multiple packets for transmission.
>
> [1] Wang, Ganyu, et al. "A unified solution for privacy and communication efficiency in vertical federated learning." Advances in Neural Information Processing Systems 36 (2024).
>
> [2] Khanduri, Prashant, et al. "Stem: A stochastic two-sided momentum algorithm achieving near-optimal sample and communication complexities for federated learning." Advances in Neural Information Processing Systems 34 (2021): 6050-6061.
>
> [3] Hönig, Robert, Yiren Zhao, and Robert Mullins. "DAdaQuant: Doubly-adaptive quantization for communication-efficient federated learning." International Conference on Machine Learning. PMLR, 2022.
>
> [4] Zheng, Zhong, et al. "Federated Q-learning: Linear regret speedup with low communication cost." arXiv preprint arXiv:2312.15023 (2023).
>
> [5] Isik, Berivan, et al. "Sparse random networks for communication-efficient federated learning." arXiv preprint arXiv:2209.15328 (2022).
>
> [6] Huang, Xinmeng, Ping Li, and Xiaoyun Li. "Stochastic controlled averaging for federated learning with communication compression." arXiv preprint arXiv:2308.08165 (2023).
>
> [7] AC-C, Y. A. O. "Some complexity questions related to distributed computing." Proc. 11th Annual ACM Symposium on Theory of Computing, 1979. 1979.
>
> **Our response to Q12:**
> Thanks for the question. We believe that there might still be some misunderstanding about $\mathcal{O}$ function. As we explained in our previous response to Q12, $\sigma_G^2 + \sigma^2$ **is** treated as a constant in Eq. (6). However, we explicitly retain it within the $\mathcal{O}$ notation in this equation to highlight the sources of error in the convergence bound—specifically, those arising from stochastic gradient noise and data heterogeneity. This provides **a clearer interpretation of the convergence rate and facilitates comparisons with the rates of zeroth-order methods in other settings, such as centralized learning.**
>
> In convergence analysis for optimization, the focus is on understanding how quickly an algorithm performs over a class of functions/problems. Hence, problem-dependent parameters, such as the smoothness parameter $L$, stochastic gradient variance $\sigma$, and data heterogeneity index $\sigma_G$, are typically treated as constants. The $\mathcal{O}$ function is commonly used to represent convergence rates in terms of the key parameters of the algorithm, with other constants absorbed for simplicity and generality. In our case, the key parameters influening the convergence rate are $m$, $R$, $P$ and $\kappa$, while other parameters, including $\sigma_G$ and $\sigma$, are treated as constants and absorbed into the $\mathcal{O}$ notation.

---

> > ### Author Response · Authors · 2024-11-26
> >
> > We appreciate the reviewer's valuable feedback on our paper. We hope our response and the revision of paper has adequately addressed your concerns of our paper. If you have any questions remain, we are happy to provide further clarification.

---

> > > ### Author Response · Authors · 2024-12-02
> > > **Rebuttal Follow-Up**
> > >
> > > Dear Reviewer tpD4,
> > >
> > > Thank you again for taking the time to review our paper and provide your valuable feedback. As the rebuttal period comes to a close, we would like to kindly check if our responses and clarifications have satisfactorily addressed the concerns you raised in your initial review. If so, we respectfully request you to consider updating your review score.
> > >
> > > If you have any additional questions or require further clarification, please do not hesitate to let us know.
> > >
> > > Once again, we sincerely appreciate your time and effort, and your prompt response would be greatly appreciated.
> > >
> > > Best regards,
> > >
> > > Authors of the paper

---

### Official Review · Reviewer_GgPr · 2024-11-07

**Soundness:** 2
**Presentation:** 3
**Contribution:** 2
**Rating:** 5
**Confidence:** 4

**Summary:**

The authors consider the practical problem of communication costs in federated learning with increasingly large models especially in the era of LLMs. The authors propose DeComFL, which decomposes the local gradient updates from the clients into a scalar magnitude and a pseudo-random perturbation vector. Since the pseudo-random perturbation is recoverable with known seeds, only the scalar magnitude and the random seed are required to be transmitted during each round of FL training, reducing the communication cost to $O(1)$. The author further provides convergence analyses of the proposed algorithm with and without a low effective rank assumption. Empirical experiments show competitive test accuracies and significantly reduced communication costs for DeComFL.

**Strengths:**

1. The paper is generally well-written and has a good flow.
2. The convergence analysis is necessary and duly provided. The discussion on the effective rank assumption to improve the pessimistic convergence bound is interesting. I did not check through the details for the correctness of the proof.
3. The algorithm design is sound.

**Weaknesses:**

1. I am not convinced about the critical role of zeroth-order optimization in the problem setting to reduce communication costs.
2. Parts about the related works and the experiments could be improved, as detailed below in the Questions.

**Questions:**

1. It is unclear to me why zeroth-order gradients are essential in the problem setting: The authors consider the optimization of model parameters which is essentially white-box. The authors are “downgrading” to zeroth-order information for model updates when first-order information is accessible (since the whole model architecture and parameters are known), which may be suboptimal.
2. Related to the question above, the authors claim in Line 63 that “decomposition into a gradient scalar and a perturbation vector” is a unique property of zeroth-order gradients. Please clarify and justify this. Can I achieve a similar effect (eventual convergence, though might be different rates) by projecting the first-order gradient to a specific direction of a perturbation vector?
3. In related works, please clarify the similarities and differences between DeComFL and FedZO? Is DeComFL an extension to FedFL by exploiting the $\kappa$-effective rank assumption?
4. In Table 2, no doubt that the communication saving is substantial. However, the last column is misleading as it has a larger P than FedZO (4th column). I would prefer it removed to avoid confusion.
5. There is no validation of the practical assumption for effective rank. I suggest a comparison of the convergence rate of several differently-sized LLMs. If similar practical convergence rates are observed (with respect to communication cost, I understand the communication costs should be the same for differently-sized models), then $d$ is shown to be pessimistic.

---

> ### Author Response · Authors · 2024-11-19
> **Response to Weakness 1 and Questions 1 & 2**
>
> **Our response to W1 & Q1 & Q2 [Role of ZO to reduce communication costs]**:
>
> We appreciate the reviewer’s insightful questions and comments regarding the role of zeroth-order optimization (ZOO) in reducing communication costs. Below, we jointly address Weakness 1 (W1) and Questions 1 and 2 (Q1, Q2), as they are closely related.
>
> 1. **Practical Value of ZOO Beyond First-Order Methods.** Our work is framed within the context of zeroth-order optimization (ZOO), which has practical relevance in scenarios where gradient information is not readily accessible. Specifically: (1) Black-box model architecture: ZOO doesn't require the knowledge of the model's internal structure, i.e., black-box. (2) Limited memory consumption: ZOO does not need to store gradients and extra activations, thereby having lower memory requirements. (3) Lower computational overhead: ZOO only requires the forward process and thus has cheaper per-iteration computation. We believe it is unfair to evaluate our work solely under a first-order oracle assumption, where gradient information is easily accessible, and conclude that ZOO is suboptimal. Even in white-box scenarios, ZOO remains valuable in settings where first-order approaches may be less suitable. Furthermore, when focusing on communication costs in federated learning (FL), our method demonstrates superiority over existing first-order methods. Specifically, our approach achieves dimension-free communication complexity in theory and significantly reduces communication costs in practice, as evidenced by both theoretical analysis and empirical results presented in the paper. Further discussion is provided below.
> 2. **Decomposition as a Unique Property of ZOO.** We believe that the decomposition of ZOO into a gradient scalar and a perturbation vector is a unique property that enables dimension-free communication. The reviewer raises the possibility of achieving similar effects by projecting the first-order gradient onto a perturbation vector, which is known as random project $\langle \nabla f(x), z \rangle$ ($z$ is the perturbation vector). Mathematically, this random projection is the first-order approximation of our ZOO approach $\frac{1}{\mu} [f(x+\mu z) - f(x)]$, as easily seen from Taylor expansion $\frac{1}{\mu} [f(x+\mu z) - f(x)] = \nabla f(x)^T z + \frac{\mu}{2} z^T \nabla^2 f(x) z + \cdots$. Similarly, we can apply this idea to second-order methods and even higher-order methods by the corresponding second and higher-order approximation of our ZOO. As a result, we can see that our ZOO framework in the paper is a unique and general framework for achieving dimension-free communication design.
> 3. **Empirical Evidence Supporting ZOO’s Superiority over Random Projection.** We implemented the random projection as suggested by the reviewer and compared it to our DecomFL. As shown in Figure 8 (Appendix B.4), our DeComFL consistently outperforms the random projection in terms of convergence. Moreover, we also found that the random projection approach suffers from a narrower range of numerical stability, requiring a smaller learning rate for stable operation. These observations align with our theoretical understanding that the random projection is just a first-order approximation of our ZOO approach.
>
> Besides the above points, we would like to expand on this discussion further, even though it is beyond the scope of our paper. Our work demonstrates that ZOO provides a sufficient approach for achieving dimension-free communication in distributed/federated learning. The reviewer raises a thought-provoking question: **Is our ZOO approach not only sufficient but also necessary for dimension-free communication?** While a rigorous formal analysis is beyond this paper’s scope, we propose an informal argument: 1) A high-dimensional direction vector can be efficiently represented using a seed since there is an implicity condition -- a shared random number generator, enabling compact communication. 2) If the direction encodes model-specific information rather than being random, compressing this $d$-dimensional information into a few scalars seems implausible due to fundamental information-theoretic constraints. This reasoning suggests that reliance on a shared random number generator—and, by extension, a ZOO-style algorithm—may be unavoidable for achieving dimension-free communication. While our argument remains speculative, we hope it provides valuable insight and inspires further research into this fundamental question.

---

> > ### Comment · Reviewer_GgPr · 2024-11-22
> >
> > I thank the authors for the detailed response. I still have doubts about this. When I said projection, I was referring to vector projection: Essentially, we get the gradients of the model parameters and then project this gradient vector to the direction of a perturbation vector (this perturbation vector is determined by the seed in your algorithm). Using this projection still achieves dimension-free communication. I am not entirely sure whether the authors are referring to the same thing in the above rebuttal and also Appendix B.4.
> >
> > To my understanding, zeroth-order gradients (also as the authors wrote in lines 162-163) are approximations of the first-order gradients.
> >
> > Hence, it can be suboptimal to rely on ZOO when we are optimizing/updating the model anyway (i.e., it's white-box anyway).
> >
> > Please clarify if I made a mistake here.

---

> ### Author Response · Authors · 2024-11-19
> **Response to Q3-Q5**
>
> **Our response to Q3 [Comparison with FedZO]:**
>
> Thanks for your comments. Both FedZO and our DeComFL leverage ZOO methods in federated learning. However, our goal of achieving dimension-free communication requires a new algorithm design and novel theoretical insights, which are integral to DeComFL. Importantly, even under the $\kappa$-effective rank assumption, FedZO cannot achieve dimension-free communication, highlighting the fundamental differences between the two approaches.
>
> Specifically, DeComFL is designed to achieve dimension-free communication efficiency. This is accomplished through a well-designed communication protocol between the client and server, a new synchronization mechanism among clients, and a carefully crafted local update process that balances the optimization process and memory usage along with communication efficiency. We show the point-to-point difference between FedZO and our DeComFL.
>
> |Difference Comparison|FedZO|DeComFL(Ours)|
> |------|-----|-------|
> |Server$\rightarrow$client|Send the global model|Send gradient scalars, random seeds|
> |Client$\rightarrow$server|Send local model update|Send gradient scalars|
> | Client side| Standard local ZO gradient descent |Local ZO gradient descent with reset, i.e., nothing changed after client local update|
> |Client Pull Model | Pull model from the server directly | Get the seeds and gradient scalars, then reconstruct the model from the local model.|
> |Random seed|Each sampled client independently generates random Gaussian vector since no seed is transmitted between the client and server.  | Given round $r$, server generates a seed set $\\{s_r^k\\}_{k=1}^K$ for each local update and distributes this set to all sampled clients. It means that in the same round and the same $k$-th local update, each sampled client will use the same seed.|
> |Memory Usage| Client side requires around $2d$ ($d$ is model dimension) memory. The server side requires around $3d$ memory.  | Client side requires aroudnd $d$ memory. The server side requires almost negligible memories. See Appendix A.2 for more detailed explanations. |
> |Global model at server|Required|Optional|
>
> In addition, we have reinforced our discussion of FedZO in the related works section (under the Zeroth-Order Optimization (ZOO) paragraph). This addition clarifies the connections, similarities, and key differences between FedZO and DeComFL, providing a more comprehensive context for our contributions.
>
> **Our response to Q4 [Table 2 Multi-Perturbation column]**:
>
> Thank you for your suggestion. In the revision, we have ensured that the same $P$ ($P=10$) is used for both DeComFL and FedZO in the main paper to maintain fairness and clarity for comparison in Table 2 in the revision.
>
> **Our response to Q5 [Evidence of low-rank assumption]**:
>
> Thank you for this constructive suggestion. In response, we have extended our experiments to include the OPT model series with four different sizes, ranging from 125M to 2.7B parameters. As shown in **Figure 4 in Appendix B.3.1 of our revision, all models converge at nearly the same rate regarding communication rounds.** Specifically, for fine-tuning the SST-2 task, all models converge around the 1250th round. These consistent communication costs across models of varying sizes strongly validate the practicality of our low effective rank assumption. Hence, the $d$-dimensional dependent rate is a quite pessimistic estimation. Additionally, we are extending our results to include more LLMs and datasets, and we will incorporate these findings once they are complete.

---

> ### Author Response · Authors · 2024-11-24
> **More reply about DeComFL with the ZOO versus DeComFL with the gradient projection**
>
> Thanks for your comment. We'd like to clarify the comparison between DeComFL with the ZOO versus DeComFL with the gradient projection (listed as Algorithm 5 in the revision).
>
> First, we would like to confirm the vector projection method proposed by the reviewer. Given two vectors $a$ and $b$, the projection of $a$ onto $b$ is defined as $Proj_b(a) = \frac{\langle a, b \rangle}{\langle b, b \rangle}b$. Accordingly, the gradient projection of $\nabla f(x)$ onto $z$ is $\langle \nabla f(x), z \rangle z$ when $z$ is a unit vector or appropriately scaled. For clarity, we denote the key updates as $g_{ZO} = \frac{1}{\mu} (f(x+\mu z) - f(x)) z$ for the ZOO method and $g_{PJ} = \langle \nabla f(x), z \rangle z$ for gradient projection. If this formulation differs from what the reviewer intended, we kindly request the mathematical formulation, and we would be delighted to address your concerns further.
>
> Second, it is incorrect to conclude that ZOO is suboptimal. While we acknowledge that $g_{ZO}$ is an estimation of the gradient (as elaborated below) and that ZOO understandably converges more slowly than gradient descent, **this does not mean ZOO is suboptimal compared to gradient projection**. Gradient projection is inherently less effective than the gradient itself, and its performance cannot be directly equated to that of gradient descent. **Our extensive comparisons—spanning experiments, theoretical viewpoint, and computational cost evaluations—demonstrate that the ZOO approach outperforms gradient projection.** To provide further clarity, we offer a deeper explanation of ZOO from two key perspectives.
>
> 1. **$g_{ZO}$ is an estimation of the gradient of a smoothed function $f^{\mu}$, i.e., $E_z [g_{ZO}] = \nabla f^\mu(x)$.** It is a known fact and listed in **Lemma 2 in Appendix C.1** in our paper that:
> $$\nabla f^\mu(x) = \frac{1}{(2\pi)^\frac{d}{2}} \int \frac{f(x +\mu z) - f (x)}{\mu} z e^{- \frac{1}{2} \|z\|^2} dz = E_z [\frac{f(x +\mu z) - f (x)}{\mu} z],
> $$
> where the smoothed function $f^\mu(x) := \frac{1}{(2\pi)^\frac{d}{2}} \int f (x + \mu z) e^{-\frac{1}{2} \| z \|^2} dz = \mathbb{E} [f (x + \mu z)].$ Note that the expectation of $g_{ZO}$ is the exact gradient of this smooth function instead of the finite approximation. The smoothed function $f^{\mu}$ has a better function property and curvature to optimize, such as a better Lipschitz condition number (lemma 2 (a)). Thus, it leads to improved convergence properties for $g_{ZO}$. This is verified in our experiments (refer to B.4 in Appendix): 1) ZOO exhibits insensitivity to hyper-parameter learning rate while gradient projection is unstable for different learning rates, and 2) ZOO has a faster convergence.
>
> 2. **$g_{ZO}$ is better than the random gradient projection $g_{PJ}$ as $g_{ZO}$ contains more information**. $\left<\nabla f(x), z \right>$ gives the directional derivative of $f(x)$ along $z$. This is purely a first-order approximation of how $f(x)$ changes along $z$. In contrast, $g_{ZO}$ uses function values at $x$ and $x+\mu z$ to estimate the rate of change in $f(x)$ along $z$, incorporating higher-order effects. This can be formally proved by Taylor's expansion: $\frac{1}{\mu} [f(x+\mu z) - f(x)] = \nabla f(x)^T z + \frac{\mu}{2} z^T \nabla^2 f(x) z + \cdots$. That is, $g_{ZO} = g_{PJ} + (\frac{\mu}{2} z^T \nabla^2 f(x) z + \cdots)z$. Thus, $g_{ZO}$ captures more information about the function's behavior along z than $g_{PJ}$.
>
> Third, as discussed in the previous "Our response to W1&Q1&Q2", we highlight the practical value of ZOO beyond the first-order method like gradient projection. Besides these, we provide further concrete evidence for this claim by analyzing the computational costs associated with gradient projection. For each step, the extra computation required for gradient projection is one backward propagation and one projection step. In high-dimension models (e.g., $d \sim 10^9$ for LLMs), **this extra computation cost is substantial and cannot be ignored. Furthermore, gradient projection methods have significantly larger memory requirements.** To support this point, check our ZOO implementation (see supplementary material), which does not involve `loss.backward()`. The loss is calculated under the `torch.no_grad()` context manager, meaning gradient information is never computed.
>
> Lastly, we would like to emphasize the broader contributions of our DeComFL framework in achieving dimension-free communication. **This is not merely a direct extension of ZOO to FL but a completely novel design tailored for federated learning.** While ZOO constitutes a key algorithmic step, our framework also incorporates other critical components, such as communication design, a seed-sharing mechanism, and model synchronization/reset mechanism. Although much of the discussion has focused on ZOO, the holistic contribution of the entire framework, along with its accompanying theoretical analysis, should not be overlooked.

---

> > ### Author Response · Authors · 2024-11-26
> >
> > We appreciate the reviewer's valuable feedback on our paper. We hope our response and the revision of paper has adequately addressed your concerns regarding ZOO. If you have any remaining questions, we are happy to provide further clarification.

---

> > > ### Author Response · Authors · 2024-12-02
> > > **Rebuttal Follow-Up**
> > >
> > > Dear Reviewer GgPr,
> > >
> > > Thank you again for taking the time to review our paper and provide your valuable feedback. As the rebuttal period comes to a close, we would like to kindly check if our responses and clarifications have satisfactorily addressed the concerns you raised in your initial review. If so, we respectfully request you to consider updating your review score.
> > >
> > > If you have any additional questions or require further clarification, please do not hesitate to let us know.
> > >
> > > Once again, we sincerely appreciate your time and effort, and your prompt response would be greatly appreciated.
> > >
> > > Best regards,
> > >
> > > Authors of the paper

---

### Meta-Review · Area_Chair_DNdU · 2024-12-12

**Metareview:**

The reviews on this paper were a bit mixed, with some concerns about various aspects including zero-order vs projected first-order approaches, experimental aspects, assumptions, and prior work.  For the most part, these concerns were resolved following the discussion.  One reviewer still wanted to see more motivation on whether there are settings of interest where clients have no gradient access and/or projected gradients are too costly.  Another reviewer noted that the connections/comparisons of ZOO vs projection can be improved.

Since the authors did incorporate a fairly significant discussion on this in the revised paper, the decision leans slightly towards acceptance.  Further editing to address these points should also be considered; for that, the following excerpt from the discussion may be useful: *"...authors didn't correctly state a connection between gradient projected to some direction, and zero order gradients over the same direction. I also agree that a better comparison of ZOO in the projection of the true gradient to that direction would be nice to have in the paper that is currently lacking it."*

**Additional Comments On Reviewer Discussion:**

The main discussion points are already outlined above.

---

### Decision · Program_Chairs · 2025-01-22

Accept (Poster)